# Ice flow dynamics of the northwestern Laurentide Ice Sheet during the last deglaciation

Benjamin J. Stoker[1*], Helen E. Dulfer[1,2], Chris R. Stokes[3], Victoria H. Brown[3,4], Christopher D. Clark[2], Colm Ó Cofaigh[3], David J.A. Evans[3], Duane Froese[5], Sophie L. Norris[6] and Martin Margold[1*]

1 Department of Physical Geography and Geoecology, Faculty of Science, Charles University in Prague, Albertov 6, 12800, Praha, Czech Republic.

2 School of Geography and Planning, The University of Sheffield, Sheffield S102TN, UK.

3 Department of Geography, Durham University, Durham, United Kingdom.

4 St Chad's College, Durham, United Kingdom

5 Department of Earth and Atmospheric Sciences, Faculty of Science, University of Alberta, Edmonton, AB, Canada.

Department of Geography, David Turpin Building, University of Victoria, Victoria, V8P 5C2, British Columbia, Canada.

*Correspondence to*: Benjamin J. Stoker (stokerb@natur.cuni.cz) or Martin Margold (margold@natur.cuni.cz)

**Abstract.** Reconstructions of palaeo-ice stream activity provide an insight into the processes governing ice stream evolution over millennial timescales. The northwestern sector of the Laurentide Ice Sheet experienced a period of rapid retreat driven by warming during the Bølling–Allerød (14.7 – 12.9 ka) which may have contributed significantly to global mean sea level rise

during this time. Therefore, the northwestern Laurentide Ice Sheet provides an opportunity to investigate ice sheet dynamics during a phase of rapid ice sheet retreat. Here, we classify coherent groups of ice flow parallel lineations into 326 flowsets and then categorise them as ice stream, deglacial, inferred deglacial or event type flowsets. Combined with ice marginal landforms and a new ice margin chronology (Dalton et al., 2023), we present the first reconstruction of ice flow dynamics of the northwestern Laurentide Ice Sheet at 500-year timesteps through the last deglaciation (17.5 – 10.5 ka). At the local Last Glacial

Maximum (17.5 ka), the ice stream network was dominated by large, marine-terminating ice streams (>1000 km long) that were fed by the Laurentide-Cordilleran ice saddle to the south and the Keewatin Ice Dome to the east. As the ice margin retreated onshore, the drainage network was characterised by shorter, land-terminating ice streams (<200 km long), with the exception of the Bear Lake and Great Slave Lake ice streams (~600 km long) that terminated in large glacial lakes. Rapid reorganisation of the ice drainage network, from predominantly northerly ice flow to westerly ice flow, occurred over ~2000

years, coinciding with a period of rapid ice sheet surface lowering in the ice saddle region. We note a peak in ice stream activity during the Bølling–Allerød that we suggest is a result of increased ablation and a steepening of the ice surface slope in ice stream onset zones and the increase in driving stresses which contributed to rapid ice drawdown. The subsequent cessation of ice stream activity by the end of the Bølling–Allerød was a result of ice drawdown lowering the ice surface profile, reducing driving stresses and leading to widespread ice stream shut-down.

## 1 Introduction

Ice streams are narrow regions of fast flowing ice that exert an important influence on ice sheet mass balance by drawing down ice to lower elevations, where it is lost through melting (surface or basal) or, where there is a marine or lacustrine margin, calving (Bamber et al., 2000; Bennett, 2003; Robel and Tziperman, 2016). As such, ice stream behaviour is a key process in numerical models to project the future evolution of contemporary ice sheets (Gandy et al., 2019). The geomorphological record created by Late Quaternary ice sheets provides a valuable opportunity to investigate ice stream dynamics, including the controls on ice stream activation, acceleration and shut-down (Winsborrow et al., 2010; Margold et al., 2018; Stokes et al., 2016). Crucially, the palaeoglaciological record allows us to investigate these systems over much longer timescales (1000s of years) than the contemporary observational record (10s of years) (Stokes et al., 2016; Gandy et al., 2019). The Laurentide Ice Sheet (LIS) was the largest component of the North American Ice Sheet Complex, which also included the Cordilleran Ice Sheet (CIS) and Innuitian Ice Sheet, during the last glaciation (Dyke et al., 2002, 2003; England et al. 2006; Seguinot et al. 2016; Stokes, 2017; Dalton et al., 2023). At the global Last Glacial Maximum (LGM), the LIS attained a similar extent to the present-day Antarctic Ice Sheet, providing an ideal location to study the behaviour of palaeo-ice streams.

The LIS reached its all-time maximum extent during the global LGM (~26.5 – 19.0 ka) when the western margin coalesced with the CIS for the first time in the Quaternary, creating an ice saddle between the two ice sheets (Levson and Rutter, 1996; Jackson et al., 1997; Bednarski and Smith, 2007; Clark et al., 2009). The LIS reached its maximum areal extent at 22 ka (Dalton et al., 2023). However, there are significant regional variations in the timing of different ice sheet sectors in reaching their maximum position and the northwestern sector of the LIS reached it's maximum extent at ~17.5 ka, after the global LGM (Clark et al., 2009; Dalton et al., 2023). Numerical modelling indicates that abrupt climate warming during the Bølling-Allerød interval (14.7 – 12.9 ka) led to the expansion of the ablation area in the region of the Cordilleran-Laurentide ice saddle (Gregoire et al., 2016). In these model simulations, this resulted in a mass balance-elevation feedback and rapid surface lowering in the ice saddle area, which is hypothesized to have caused rapid global mean sea level rise (Gregoire et al., 2012; Gomez et al., 2015; Gregoire et al., 2016). Recently published chronological constraints from the Northwest Territories identify a period of rapid ice sheet thinning in the ice saddle region during the Bølling-Allerød interval (Stoker et al., 2022). However, this scenario is subject to debate, with Pico et al. (2019) linking the demise of the ice saddle with the shut-down and retreat of the Amundsen Gulf Ice Stream between 13 ka and 11.5 ka.

The rapid collapse of the Cordilleran-Laurentide ice saddle is likely to have had a dramatic impact on the ice drainage network and ice stream activity during the last deglaciation, but this has not yet been investigated in detail (Margold et al., 2015a, b, 2018). A review of geomorphic-based evidence of ice stream activity of the LIS through the last deglaciation (Margold et al., 2015a, b; 2018) shows that there is clear evidence for major ice stream systems that drained the ice saddle and Keewatin Dome of the LIS during the last deglaciation (Figure 1) (Winsborrow et al., 2004; Kleman and Glasser, 2007; Brown et al., 2011, Brown, 2012). However, it remains unclear whether these large ice stream systems operated synchronously over large

distances, or time-transgressively, switching on and off, with their trajectories migrating through time (Brown, 2012; Margold et al., 2015b, 2018). To the south of the saddle, on the southern Interior Plains in Alberta and Saskatchewan, for example, the

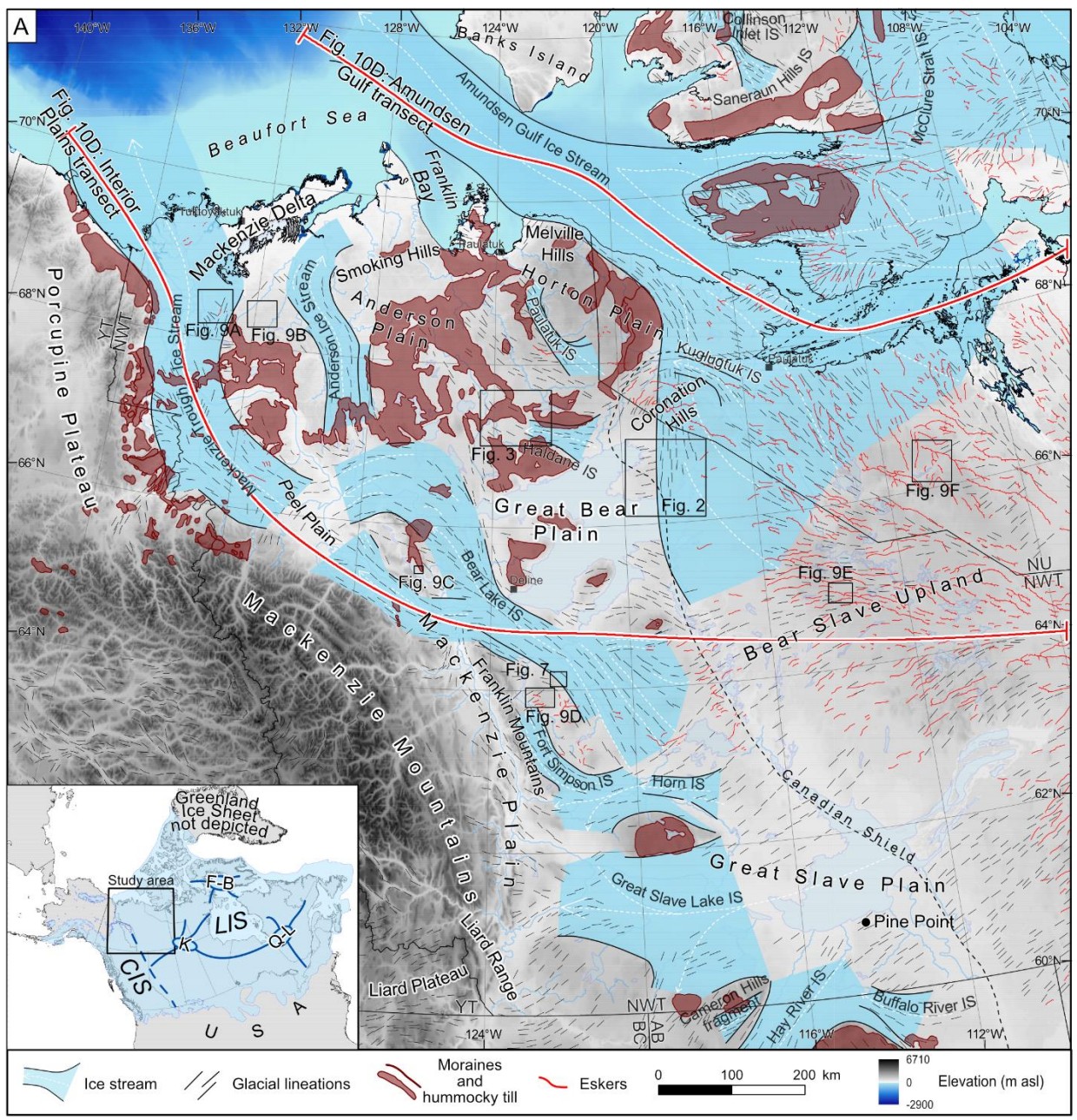

**Figure 1A: The ice stream reconstruction of the northwestern Laurentide Ice Sheet from Margold et al. (2015b) and selected glacial geomorphology (eskers, moraines and lineations) from the Glacial Map of Canada (Prest, 1968; Fulton, 1995). In the inset figure, the North American Ice Sheet Complex extent during the local LGM (17.5 ka) of the northwestern Laurentide Ice Sheet is depicted from Dalton et al. (2023). The approximate location of ice domes and ice divides is based on Margold et al. (2018) (LIS = Laurentide Ice Sheet, CIS = Cordilleran Ice Sheet, K = Keewatin, Q-L = Québec-Labrador dome, and F-B = Foxe-Baffin dome).**

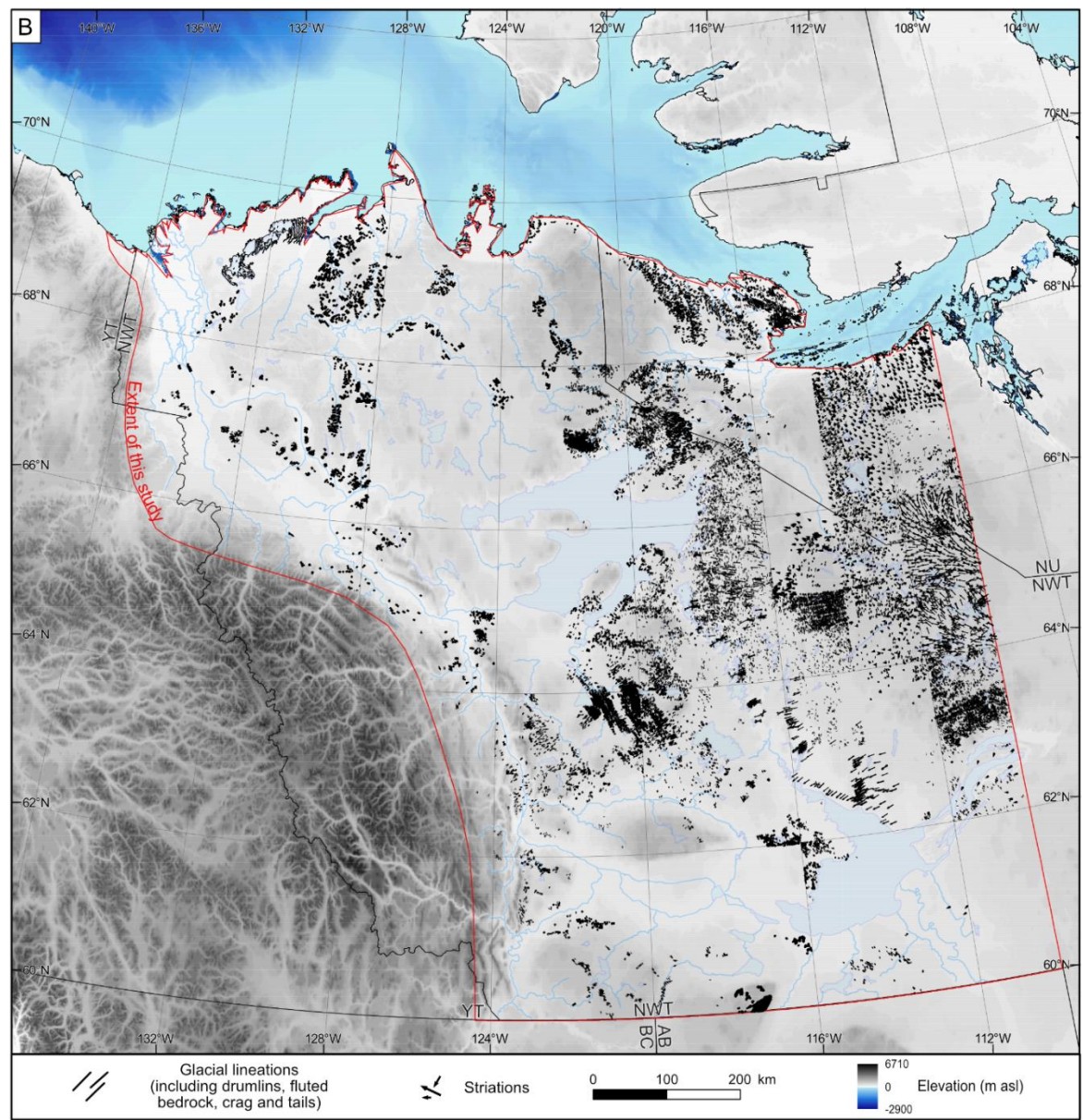

**Figure 2B: A compilation of selected ice flow indicators from previously published maps (Duk-Rodkin, 1989, 1992, 2009a, 2009b, 2010a, 2010b, 2011a, 2011b, 2011c, 2011d; Duk-Rodkin and Couch, 2004; Duk-Rodkin and Hughes, 1992a, 1992b, 1992c, 1992d, 1992e, 1992f, 1993a, 1993b; Duk-Rodkin and Huntley, 2018; Ednie et al., 2014; Evans et al., 2021; Geological Survey of Canada, 2014a, 2014b, 2014c, 2015, 2016a, 2016b, 2016c, 2016d, 2017a, 2017b, 2017c, 2018a, 2018b, 2019a, 2019b, 2022a, 2022b, 2022c; Hagedorn et al., 2022; Huntley et al., 2008; Kerr, 2014, 2018, 2022a, 2022b, 2022c, 2022d, 2022e; Kerr et al., 2014, 2016, 2017a, 2017b; Kerr and O'Neil, 2017, 2018a, 2018b 2019a, 2019b, 2019c, 2020, 2021; Morse et al., 2016; Normandeau and McMartin, 2013; Olthoff et al., 2014; Paulen and Rutter et al., 1980; St-Onge, 1988; Stevens et al., 2017; Smith, 2022; Smith et al., 2021; Veillette et al., 2013a, 2013b) highlighting the progress in surficial mapping across the study region since Fulton (1995). In some locations the lineation pattern may have been simplified to fit the scale of the figure, although the majority of ice flow indicators have been retained to illustrate the variations in mapping resolution across the study area. A high-resolution pdf version of this figure is**

**provided in the supplementary materials folder to allow for the interpretation of the geomorphology which is not possible at the scale produced here (Figure S1).**

geomorphological record presented by Norris et al. (2023) indicates that major shifts in ice flow were associated with rapid downwasting of the Cordilleran and Laurentide ice saddle (Norris et al., 2022), but similar evidence has yet to be identified in the Northwest Territories.

Recent advances in the resolution of remotely sensed data allow us to investigate large areas of the subglacial bed of former ice sheets in detail (Chandler et al., 2018). Dulfer et al. (2023) recently used the high-resolution (2 m) ArcticDEM to produce

a detailed glacial landform map of the northwestern LIS. Here we use this map to resolve the ice flow dynamics across the northwestern sector of the ice sheet. We employ a flowset mapping approach (e.g. Clark 1993, 1997; Kleman et al., 1997, 2006; Hughes et al., 2014) to reconstruct complex ice flow geometries and the relative ice flow dynamics, together with the distribution of ice-marginal landforms to identify former ice margin positions during ice retreat. The 'optimal' ice margin chronology from Dalton et al. (2023) provides a temporal framework for our reconstruction, which allows us to investigate the

following questions:

-   How did the ice stream drainage network evolve spatially and temporally?
-   How did the lobation and thermal regime of the ice margin evolve through deglaciation?
-   How did the coalescence and separation of the Cordilleran and Laurentide ice sheets affect the configuration, timing and dynamics of the ice drainage network?
-   What controlled the activation and shut-down of ice streams of the northwestern LIS?

## 2 Regional glacial history

Despite the remote nature of the region, the northwestern LIS, which spans western Nunavut and the majority of the Northwest Territories in Canada, has a long glacial research history. Since the early 1970s, extensive field surveys and aerial image mapping campaigns undertaken by the Geological Survey of Canada have detailed the surficial geology across the majority of

the former bed of this ice sheet sector (Figure 1b; Hughes and Hodgson, 1972; Hughes, 1987; Rutter et al., 1993). This mapping has been at a range of scales, from detailed 1:50,000 maps (e.g. Bednarski, 2002, 2003a-o) to coarser scale 1:1,000,000 compilation maps (e.g. Duk-Rodkin, 1999; Duk-Rodkin, 2022) and the majority of the work has been conducted at the relatively coarse 1:125,000 scale (e.g. Klassen, 1971; Rutter et al., 1980; Paulen and Smith, 2022). Recent projects have sought to deliver more comprehensive coverage or to improve upon the existing mapping (e.g. Smith et al., 2021; Hagedorn et al.,

2022; Paulen and Smith, 2022). These studies have contributed to a detailed understanding of the former ice sheet limits and retreat pattern across the region, although it remains understudied compared to many other sectors of the former ice sheet (Rampton, 1988; Margold et al., 2018; Duk-Rodkin, 2022).

Our understanding of the northwestern LIS prior to the local LGM is limited by the patchy record of older glaciations. Based on remote sensing evidence, Kleman et al. (2010) identified only a single westerly-directed ice flow event, which they correlated to Marine Isotope Stage (MIS) 4. In the central Keewatin region, a series of six till units from drillcore logs record a counter-clockwise shift in the orientation of the Keewatin Ice Divide (Hodder et al., 2016). The single point nature of these subsurface observations makes it difficult to extrapolate over a larger area or to correlate with former ice margin positions. Evans et al. (2021) integrated sedimentological investigations, geomorphological mapping and cosmogenic nuclide burial dating to reconstruct a detailed glacial history of the Smoking Hills area (Figure 1). This invoked a significant revision of the glacial history of the region, with the concept of an early local glaciation of the Horton Plain by the Horton Ice Cap being rejected in favour of continental glaciation of the region at $2.9 \pm 0.3$ Ma and previous notions of multiple glaciations being replaced by a model of complex glacigenic sequences relating entirely to the Late Wisconsinan. These studies demonstrate that the northwestern LIS was characterized by complex dynamics during the last glacial cycle and the potential for multiple glaciations taking place throughout the Quaternary (cf. Batchelor et al., 2019) but details on the configuration and maximum extent of early glaciations in the region remain poorly constrained.

The configuration of the northwestern LIS since the local LGM is comparatively much better understood. The maximum extent is well established along the eastern range front of the Mackenzie and Richardson mountains (Hughes, 1972, 1987; Duk-Rodkin, 1999, 2022). In the Mackenzie Delta region, early reconstructions suggested that the ice sheet did not extend offshore and placed the local LGM extent at moraines related to the Sitidgi Stade (Rampton, 1982, 1988). However, offshore geophysical surveys have revealed at least two undated advances to the shelf break, the most recent of which has been proposed to relate to the last glaciation (Batchelor et al., 2013, 2014; Riedel et al, 2021). Consequently, the Sitidgi Stade has been reinterpreted as a deglacial ice-margin position (Dalton et al., 2023) as part of a series of well-established deglacial limits (e.g. the Tutsieta and Kelly Lake phases defined by Hughes, 1987).

The advance and retreat of the northwestern LIS led to significant reorganisations of the drainage network and formation of glacial lakes (Smith, 1992; Lemmen et al., 1994; Smith, 1994; Dyke, 2004). As the LIS reached its maximum position along the northern Canadian Cordillera, it dammed the predominantly easterly-flowing fluvial drainage and formed a series of small lakes in the valleys of the Mackenzie Mountains (Duk-Rodkin and Hughes, 1991; Lemmen et al., 1994). In the northernmost region, in the unglaciated areas of the Yukon and beyond the Richardson Mountains, this resulted in the formation of Glacial Lake Old Crow, which has been used to constrain the timing of LIS advance to the local LGM (Lemmen et al., 1994; Zazula et al., 2004; Kennedy et al., 2010). In the early stages of deglaciation, glacial lakes Nahanni and Liard formed near the Liard Range and were dammed by the LIS margin following its separation from the CIS (Bednarski, 2008). Continued deglaciation exposed a series of topographic basins formed by glacial isostatic depression and resulted in the formation of proglacial lakes along the ice sheet margin (Lemmen et al., 1994). This includes glacial lakes Mackenzie and McConnell, which coalesced with Glacial Lake Peace during their maximum stage to form one of the largest Pleistocene glacial lakes in North America (Smith, 1992, 1994; Lemmen et al., 1994). Although the evolution of these lakes has been reconstructed (Lemmen et al., 1994; Dyke, 2004), their influence on the dynamics and retreat of the northwestern LIS are less well known.

The sparse chronological constraints across the region have led to conflicting reconstructions of the timing of deglaciation. Radiocarbon dating has been used to suggest an early local LGM extent at ~30ka in this sector, with a significant readvance at ~22ka (Duk-Rodkin et al., 1996, 2004; Zazula et al., 2004) and a phase of rapid retreat during the Younger Dryas Stade

(12.9 – 11.7ka; Dyke, 2004). The use of luminescence and cosmogenic dating methods, alongside a re-evaluation of the existing radiocarbon constraints using modern standards do not support an early local LGM at ~30ka and instead indicate a much later advance at ~20ka to a short-lived local LGM that lasted only a few thousand years (Kennedy et al., 2010; Murton et al., 2015) with deglaciation beginning at around 18-17ka (Stoker et al., 2022). In this new chronological model, a period of rapid retreat occurred during the Bølling–Allerød interval, with a period of stabilisation of the ice sheet margin on the Canadian

Shield during the Younger Dryas Stade (Reyes et al., 2022; Stoker et al., 2022; Dalton et al., 2023).
Our understanding of the ice flow dynamics of the northwestern LIS suffers from the disconnected nature of studies at varying scales. During the 1990s, the methodology for reconstructing ice flow dynamics using remote sensing data was established and provided fundamental insights into ice sheet drainage network through investigations of the entire LIS bed (Boulton and Clark, 1990; Clark, 1997; Kleman and Glasser, 2007; Kleman et al., 2010). However, this work did not result in a detailed

reconstruction of the ice flow dynamics of the northwestern ice sheet sector, as it did with other sectors (e.g. Clark et al., 2000; De Angelis and Kleman, 2005, 2007; Stokes et al., 2009). Regional scale studies have provided detailed ice flow reconstructions for some regions (e.g. McMartin and St-Onge, 1990; Bednarski, 2008; Evans et al., 2021) but have limited coverage elsewhere. In particular, the Amundsen Gulf Ice Stream has been the subject of multiple studies both onshore and offshore (Stokes et al., 2006; Kleman and Glasser, 2007; Stokes et al., 2009; Batchelor et al., 2014), while the Mackenzie

Valley region has received comparatively little attention since Beget (1987). The glacial geomorphological map of Brown et al. (2011), which covered the entire ice sheet sector, was limited by the resolution of the available datasets at that time (Brown, 2012). The reconstruction of ice stream activity by Margold et al. (2018) integrated the observations from these studies across the whole ice sheet but highlighted large uncertainties in the northwestern sector relating to the influence of the ice saddle on ice flow dynamics.

Recent regional-scale investigations have led to a reinterpretation of the ice flow dynamics of the northwestern LIS. In the Smoking Hills-Horton River area (Figure 1), field observations of glacial stratigraphy and broader-scale geomorphological mapping have highlighted the nature of complex decoupling of ice margins relating to ice flow units derived from the south, east and southwest, with moraine belts recording the establishment of polythermal, debris-charged snouts during overall recession (Evans et al., 2021). South of Great Slave Lake, the formation of the Cameron Hills ice stream fragment had been

tentatively interpreted as a deglacial flowset by Margold et al. (2018) but the till characteristics and fragmented nature of this flowset have led to a reinterpretation that implies a much earlier formation when the CIS and LIS merged (Hagedorn, 2022). Till fabrics at Pine Point, on the southern shore of Great Slave Lake, provide evidence for early ice flow to the SW across this region (Sapera, 2023). The coalescence of the LIS and CIS resulted in the NW ice flow during the local LGM, until separation of the ice sheets led to the re-establishment of ice flow to the WSW during deglaciation (Sapera, 2023). These studies provide

a detailed record of relatively local glacial histories and have significantly improved our understanding of the regional ice flow

dynamics generally but are ultimately limited in extent. This means that large uncertainties still exist in our understanding of the changes in the ice drainage network at the ice sheet sector scale.

## 3 Methods

Our landform-driven reconstruction is based on the glacial geomorphological map of the northwestern sector of the LIS by
Dulfer et al. (2023) which followed a consistent, manual mapping approach across the entire region and was verified against surficial geological maps. As such, this reconstruction acts as a companion paper to Dulfer et al. (2023). This recent mapping facilitates a consistent ice flow reconstruction across the entire study area at relatively high-resolution, albeit with some inherent limitations (discussed in Section 3.4). The map contains twelve landform categories, including ice flow parallel lineations, subglacial ribs, crevasse fill ridges (geometric ridge networks), major and minor moraine crests, hummocky terrain
complexes and ridges, shear margin moraines, major and minor meltwater channels, lateral and submarginal meltwater channels, esker ridges and complexes, glaciofluvial complexes, perched deltas, raised shorelines and aeolian dunes. Here we use the spatial distribution and characteristics of these glacial landforms to determine how ice flow and the related ice margins evolved over time, as outlined below.

### 3.1 Flowset mapping

We use the ~76,000 ice flow parallel lineations mapped by Dulfer et al. (2023) to decipher ice flow variation over time following the established approach of glacial flowset mapping (e.g. Clark 1993, 1997; Kleman et al., 1997, 2006; Greenwood and Clark, 2009; Hughes et al., 2014). We first reduced the amount of lineation data by drawing generalized ice flow lines parallel to the lineations and then used these flowlines to group coherent patterns of lineations into discrete flowsets (Kleman and Borgström, 1996; Kleman et al., 2006; Figure 2). We acknowledge that this process is subjective and some flowsets may
have formed time-transgressively, but the different flow phases could not be distinguished in the landform record. To reduce the subjectivity, flowset delineation was completed by two independent researchers and then the results compared to reach the final flowset map. Each flowset was assigned a number and the characteristics of the flowset were recorded, including the broad flowset morphology (i.e. converging or diverging), the ice flow direction, the location of cross-cutting lineations and associated glacial landforms (see Table S1). During the creation of the final flowset reconstruction, some flowsets were deleted,
merged or split, leading to some flowset numbers being absent from the final map (e.g. Fs 210, 211, 264).

Flowsets were classified into one of the following categories based on cross-cutting relationships and an assessment of the diagnostic criteria listed in Table 1: ice stream, deglacial, inferred deglacial and event flowsets (see also Figure 2). The ice stream flowset category represents areas of past fast ice flow and are principally defined based on the elongation and parallel conformity of lineations, and the abrupt lateral edge of the flowset (Clark, 1993, 1999; Stokes and Clark, 1999; Kleman et al.,
2006). Ice stream flowsets can form in any position within an ice sheet, either near to the margin or towards the interior (Table 1, Figure 2e). In contrast, both event and deglacial flowsets formed under a slower ice flow regime (Table 1) and are placed

into subcategories based on their location of formation within the ice sheet (Figure 2e). Deglacial flowsets formed near to the ice sheet margin and are identified based on their association with other deglacial landforms (e.g. eskers or moraines, Table 1). Conversely, inferred deglacial flowsets are not directly associated with deglacial landforms but are suggested to have formed near the ice sheet margin. Rather, the proposed ice-marginal formation of inferred deglacial flowsets is based on fan-shaped lineation patterns or a clear topographic influence on lineations (Table 1). Event flowsets are interpreted to have formed within the interior of the ice sheet and are identified based on the absence of, or discordance with, deglacial landforms and the high parallel conformity of lineations and their often fragmented nature (Table 1). As such, ice stream flowsets can form in either an 'event' or 'deglacial' position relative to the ice sheet margin, but are not placed into subcategories based on their location of formation. Where there was insufficient information to classify the flowset, it was labelled as unclassified and not used in our reconstruction.

**Table 1: Diagnostic criteria for flowset classification (modified from Kleman and Borgström, 1996; Stokes and Clark, 1999; Kleman et al., 2006; Greenwood and Clark, 2009; Hughes et al., 2014).**

| Flowset type | Characteristics | Timing of formation within the flowset | Spatial classification/position in ice mass |
|---|---|---|---|
| **Ice stream** | - Highly attenuated bedforms.<br>- Highly convergent flow patterns.<br>- Abrupt lateral margins.<br>- Lateral shear margin moraines.<br>- Dispersal trains.<br>- Lineations may be overprinted. | Synchronous or time transgressive | Ice marginal or interior |
| **Deglacial** | - Lineations aligned with eskers<br>- Lineations associated with ice contact landforms, such as moraines.<br>- Lineations influenced by topography.<br>- No overprinting landforms. | Synchronous or time transgressive | Ice marginal |
| **Inferred deglacial** | - Lineations influenced by topography.<br>- No overprinting landforms.<br>- Fan-shaped lineation pattern | Synchronous or time transgressive | Ice marginal |
| **Event** | - High parallel conformity.<br>- Lineations without meltwater landforms (e.g. eskers). | Synchronous | Interior |

| | | No association with moraines. | | |
| | | Abrupt lateral margins. | | |
| | | Lineations may be overprinted. | | |


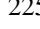

Figure 2: Example of the glacial flowset mapping methodology. (A) ArcticDEM-derived hillshade imagery (Porter et al., 2018), (B) ice flow parallel lineations from Dulfer et al. (2023), (C) generalisation of the lineation data (green) with the ice flow parallel lineations underneath (black), (D) grouping coherent patterns of lineations into discrete flowsets
with the generalised lineation data underneath (green). See Table 1 for the diagnostic criteria for classifying the

**flowsets. Arrows show the ice flow direction where it could be determined and the flowset numbers correspond to the numbers listed in Table S1. Note the inferred deglacial flowset category is missing from this example. The location of this figure is shown on Figure 1. (E) Hypothetical ice sheet showing where the deglacial and event flowsets are formed within the ice body. Ice stream flowsets may form in either an 'event' or 'deglacial' position. The dashed blue line represents the ice divide, and the blue arrows show ice flow direction.**

### 3.2 Deglacial ice margin retreat pattern

The retreat pattern of the northwestern LIS is defined by a range of deglacial landforms, including moraines, hummocky terrain, eskers and lateral meltwater channels (Dulfer et al., 2023). Here we use this landform record to delineate past ice margin positions and inferred ice margin positions across our study area (Figure 3; Table 2) following protocols employed by Greenwood et al. (2007) and Clark et al. (2012, 2022). The inferred ice margins are necessarily subjective because they typically include extrapolation between geomorphological evidence. In some places this may be the extrapolation from a moraine or meltwater channels to depict an ice lobe based on glaciological process-form relationships and the influence of local topography. Additionally, we mark the former ice flow direction associated with each ice margin wherever it could be determined using the glacial landform record. While glacial lakes are known to have formed along the retreating margin of the northwest LIS, we do not reconstruct glacial lakes here. Instead, we use the broad-scale reconstruction of lakes in Dyke et al. (2003) to make preliminary comparisons between ice stream activity and the location of large proglacial lakes.

### 3.3 Temporal framework

When we build our reconstruction of ice flow dynamics through deglaciation we use the 'optimal' ice margins from NADI-1 as a temporal framework for our reconstruction (Dalton et al., 2023). NADI-1 consists of isochrones of the entire North American Ice Sheet Complex from 25 ka to present in 500-year intervals based on all available geochronological constraints. Our geomorphology-based ice margin positions (see section 3.2) add considerable detail to the margin retreat pattern when compared with NADI-1. This is because they are not based on 500-year timesteps and so better capture the detail of the ice retreat and separation of multiple ice lobes in between the regular 500-year time-steps. However, as most of the geomorphology-based ice margins are not directly dated, it is not possible to provide a more detailed chronological placement for them beyond the temporal framework of NADI-1 (Dalton et al., 2023). In summary, we do not adjust the ice margin outline of the NADI-1 isochrones, as to do so would require a re-examination of the chronological constraints on deglaciation, which is beyond the scope of this paper (cf. Dalton et al., 2023). Instead, we provide our geomorphology-based ice margins as a supplement to this temporal framework to aid in the understanding of ice flow evolution changes.

**Table 2: Description of ice marginal landforms used to map former ice marginal positions during the last deglaciation.**

| Landform | Description | Formational process | Reference | Example |
|---|---|---|---|---|
| Terminal moraine | Sharp-crested straight or arcuate-shaped ridges of sediment. | Form by deposition by bulldozing or transit to the margin or deformation of glaciogenic sediment along active ice margins. | Benn and Evans (2010) |  |
| Lateral meltwater channel | A series of parallel or subparallel channels that all dip in the same direction. Often occur as a series of channels perched on valley sides. | Form by water flowing along the ice margin. Sequences of lateral meltwater channels can delineate ice sheet surface lowering and ice marginal retreat through time. | Mannerfelt, (1949), Greenwood et al. (2007, 2016). |  |
| Hummocky Terrain | Irregular surface containing mounds of sediment alternating with depressions. Could also be referred to as controlled moraines. | Form at a stagnating ice margin by a combination of processes, including the formation of sharp-crested ridges by the bulldozing of sediment during ice margin readvances or the formation of flat-topped surfaces within ice-walled lake plains. | Dyke and Evans (2003), Evans et al. (2021) |  |

| Esker | Sinuous depositional ridges of glaciofluvial sand and gravel. | Deposited by meltwater flowing through, beneath or above ice. Eskers form time-transgressively normal to the ice margin. | Shreve (1985), Hebrand and Åmark (1989), Storrar et al. (2014a). |  |
|-------|------|------|------|------|

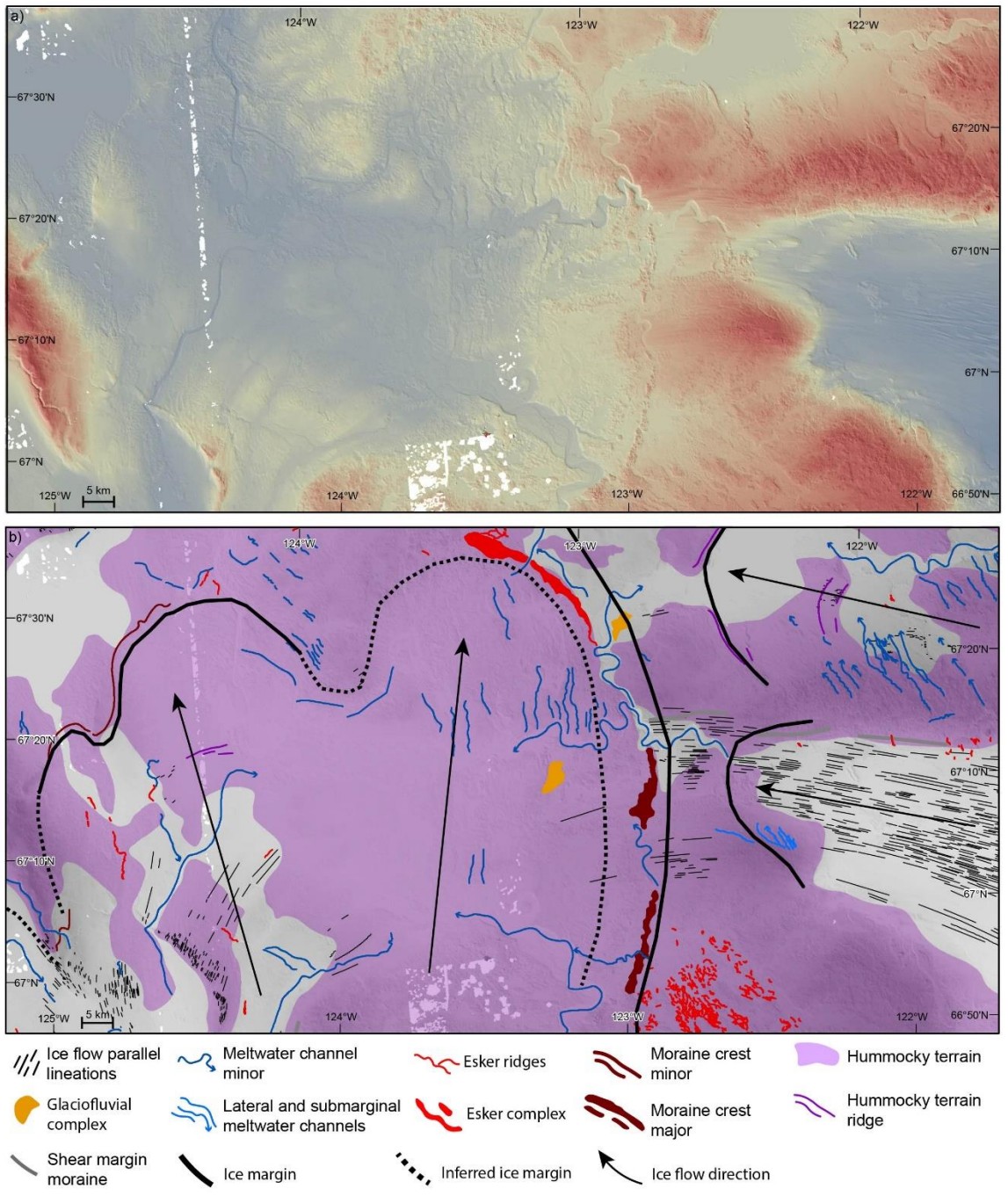

**Figure 3: Delineating former ice margin positions from the glacial landform record. (A) ArcticDEM-derived hillshade imagery (Porter et al., 2018) and (B) glacial geomorphological mapping from Dulfer et al. (2023). The ice contact landforms, including major (>200m wide) and minor (<200m wide) moraines, hummocky terrain ridges, single esker ridges (polyline), esker complexes (polygon) and lateral and submarginal meltwater channels, have been used to draw the ice margin and inferred ice margin positions, which are shown by the black solid and dashed lines, respectively.**

**The ice flow direction is shown by the black arrows. This is an example of interlobate ice margins. Note how the main**
**deglacial landforms (moraines and hummocky terrain) are overprinted on the westwards ice flow signature. The location of this figure is shown on Figure 1.**

### 3.4 Limitations and uncertainties

The reconstruction of Late Quaternary glaciations from remote sensing data is a low-cost and relatively time-efficient method for covering large areas, but also includes fundamental limitations irrespective of the used data resolution (Chandler et al.,
2018). In this study, we reconstruct the ice flow evolution of the northwestern LIS based on the glacial geomorphological map of Dulfer et al (2023). This map is composed solely of glacial landforms visible at the earth surface from the ArcticDEM (2 m resolution; Porter et al., 2018). Despite the high-resolution of these data, the landform record has clear limitations when trying to reconstruct the glacial history in areas with complex, changing flow patterns or a history of multiple cycles of glaciation. The most recent glaciation often erodes and reworks the existing surficial record, meaning that the surface geomorphology is
only a snapshot of the last landform-creating glacial event over a region and not a composite of the entire glacial history. The last glaciation represents the all-time maximum extent of the northwestern LIS, so there is limited geomorphological evidence of older glaciations at the surface. The subsurface record may retain a signal of earlier glaciations and can often be used to reconstruct the longer-term history or better constrain the relative sequence of flow events (Hodder et al., 2016). Multiple studies (e.g. Bednarski, 2008; Stokes et al., 2006; Evans et al., 2021), alongside surficial geological mapping campaigns by
the Geological Survey of Canada (e.g. Hagedorn et al., 2022; Smith et al., 2022), provide more detailed, but more narrowly-focused reconstructions of the glacial history. Glacial striations also provide an opportunity for reconstructing former ice flow patterns and may preserve older flow traces on bedrock outcrops where abrasion was limited during deglaciation (Kleman, 1990). The key advantage of our reconstruction is that we reconstruct the ice flow dynamics at the ice sheet sector scale to gain an insight into the LIS evolution during deglaciation.
The isochrones of Dalton et al. (2023) for the northwestern LIS are largely anchored by cosmogenic nuclide exposure ages. Cosmogenic nuclide exposure dating includes uncertainties of up to 10% (Stoker et al., 2022) of the measured age, meaning uncertainties can exceed 1,000 years for samples from the LGM (Stoker et al., 2022). As there is no universal method for calculating exposure ages, a range of different calibration data, scaling schemes and correction factors may be applied and the chosen approach can lead to calculated exposure ages that differ by over 1,000 years (Stoker et al., 2022; Reyes et al., 2022).
However, the rates of retreat are typically consistent between exposure age calculation approaches, and it is the timing of deglaciation that is shifted. As our ice flow reconstruction is constrained by cosmogenic nuclide exposure ages it is subject to change due to advances in our understanding of cosmogenic nuclide exposure age dating.

# 4 Results

## 4.1 Ice flow reconstruction

We identified 326 flowsets across the bed of the northwestern LIS (Figure 4; see Figure S2 for A0 version where all flowsets are labelled). This includes 62 deglacial, 133 inferred deglacial, 53 ice stream, 33 event, and 46 unclassified flowsets. These flowsets range in size from a few $km^2$ to over 80,000 $km^2$ (e.g. Fs-249). The 2 m resolution of the ArcticDEM allowed us to classify lineations into separate flow events in greater detail than the previous broad-scale work (e.g. Margold et al., 2018). In this section, we provide a broad overview of our flowset map (Figure 4). A detailed description of individual flowsets can be

found in Table S1 and all shapefiles are available in the Supplementary Folder. We then place this ice flow evolution into a temporal framework in Section 5.

### 4.1.1 Inferred deglacial flowsets

Inferred deglacial flowsets are the most common flowset type (*n*=133) across the study region. Most of the flowsets in this category are small, with only one flowset exceeding 3,500 $km^2$ in area. Unlike deglacial flowsets, these flowsets are not directly

associated with any deglacial landforms. However, they commonly exhibit a lobate geometry or display deflections around the local topography, which is indicative of a thinner ice sheet and formation near the ice sheet margin (Kleman and Hättestrand, 1999; Kleman et al., 1999; Hughes et al., 2014). The geomorphic imprint of these flowsets typically displays well-preserved lineations that are not overprinted by other landforms. These flowsets are more commonly found at lower elevations, along the valley floors, and are rarely located on summits, ridgelines or elevated plateaus. South of 62°N, these flowsets are

broadly oriented towards the southwest. North of 62°N, the dominant orientation is towards the west or northwest. Similar to the deglacial flowsets, there is a strong topographic influence on flowset orientation which results in local-scale complexity. In the central Mackenzie Valley, between 63°N and 66°N, this influence is clear, with multiple flowsets deflected around the Franklin Mountains and funnelled up the Mackenzie Valley to the south (Fs 24, 25, 27, 30, 31, 32, 33, 47, 80), against the direction of flow indicated by adjacent flowsets in the Mackenzie Valley to the north and south that appear largely uninfluenced

by topography. Towards the Canadian Shield, in the east, ice flow is oriented more broadly towards the west.

### 4.1.2 Deglacial flowsets

Deglacial flowsets are the second-most prevalent flowset type (*n*=62) and are observed across the whole region (Figure 4). The lineations in this flowset-type are generally well preserved and not overprinted by other landforms. However, the

northerly-oriented deglacial flowsets tend to be overprinted by westerly-oriented deglacial flowsets. The complexity of these overprinting patterns is described in Section 5. They exhibit the largest range in flowset size. The smallest flowset is ~5 $km^2$ and the largest is just over 80,000 $km^2$. There are two main flowset geometries: a lobate flowset pattern of diverging lineations

and a flowset geometry composed of parallel lineations. The lineation pattern in these flowsets is often influenced by the local topography.

On the Northern Interior Plains, the deglacial flowsets are predominantly composed of drumlins, which can be parallel or diverging in a lobate pattern. These flowsets display a range of orientations. In the foothills of the Mackenzie Mountains, their orientation is strongly influenced by the local topography. For example, in the central Mackenzie Valley, north-oriented flowsets were formed when the ice surface slope was the dominant control on ice flow direction underneath a thick ice sheet during the local LGM. In contrast, the adjacent south-oriented flowsets record the increasing importance of the topographic

relief in funnelling ice flow up the Mackenzie Valley during deglaciation, as the ice sheet thinned.

### 4.1.3 Ice stream flowsets

Ice stream flowsets ($n$=53) are composed of highly attenuated subglacial bedforms, including elongate drumlins and mega scale glacial lineations (MSGL; elongation ratio of >10:1). They exhibit a wide range of sizes, from small flowset fragments (~15 km$^2$) to large flowsets up to 40,000 km$^2$. The geometry of the larger flowsets is varied, with converging (e.g. Fs 10, 129,

239, 284), diverging (e.g. Fs 55, 62, 154, 173) and hourglass-shaped (e.g. Fs 65, 70, 115, 138, 206) lineation patterns. Ice stream flowsets are observed across the entirety of the Northern Interior Plains region in the west but are absent from the Canadian Shield region in the east (Figure 4). The dominant orientation of ice stream flowsets is to the north and the northwest ($n$=38). There are also multiple ice stream flowsets indicating ice flow to the west ($n$=15). In general, the westerly-oriented ice stream flowsets are superimposed on, or cross-cut, the northerly-oriented flowsets.


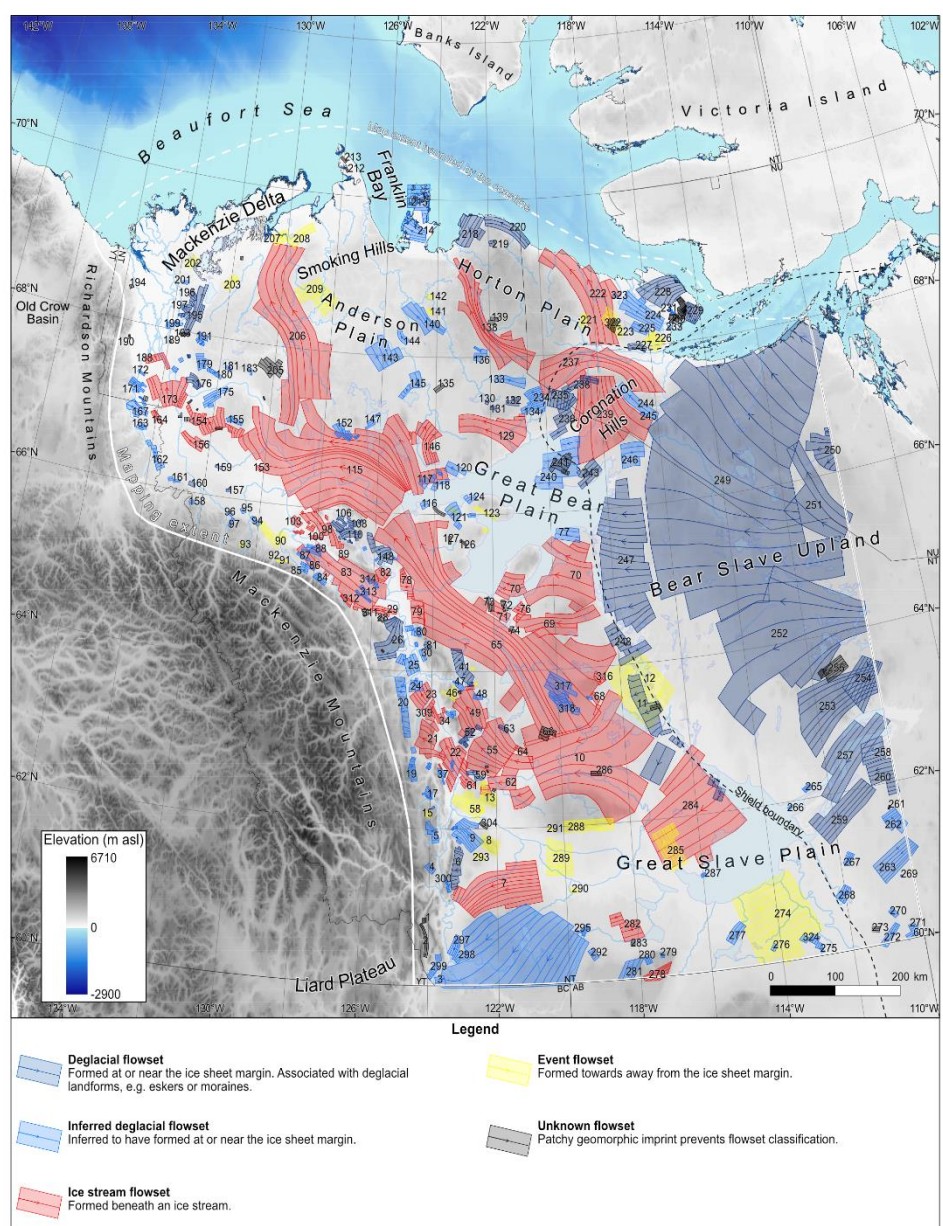

**Figure 4: Flowset map of the northwest sector of the Laurentide Ice Sheet. Flowsets have been assigned a number with corresponding information available in Table S1. Due to the scale of the figure, only the largest flowsets are labelled in this figure. An A0 version of this map is provided in the supplementary materials, which includes labels for each flowset and information on the cross-cutting of different flowsets (Figure S2). The flowsets are classified according to the criteria in Table 1.**


On the Canadian Shield, crag-and-tails and streamlined bedrock are the dominant lineation type although some pre-crag features are also observed. These lineations are generally shorter than those observed on the Northern Interior Plains. Flowsets on the Canadian Shield tend to form a large, diverging flow pattern of subparallel lineations that are broadly oriented towards the west. There is overprinting of the large diverging flow pattern (Fs 249, 252) by smaller flowsets, which appear to be funnelled by the local topography.

### 4.1.4 Event flowsets

Event flowsets are the least common flowset ($n$=33) and are composed of short drumlins and streamlined bedrock. Lineations display a high parallel conformity regardless of the local topographic factors. Where these flowsets are located at lower elevations they are often overprinted by other flowsets (e.g. Fs 12, 274, 285) and at higher elevations they are typically well-preserved, with little overprinting (e.g. Fs 48, 288). The majority of event flowsets are smaller than 1,000 km$^2$, but they can be up to 11,000 km$^2$. There is no dominant orientation of event flowsets which likely indicates these flowsets are formed over different events and times of glaciation.

### 4.1.5 Unclassified flowsets

Unclassified flowsets ($n$=46) are generally composed of poorly preserved lineations with a patchy geomorphic signature. These flowsets are often overprinted by other glacial landforms or disturbed by postglacial processes. They are mostly small (less than 30 km$^2$) and with no clear preferred orientation. A small number of the unclassified flowsets have a clear geomorphic signature, but the absence of any relationship with other flowsets or landforms make them difficult to classify or place in a relative time sequence (Fs 1, 2).

### 4.1.6 Comparison of flowsets with previously-published work

The ice flow patterns of the northwestern LIS have been reconstructed at various scales in previous work. This includes multiple broad-scale reconstructions that cover the entire study area (Kleman et al., 2010; Shaw et al., 2010; Margold et al., 2015a, b, 2018) and one unpublished flowset reconstruction by Brown (2012), which covers almost the entire area of this study. Our flowsets were mapped independently of previous work, but we note whether each flowset has been previously mapped in Table S1.

Margold et al. (2015a, b, 2018) mapped past ice stream activity at the ice sheet scale and their mapping matches 64% of our mapped ice stream flowsets (based on the number of our ice stream flowsets that are located within those of Margold et al. (2015a, b, 2018). Our reconstruction provides some additional detail compared to that of Margold et al. (2015a, b, 2018). Firstly, our reconstruction attempts to capture all past ice flow events in the landform record and not simply the fast ice flow signature. Secondly, our reconstruction is at a higher resolution, which can provide greater detail for reconstructing past ice stream activity.

The flowsets of Kleman et al. (2010) match 37% of the flowsets in our reconstruction. Only 26% of the flowlines mapped by Shaw et al. (2010) match our flowsets, and these flowlines are not categorized into flowsets. The reconstruction of Brown (2012) only extends to 115° W, while our reconstruction extends to 110° W, but in the shared map area, Brown (2012) identified ~45% of the flowsets in our reconstruction. In general, this is the result of the high-resolution ArcticDEM allowing the identification of small flowsets that were beyond the resolution of these previous studies and for categorizing previously mapped flowsets with much higher confidence.

Our flowset map broadly matches the existing ice flow reconstructions that have been undertaken at a more local scale, but often contains less detail (Bednarski, 2008; Evans et al., 2021; Normandeau and McMartin, 2013; Stokes et al., 2006, 2009). In the region surrounding the Amundsen Gulf Ice Stream (Fig. 1), which has figured in previous ice flow reconstructions (e.g. McMartin and St-Onge, 1990; Stokes et al., 2006, 2009), our flowset map captures previously described flow events but the flowsets often have a different geometry, as they have been drawn based on different datasets. Compared to the reconstruction of McMartin and St-Onge (1990), we capture the same sequence of earlier northwesterly flow which became oriented to the west, with ice sheet thinning resulting in ice lobes splitting into the valleys and a final southerly ice flow phase from Victoria Island. In this region, our flowset map is also limited by the lack of offshore data in our reconstruction, so does not attempt to reconstruct or copy these flow events across from previous studies. In the Smoking Hills region (Figure 1), the flowset map of Evans et al. (2021) exceeds the detail presented in our map and hence a few small (<400km$^2$) flowsets are not depicted in our reconstruction. Just east of the Liard Plateau (Fig. 1), Bednarski (2008) reconstructed the ice flow patterns and glacial lake history during deglaciation. While this reconstruction does not present a formalized flowset map, the ice flow patterns presented match well with our flowset map. A comparison with the existing surficial geological mapping (Figure 1b) reveals that our reconstruction broadly captures the major flow events that occurred during deglaciation. However, it also highlights the main deficiency of our work in identifying any flow events that occurred prior to the last deglaciation. This is illustrated by a clear north/northwesterly flow event recorded by striations during the early stages of the glaciation (Normandeau and McMartin, 2013) that is absent from our landform-driven reconstruction. As such, the reconstruction we present is heavily focused on the ice flow changes during the last deglaciation.

## 4.2 Ice margin retreat record

Terrestrial ice-contact landforms have been used to mark deglacial ice margin positions and associated ice flow directions across the study area (Figure 5). In the northwest, hummocky (controlled) moraine belts are the dominant ice-contact deglacial landform (Evans et al., 2021), but former ice margin positions in this region are also delineated by end moraines and lateral/submarginal meltwater channels. On the Canadian Shield to the east, these landforms become much less common and the deglacial retreat pattern is recorded by eskers. Here the ice margin is extrapolated between eskers based on the assumption that eskers form time-transgressively perpendicular and proximal to the ice margin (Shreve, 1985; Storrar et al. 2014a; Livingstone et al., 2015).

Our record of deglacial ice margin positions shows a large variation in ice flow direction throughout deglaciation (Figure 5). In the north of the study area, two opposing ice retreat directions are mapped: (1) ice retreat to the west, marked by the Husky Lake Moraines; and (2) ice retreat towards Victoria Island in the east (Figure 5). On the Horton Plain and in the northern Mackenzie Valley, the ice-contact landforms were deposited as the ice margin retreated to the south and southeast respectively

420  (Figure 5). Over the eastern portion of the study area, including the Great Slave Plain and the Canadian Shield, the former ice margins record a general ice retreat pattern towards the east, with localised southerly ice flow around topographic features (Figure 5).

In the north, where opposing ice retreat patterns are mapped, the ice margin retreat pattern is complex due to the separation of different major ice lobes, with the landform record documenting several interlobate ice margins. The most notable example is

425  located in the Coronation Hills and is delineated by a prominent moraine and esker system. Deglacial interlobate ice configurations have also been previously identified during deglaciation of the LIS in the Smoking Hills region of our study area (Evans et al., 2021). We provide a simplified reconstruction of the ice margin retreat pattern in this region.

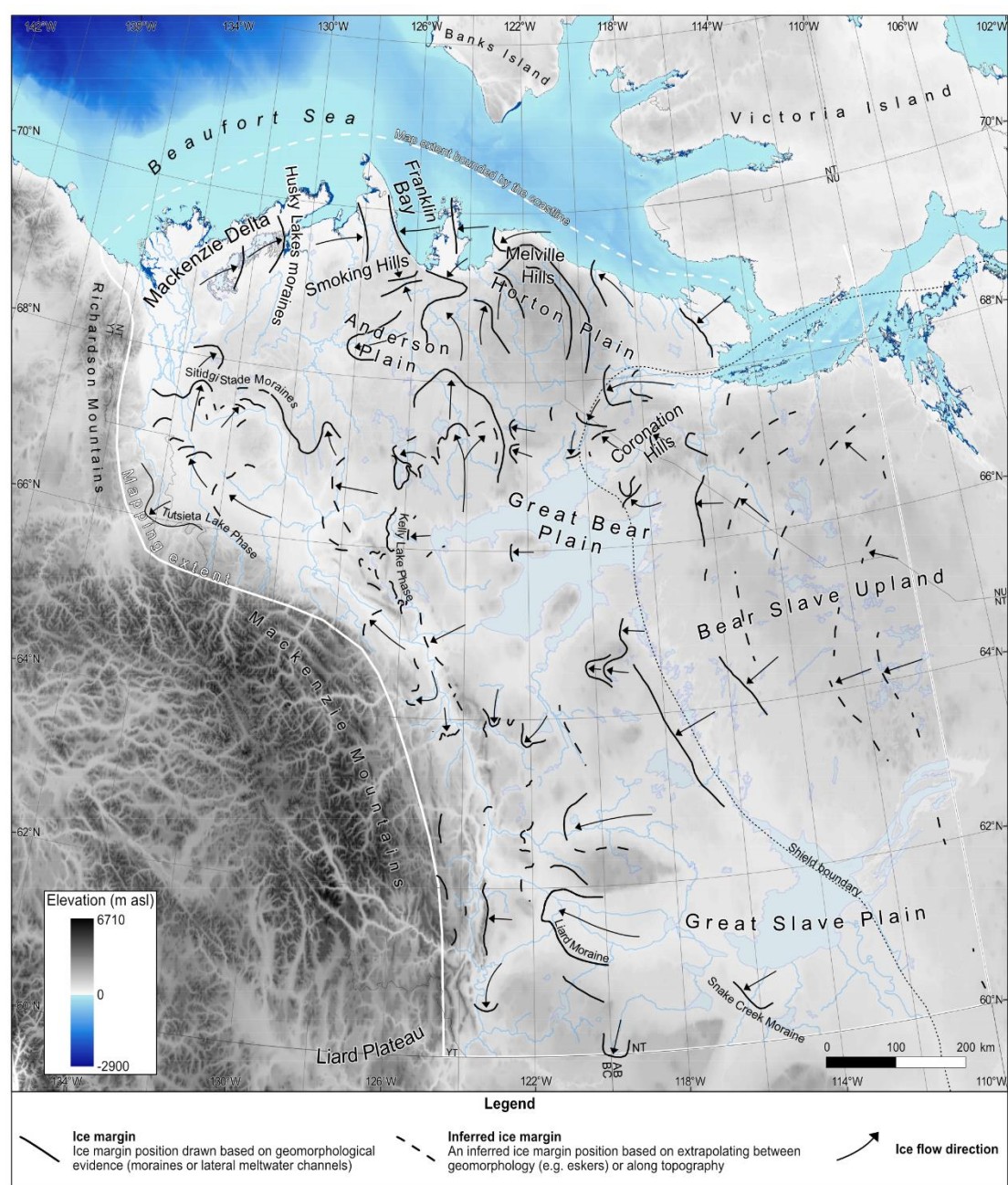

**Figure 5: The ice margin retreat pattern of the northwestern Laurentide Ice Sheet based on the deglacial landforms** from Dulfer et al. (2023). We label the landforms that have previously been identified and used to establish the deglacial retreat pattern in the literature. This includes the Liard Moraine (Smith, 1994), Snake Creek Moraine (Lemmen et al., 1994), landforms of the Kelly Lake Phase and Tutsieta Lake Phase (Hughes, 1987), Sitidgi Stade moraines (Rampton, 1988), and Husky Lakes thrust moraines (Dyke and Evans, 2003; Evans et al., 2021). The ice margin retreat pattern in

**the Smoking Hills area is simplified due to the scale of the map, for a more detailed ice margin reconstruction in this area see Evans et al. (2021).**

## 5 Interpretation

### 5.1 Detailed reconstruction of ice flow

In the following section, we describe the detailed changes in ice flow direction and dynamics at 500-year timesteps, constrained by the ice margin chronology of Dalton et al. (2023) (Figure 6). At various timesteps, we use our reconstructed ice margins (Figure 5) to provide further detail on the ice margin retreat pattern. The relatively coarse timestep of the ice margin chronology compared to our ice flow reconstruction means that conflicting flow directions may occur within a single timestep. In these situations, we define the distinct flow events as flow phases (whereby flow phase 1 represents an earlier flow event, which has been overprinted by flow phase 2) and in Figure 6 we use different ice flow line styles to depict these separate flow phases.

### 5.1.1 Pre-LGM flowsets

The signature of pre-LGM ice flow is sparse across the study area because the surficial geomorphological record typically only records the most recent flow event (for more detail see section 3.4; Dalton et al., 2023). Despite this, we assign multiple flowsets to the pre-LGM period (Figure 6A). Fs-240 (Figure 6A, mid-figure) is one of the oldest flowsets we identify and it represents early ice flow towards the west, as the LIS advanced over this region and before the initiation of the Amundsen Gulf Ice Stream. This pre-LGM westwards advance is well-established (Kleman and Glasser, 2007; Margold et al., 2018). The Amundsen Gulf Ice Stream operated as a major drainage outlet of the northwestern LIS throughout the Quaternary and likely operated prior to the local LGM in this region (Batchelor et al., 2014). We interpret Fs-221, 223, and 226 (Figure 6A, top-right) as forming in the onset zone of an early Amundsen Gulf Ice Stream. In contrast, the Mackenzie Trough has a relatively short history of glaciation, with the offshore stratigraphic record suggesting only two Quaternary ice advances (Batchelor et al., 2013). The most recent advance occurred during the last glaciation and likely contributed to the advance of the LIS to its maximum extent at the continental shelf break (Bateman and Murton, 2006; Batchelor et al., 2013). Flowsets 89, 98, 100, 103, and 104 all indicate fast ice flow over undulating terrain with a deflection to the west (Figure 6a, mid-left). This deflection aligns the lineations within these flowsets with ice flow down the Mackenzie Valley. The lack of influence of the subglacial topography on these flowsets suggest that they did not form in an ice marginal location but were formed some distance up-ice (Siegert et al., 2004; Winsborrow et al., 2010; Hughes et al., 2014). Therefore, we interpret these lineations as forming in an up-ice area of the Mackenzie Trough Ice Stream. The exact timing of operation of these flowsets is difficult to identify, but they are cross-cut by multiple younger flowsets, suggesting they formed early during glaciation. Finally, we suggest that the Cameron Hills Fragment (Fs-278, bottom-centre) represents an early ice stream that operated as the LIS and CIS coalesced prior to the local LGM, following the interpretation of Hagedorn (2022). The Cameron Hills Fragment is overprinted by

multiple younger flowsets and located at a high elevation, which supports the interpretation of formation during an early flow

event (Hagedorn, 2022), as opposed to formation during deglaciation as proposed by Margold et al. (2018).

### 5.1.2 Local LGM (17.5 to 17.0 ka)

Based on a combination of luminescence and radiocarbon constraints, the local LGM of the northwestern LIS was short-lived, reaching its maximum extent at 17.5 ka and subsequently retreating by 17.0 ka (Figure S3; Bateman and Murton, 2006; Kennedy et al., 2010; Dalton et al., 2023). During this period, the Amundsen Gulf Ice Stream operated in the eastern Beaufort

Sea, fed by ice flow from the Keewatin Ice Dome (Fs-221, 223, and 226, top-right) and from Victoria Island (Stokes et al., 2006 and 2009; Batchelor et al., 2014). As the Amundsen Gulf Ice Stream advanced to the west, it deflected ice flow across the Anderson Plain and Mackenzie Delta region to the west, as indicated by Fs-207 (Figure 6B, top-left). The Mackenzie Trough Ice Stream was also active at this time and deflected to the west. This westward deflection is necessary to form the thrust mass of Hershel Island (Dyke and Evans, 2003; Wetterich et al., 2023). Widespread fluvial reworking, periglacial

processes and deglacial flow signatures in the Mackenzie Delta region mean that there is a limited signal of the Mackenzie Trough Ice Stream, with the clearest signature being observed further up-ice (Fs 89, 98, 100, 103, and 104, mid-left). It is likely that some of the unclassified flowsets (e.g. Fs 189, 190, 194, 204; Figure 6A, top-left) in this area were formed during the operation of the Mackenzie Trough Ice Stream but have been modified following deglaciation. The ice flow pattern at this time displays a clear orientation to the north and northwest, indicative of two main ice source areas: the Cordilleran-Laurentide

ice saddle to the south and the Keewatin Ice Dome to the east.

### 5.1.3 Early deglaciation (16.5 to 16.0 ka)

Between 16.5 ka and 16.0 ka, the Amundsen Gulf Ice Stream remained as the main outlet draining ice from the Keewatin Dome to the eastern Beaufort Sea (Figure 6D and E). During this time, there was a reorganisation of the ice stream network on the western Northern Interior Plains, as neighbouring ice streams switched off and on. On the Anderson Plain, highly

attenuated and continuous lineations (Fs-206; Figure 6D and E, top-left) record fast ice flow as the Anderson Ice Stream switched on and became the main drainage outlet from the Cordilleran-Laurentide ice saddle region to the western Beaufort Sea.

The terminus of the Anderson Ice Stream (Fs-206; Figure 6D and E, top-left) is deflected to the east with no clear ice flow to the west. This deflection has two important implications for the ice sheet configuration at this time: (1) the Amundsen Gulf

Ice Stream had retreated far enough to the east that it no longer provided a buttressing effect on ice flow across the Anderson Plain; and (2) that ice was present in the adjacent Mackenzie Valley, restricting ice flow in this direction. The relatively late activation of the Anderson Ice Stream following the retreat of the Amundsen Gulf Ice Stream is supported by an offshore seismic sedimentary sequence presented by Batchelor et al. (2014), who show that the youngest till within the Amundsen Gulf Trough was deposited by the Anderson Ice Stream.

Deglacial and event flowsets (Fs 195, 199, 200, 201, 202, 203; Figure 6E, top-left) provide evidence for a slower flow regime in the Mackenzie Valley at this time and the shut-down of the Mackenzie Trough Ice Stream. The ice flow patterns and deglacial landforms in the Mackenzie Valley between 68°N and 69°N record the slow retreat of an ice lobe up-valley. The broad controlled moraine belts that were deposited during the Sitidgi Stade (Rampton, 1988) and a series of recessional, push moraines record readvances of a debris-rich polythermal snout (Evans et al., 2021). Based on the change in ice flow regime in

the Mackenzie Valley, we suggest that the Mackenzie Trough Ice Stream and the Anderson Ice Stream were not active at the same time, but alternated in activity during deglaciation. Although the causes of the switch in ice stream routing are uncertain but may relate to the proximity of Amundsen Gulf ice and the buttressing it provides.

### 5.1.4 Slow deglaciation prior to the Bølling-Allerød (15.5 to 15.0 ka)

Deglaciation was relatively slow immediately prior to the start of the Bølling-Allerød interstadial (14.7-12.9 ka), with changes

in the ice drainage network resulting from the reorganisation of the active ice streams between 15.5 and 15.0 ka (Figure 6F and G). Two distinct flow phases of the Amundsen Gulf Ice Stream can be identified from the cross-cutting flowset patterns in the Coronation Hills region. The early flow phase (Fs 221, 223, 226; Figure 6B-E, top-right), depicted in previous timesteps, is broadly oriented towards the northwest and sustained mostly by ice flow from the Keewatin Ice Dome in the east. The geomorphic record of this early flow phase is fragmented and overprinted by later flow events, including Fs 222 and 239

(Figure 6F and G, top-right), which also demonstrate a northerly ice flow components and indicate the growing influence of the ice saddle which fed the Amundsen Gulf Ice Stream alongside the Keewatin Ice Dome. This strong northwesterly ice flow signal (Fs 222) has previously been recognised and termed the Inman River drumlin field by St-Onge and McMartin (1987). The exact timing of these flow phases is difficult to determine but is constrained by two events. The later flow phase must have occurred at or after the local LGM, as it is overprinted on flowsets which formed during the local LGM or pre-LGM

period (Fs 221, 223, 226). The clear northwards orientation of this flowset indicates ice flow from the saddle region, so the early flow phase must have occurred before the collapse of the ice saddle, which occurred during the Bølling-Allerød interstadial (Stoker et al., 2022). The well-preserved nature of Fs 222 and 239 suggests that they relate to the final phase of ice flow from mainland Canada, over the Coronation Hills, to the Amundsen Gulf Ice Stream. Following the end of this flow phase, any fast ice flow in the Amundsen Gulf must have been sustained by ice principally sourced from the Keewatin Ice

Dome over southwestern Victoria Island and not the ice saddle. Therefore, we suggest that this flow phase of the Amundsen Gulf Ice Stream likely occurred between 15.5 and 15.0 ka while the ice saddle remained thick enough to influence ice flow dynamics (Figure 6F).

The retreat of the LIS margin in the Northern Interior Plains led to spatial changes in the ice flow regime that occurred simultaneously across the region as ice streams reactivated. As the ice margin retreated up the Mackenzie Valley, the slower

ice flow regime (Fs 191, 192, 195, 196, 197, 198, 199; Figure 6F and G, top-left) changed to a faster ice flow regime (Fs 154, 156, 164, 173, 188; Figure 6F and G, top-left) due to the reactivation of the Mackenzie Trough Ice Stream around 67°N. The lobate shape of the flow patterns (e.g. Fs 154, 173, 188) suggests that they were formed at the margin of an ice lobe retreating

up the Mackenzie Valley during a period of active retreat. The more parallel, linear-pattern flowsets (Fs 101, 102, 156) were formed further up-ice, away from the margin, and indicate a switch to more northwesterly oriented ice flow in the up-ice

portions of the ice stream, compared to the more northerly oriented flow of the earlier flow phase during the local LGM (e.g. Fs 90, 101, 102; Figure 6F and G, mid-left). As ice retreated across the Anderson Plain, the northwestern section of the Bear Lake Ice Stream activated (Fs 115; Figure 6F and G, mid-left), and the morphology of the ice flow parallel lineations within this ice stream indicates that it merged with the Mackenzie Trough Ice Stream to the south and the Anderson Ice Stream to the north.

As ice retreated up the Mackenzie Valley, the flow direction was heavily controlled by the local topography with many small ice lobes exhibiting varying flow directions due to the deflection around higher terrain (Fs 175, 179, 180). At the range front of the Canadian Cordillera, the ice flow was directed westwards into the foothills of the Mackenzie Mountains (Fs 3, 4, 5, 19, 20; Figure 6G, bottom-left) and the Richardson Mountains (Fs 161, 162, 167, 171, 172; Figure 6G, top-left).

**5.1.5 Early Bølling-Allerød (14.5 ka)**

High temperatures during the early Bølling-Allerød interval may have triggered the collapse of the ice saddle between the CIS and LIS and rapid ice margin retreat, as suggested by numerical modelling simulations(Gregoire et al., 2016) and cosmogenic nuclide exposure ages collected over vertical elevation transects (Stoker et al., 2022). The ice drainage network underwent widespread reorganisation in response to this collapse, with a shift to westerly-oriented ice flow (Figure 6H). During the earlier

stages of this timestep, ice flow was still fed by the ice saddle region and directed along the Mackenzie Valley to the north (Fs 23, 34, 39, 309; Figure 6H, bottom-left). In the later stages of this timestep, the ice saddle thinned rapidly and the Keewatin Ice Dome became a more dominant ice source, leading to more westerly-oriented ice flow (e.g. Fs 51, 52, 53, 54; Figure 6H, bottom-left). These changes are also observed in the ice stream network draining through the Amundsen Gulf. The activation of the northwesterly-oriented Paulatuk Ice Stream (Fs-138; Figure 6H, top-left) indicates the increasing importance of the

Keewatin Ice Dome as a source area, compared to the Amundsen Gulf Ice Stream at the 15 ka timestep, which maintained clear ice source contribution from the ice saddle as indicated by the northerly-oriented ice flow (Fs 222, 239; Figure 6G, top-right). In the 14.5 ka timestep, we do not depict fast ice flow within the Amundsen Gulf, as the geomorphological evidence for this is likely offshore. This is despite the fact that the Amundsen Gulf Ice Stream may have persisted through this period and was sustained by ice flow from Victoria Island and possibly the Paulatuk Ice Stream (Stokes et al., 2006, 2009; Batchelor

et al., 2014; Lakeman et al., 2018). The thinning of ice lobes to the east of the Melville Hills led to the reduction in ice flow activity responsible for forming the northerly Inman River drumlin field (Fs 222) and caused ice flow to switch to a more westerly orientation (St-Onge and McMartin, 1987). The transition to westerly oriented ice flow and ice thrusting is associated with the formation of the Bluenose Lake moraine complex (St-Onge and McMartin, 1987, 1999).

The ice stream network across the Northern Interior Plains displayed a similar shift to north-westwards oriented ice flow as

the Bear Lake Ice Stream and the Fort Simpson Ice Stream became active. We draw an extensive Bear Lake Ice Stream during

this time, but the question remains as to whether the ice stream was active and forming lineations over its entire length at any single time, or the fast ice flow was concentrated near the ice sheet margin and formed the pattern of lineations time-transgressively at the margin as it retreated, or a combination of both. Despite the absence of cross-cutting relationships, we suggest that margin retreat was active with a stepwise pattern of ice margin retreat and lineations forming time-transgressively near to the ice sheet margin due to two main reasons. Firstly, the ice stream flowsets at this time are broadly aligned with ice margin retreat landforms (e.g. moraines, eskers). Secondly, the lineations in the southern portion of the Bear Lake Ice Stream (Fs-65; Figure 6H, mid-image) display a transition from straight and highly parallel MSGLs at higher elevations to a series of sinuous, sometimes paired ridges with cross-cutting relationships at lower elevations near the Keith Arm of Great Bear Lake (Figure S4). We interpret the lineations at higher elevations as forming beneath grounded ice while the lower elevation lineations formed later by groove-ploughing from ice keels beneath the partially floating Bear Lake Ice Stream while the ice margin was lightly grounded, following ice margin retreat and the development of Glacial Lake McConnell (Figure 7; Clark et al., 2003; Piasecka et al., 2018; Dowdeswell and Ottesen, 2022).

Cross-cutting relationships show that the final ice flow across the Mackenzie Valley was directed to the west as the Bear Lake Ice Stream retreated. Fast ice flow to the northwest in the Mackenzie Valley around 65°N (Fs 83, 312; Figure 6F and G, mid-left) is overprinted by the large westerly-flowing Bear Lake Ice Stream (Fs-115; Figure 6H, mid-left) and deglacial flowsets (Fs 94, 95, 96, 97, 157, 162; Figure 6H, mid-left), which display divergent flow patterns indicating the retreat of an ice lobe to the east. This shift from northwest ice flow to a final westward flow phase is representative of the change in flow direction observed across the entire region. This eastwards retreat pattern is supported by deglacial flowsets (Fs 106, 107, 108, 109, 110, 148; Figure 6H, mid-left), which are aligned with lateral meltwater channels and moraines around the Norman Range.

### 5.1.6 Rapid thinning during the Bølling-Allerød (14.0 ka)

Following a period of rapid ice sheet thinning during the saddle collapse, the LIS and CIS had separated along the eastern front of the Canadian Cordillera by 14.0 ka, based on cosmogenic nuclide exposure ages and radiocarbon constraints (Gregoire et al., 2016; Stoker et al., 2022; Dalton et al., 2023). The loss of the ice saddle meant that the Keewatin Ice Dome was the sole ice source for the northwestern sector of the LIS during the subsequent retreat. Thus, regional ice flow patterns were dominated by broad-scale ice flow towards the west (Figure 6I). Any variation from this westerly-oriented ice flow was typically caused by the localised deflection of ice flow around higher topography or down the Mackenzie Valley, including a 180° flow reversal at around 64°N (Fs-26; Figure 6I, centre-left). During this time, the ice sheet margin continued to rapidly retreat across the Northern Interior Plains (Reyes et al., 2022; Stoker et al., 2022; Dalton et al., 2023) causing rapid adjustments in the ice drainage network. As a result, we depict multiple flow phases within this single timestep.

At 14 ka, the ice stream network again underwent rapid reorganisations driven by switch-on and shut-down phases for different ice streams. During the early stages of this timestep, the Fort Simpson Ice Stream was active, with a broad convergence zone (Fs-10; Figure 6I, bottom-left) draining ice from the Keewatin Ice Dome. As the westerly ice flow of the Fort Simpson Ice

Stream approached the Franklin Mountains and Mackenzie Mountains Foothills, it was deflected to the north by the topographic relief and formed a series of time-transgressive cross-cutting flowsets during deglaciation (Fs 22, 38, 39, 40, 49, 61, 62; Figure 6I, mid-left). The onset zone of the Fort Simpson Ice Stream (Fs-10) is overprinted by the Great Slave Ice Stream (Fs-284; Figure 6I, bottom-left), suggesting that the Great Slave Ice Stream became active following the shut-down of the Fort Simpson Ice Stream in the later stages of this timestep. This likely occurred at the same time ice was flowing over the uplands of Sambaa K'e (formerly known as Trout Lake; Fs 292, 295, 296; Figure 6I, bottom-left). This sequence of ice stream activity differs from that of Margold et al (2018), who suggested that the Great Slave Ice Stream operated before the Fort Simpson Ice Stream. This was largely due to the constraints of the ice margin chronology of Dyke et al. (2003; updated by Dalton et al., 2023).

At this time, the ice stream activity in the Amundsen Gulf region was reduced as the Paulatuk Ice Stream switched off, and we find no geomorphic evidence of fast ice flow related to the Amundsen Gulf Ice Stream on mainland Canada. Instead, the Amundsen Gulf Ice Stream must have been sustained by ice flow from Victoria Island or had switched-off. Southerly ice flow, around the Horton Plain (Fs-218), indicates the dominance of ice flow from Victoria Island to the north at this time. During retreat, this ice flow became more westerly-oriented and controlled by the topography (Fs-219 and Fs-20 in the 13.5 ka timestep). The Haldane Ice Stream (Fs-129) operated early during this timestep, flowing through the Colville Hills. Following the switch-off of the Haldane Ice Stream, a second flow phase occurred. Overprinting deglacial landforms indicate that ice flow was directed locally to the north around the Colville Hills (Fs-145; Figure 6I, top-left) during the final phase of ice retreat, with geomorphological evidence providing some indication that a readvance may have occurred before final retreat at this location.

### 5.1.7 Ice sheet separation (13.5 ka)

Cosmogenic nuclide exposure ages indicate that the northwestern LIS continued to retreat rapidly towards the Canadian Shield across the Northern Interior Plains following ice sheet separation (Reyes et al., 2022; Stoker et al., 2022). As the LIS retreated out of the topographically complex areas in the foothills of the Mackenzie Mountains, ice flow patterns were less complex and dominated by a more consistent westerly flow (Figure 6j). During early stages of this timestep, the only active ice stream was the Great Slave Ice Stream (Fs-284; Figure 6J, bottom-left) because ice flow in the Great Slave Basin was directed towards the west (flow phase 1). As the ice margin retreated, the Great Slave Ice Stream switched-off and ice flow was directed more towards the southwest (flow phase 2; Fs 275, 276, 277; Figure 6J, bottom-left), including Fs-328 which is overprinted on flowsets derived directly from the Great Slave Ice Stream.

In the Horton Plain and Great Bear Plain region, to the north, the ice flow history is more complex due to the influence of the topographic relief and separation of ice sourced from Victoria Island and mainland Canada. North of the Horton Plain, ice flow (Fs-218; Figure 6, top-left) to the southwest indicates the dominance of ice flow from Victoria Island during the final stages of deglaciation. This ice flow gradually became more topographically confined and oriented towards the west (Fs-20;

top-left), along the Amundsen Gulf. These flowsets overlap with the margin reconstruction of Dalton et al. (2023) and suggest a more lobate retreat in the Amundsen Gulf at this time aligned with a series of lateral meltwater channels. To the east of the Melville Hills, in the Coronation Hills, continued ice sheet thinning and the large variations in topographic relief led to multiple cross-cutting flow patterns from different ice lobes, as indicated by deglacial landforms. These landforms include an esker with flat-topped sections, which displays a morphology similar to examples from Finland where deposition was in an interlobate position (Mäkinen, 2003). Hence, we suggest that this esker formed at the suture zone of two ice masses which were separating over the Coronation Hills. Detailed field investigations would be required to confirm this interpretation. The separation of the ice sheet margin into two ice lobes led to a switch from broadly westerly-oriented ice flow during the early stages of deglaciation, which was unaffected by topography, to a more topographically-confined flow during the later stages of glaciation. During this time, highly elongate drumlins in the Richardson Valley (Fs 237) likely record the surging of an ice lobe which was also responsible for the formation of a series of moraines along the northern slopes of the valley (McMartin and St-Onge, 1990) that was later termed the Kugluktuk Ice Stream (Margold et al., 2015). Our ice flow sequence in this region matches that of Kleman and Glasser (2007). It is also consistent with the early work of McMartin and St-Onge (1990) who recognised the importance of ice sheet thinning in dictating the ice flow patterns in this region as the ice lobes became separated into major basins, which they termed the Harding River phase of deglaciation.

### 5.1.8 End of the Bølling-Allerød (13.0 ka)

As the LIS retreated onto the Canadian Shield, cosmogenic nuclide dating indicates that the ice margin began to stabilise and the retreat rates slowed (Reyes et al., 2022; Stoker et al., 2022). Across the Canadian Shield, the ice flow was dominated by large-scale radial sheet flow patterns (e.g. Fs 249 and 252; Figure 6K, mid-right) rather than the more complex patterns of rapidly evolving ice streams across the Northern Interior Plains. These flow patterns are broadly independent of the small, local variations in topography. In the northeastern Horton Plain, ice flow is directed parallel to the Amundsen Gulf (Fs 224, 225, 231, 323; Figure 6K, top of image), but the deglacial flowsets suggest a slower ice flow regime. As such, there is no evidence of fast ice flow in the study region at this time. This does not preclude the persistence of the Amundsen Gulf Ice Stream, which might be recorded by evidence of fast ice in the offshore record; slower onshore ice flow directed down the Amundsen Gulf (Fs 224, 225, 231, 323; top of image) may also have contributed to the remnant Amundsen Gulf Ice Stream. Indeed, Margold et al. (2018) inferred that the Amundsen Gulf Ice Stream remained active, based on the assumption that a marine outlet would be conducive to fast ice flow, until 11.5 cal. ka BP but with a reduced extent.

### 5.1.9 Early Younger Dryas (12.5 ka)

The ice flow pattern remained largely unchanged at this time. Across the Canadian Shield, ice flow was dominated by the radial sheet flow from the Keewatin Ice Dome (Figure 6L). The main change in ice flow direction occurred in the Horton Plain

region, where deglacial flowsets associated with moraines indicate that the final ice retreat across the area was towards Victoria Island in the northeast (Fs-228, top of image), as previously recognised by McMartin and St-Onge (1990) and termed the Stapylton Bay phase. This either constitutes a minor readvance or simply the last stillstand phase during deglaciation of the region (Stokes et al., 2006).

### 5.1.10 Deglaciation across the Canadian Shield (12.0 to 10.5 ka)

During the final stages of the slow deglaciation of the study area, ice flow continued to be dominated by broad, radial sheet-flow patterns with small variations of locally deflected ice flow largely controlled by topography (Figure 6M-P). This includes Fs-251 (Figure 6O, centre), which is a smaller radial flowset overprinted on the large Fs-249 (Figure 6K, centre). In the final stage of deglaciation, Fs-250 was overprinted on Fs-251 by a small ice lobe flowing off the Canadian Shield (Figure 6P, top of image). Similarly, Fs-254 is overprinted on Fs-253 and formed following ice margin retreat (Figure 6P, bottom).

### 5.1.11 Unassigned flowsets

The absence of any cross-cutting patterns or relationships to other geomorphological landforms means that we could not assign the operation of certain flowsets to a specific timestep. There were over forty flowsets which remained unassigned to any timestep and these are predominantly 'unclassified' flowsets and small in spatial extent (Figure 6A). In particular, Fs-230 is somewhat perplexing. It is located in the eastern Horton Plain and displays an ice flow direction towards the northeast, which is at almost 180° to the surrounding flowsets (Fs 229, 231, 232, 233). We cannot reconcile the timing of this flowset with the surrounding features or known ice source areas and suggest it may predate the local LGM (Kleman et al., 2010). Remnant ice caps and plateau icefields were left behind following the recession of the southwestern LIS (Norris et al., 2023). While we do not observe any evidence for the presence of localised ice masses following the retreat of the northwestern LIS, we acknowledge the possibility of their presence. Future observations may recognise the presence of local ice masses on the upland of the former bed of the northwestern LIS and might provide an explanation for some of our unassigned flowsets.

### 5.2 Summary of ice flow evolution through time

Prior to the local LGM, the northwestern LIS advanced westwards towards the Canadian Cordillera (Kleman and Glasser, 2007; Margold et al., 2018) (Figure 6a). The Amundsen Gulf Ice Stream was likely active during ice advance to the local LGM extent and flowed into the eastern Beaufort Sea (Figure 6b; Stokes et al., 2006; Batchelor et al., 2014). However, there is limited landform evidence of other ice streams being active at this time, with the Cameron Hills Fragment recording the only other clear pre-LGM ice stream. The lack of evidence of ice streaming is likely not an indicator of an absence of ice streams during the advance phase.Instead, the evidence of ice streams during the ice advance phase has been overprinted by the deglacial signature or postglacial processes and is no longer visible in the surface landform record. The prevalence of deglacial flowsets indicates this overprinting activity. The local LGM of the northwestern LIS occurred relatively late compared to the

rest of the ice sheet (~17.5 ka, Kennedy et al., 2010; Murton et al., 2015; Stoker et al., 2022; Dalton et al., 2023), following

the coalescence of the LIS with the CIS and the formation of an ice saddle between the ice sheets. At this time, ice flow was

directed predominantly to the north or northwest across the study region. The ice flow network was drained by two major,

marine-terminating ice streams, the Amundsen Gulf Ice Stream and the Mackenzie Trough Ice Stream, the former fed with ice

from the Keewatin Ice Dome and the latter from the CIS-LIS ice saddle (Figure 6b; Batchelor et al., 2013, 2014; Margold et

al., 2018).

Deglaciation of the northwestern LIS was characterized by periods of rapid retreat and thinning interspersed with periods of

slow retreat as seen from the presence of periodic recessional moraines and some larger moraine crests (Reyes et al., 2022;

Stoker et al., 2022; Dalton et al., 2023). The ice flow and ice stream network evolved in response to variations in the rate of

deglaciation. The initial deglaciation of the northwestern LIS was characterized by slow ice margin retreat and a relatively

stable ice drainage network (Dalton et al., 2023). The Amundsen Gulf Ice Stream was the main drainage outlet of the LIS in

the eastern Beaufort Sea area and remained active throughout the early period of deglaciation, with minor reorganisations of

the ice stream source area (Figure 6B-F; Stokes et al., 2006, 2009). In the western Beaufort Sea, the main drainage outlet

alternated between the Mackenzie Trough Ice Stream and the Anderson Ice Stream. The Mackenzie Trough and Anderson ice

streams which were likely not active at the same time and were fed by the ice saddle between the CIS and LIS (Figure 6B-F).

The Bølling–Allerød interval (14.6 – 12.9 ka) was a millennial-scale warming event that occurred during the last deglaciation

that is associated with rapid retreat of Northern Hemispheric ice sheets and glaciers (Lambeck et al., 2014; Menounos et al.,

2017). Increasing temperatures during the Bølling–Allerød interval have been reconstructed to have triggered the collapse of

the ice saddle between the CIS and LIS, which was followed by a period of rapid ice sheet thinning and retreat (Gomez et al.,

2015; Gregoire et al., 2016; Stoker et al., 2022). Without the source from the ice saddle, the ice streams of this region began

to primarily receive input from the Keewatin Ice Dome to the east, as seen in the northerly flowsets transitioning to

northwesterly and eventually westerly-facing signatures of ice flow. (Figure 6G-J). The ice stream network also adjusted

rapidly to these changes. The loss of the ice saddle resulted in a limited contribution to the Amundsen Gulf Ice Stream directly

from the mainland of Canada, with ice for this ice stream coming from the Keewatin dome through Victoria Island (Figure

6G). The Paulatuk Ice Stream became active in the area immediately west of the former Amundsen Gulf Ice Stream and its

northwesterly-oriented flow direction reflects the increasing importance of the Keewatin Ice Dome (Figure 6H). In the northern

Mackenzie Valley region, ice stream activity became dominated by westerly flow, as demonstrated by the activation of the

Bear Lake Ice Stream (Figure 6G). Rapid retreat continued across the Northern Interior Plains following the separation of the

CIS and LIS, with this period characterized by westerly-oriented ice flow and relatively short-lived ice streams (Figure 6I and

J).

The Bølling–Allerød interstadial was followed by the Younger Dryas Stadial (12.9 – 11.8 ka), which was a cool period that

has been associated with glacier and ice sheet stabilization or advance (Lambeck et al., 2014), primarily in regions

surrounding the North Atlantic (Rea et al., 2020; Mangerud, 2021). The beginning of the Younger Dryas coincides with a

slowdown in the retreat rate of the northwestern LIS margin as it reached the Canadian Shield. This slow-down in ice retreat

was accompanied by a change in the ice flow dynamics (Figure 6K-P) from fast ice flow with numerous ice streams, which

evolved rapidly through the deglaciation on the Northern Interior Plains, to a slower ice flow regime and a radial ice flow

pattern fed by the Keewatin Ice Dome on the Canadian Shield. The exact control on this change in ice flow dynamics is

unclear but is likely related to the switch from the soft-bedded, lake-terminating margin on the Northern Interior Plains to the

hard-bedded, terrestrial margin on the Canadian Shield which is associated with less elongate streamlined subglacial

bedforms and inferred to relate to a decrease in ice flow speed and subglacial sedimentation organization (McKenzie et al.,

2022) (see section 6.3 for a more detailed discussion).

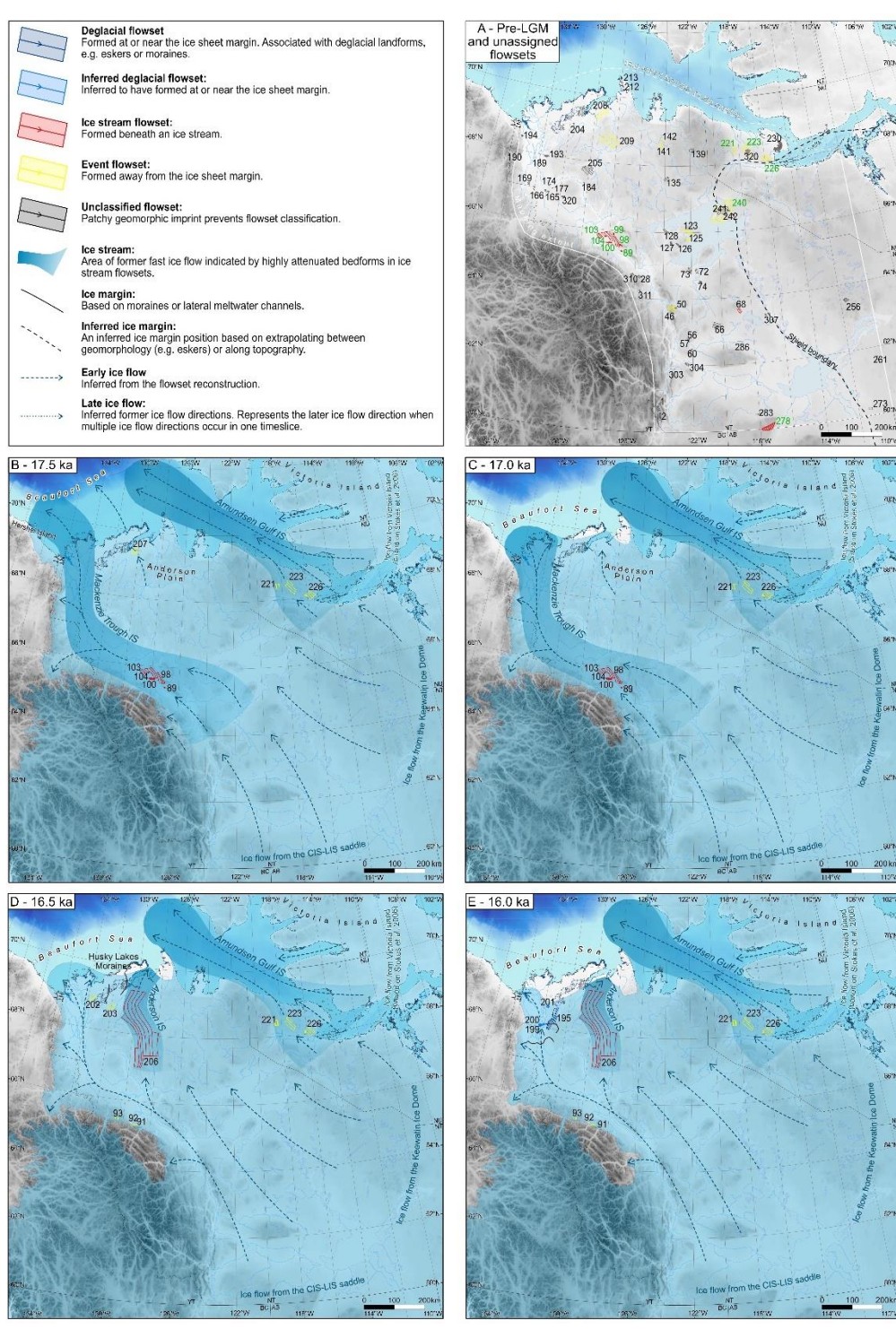

**Figure 6: The evolution of ice flow using the 500-year timeslices of NADI-1 (Dalton et al., 2023). In panel A, green numbering indicates pre-LGM flowsets and black numbering indicates unclassified flowsets. Due to the size of the figure, we only number key flowsets. An A4 version of each individual panel is provided in the Supplementary Figures document, including the age constraints on deglaciation (Figure S3).**


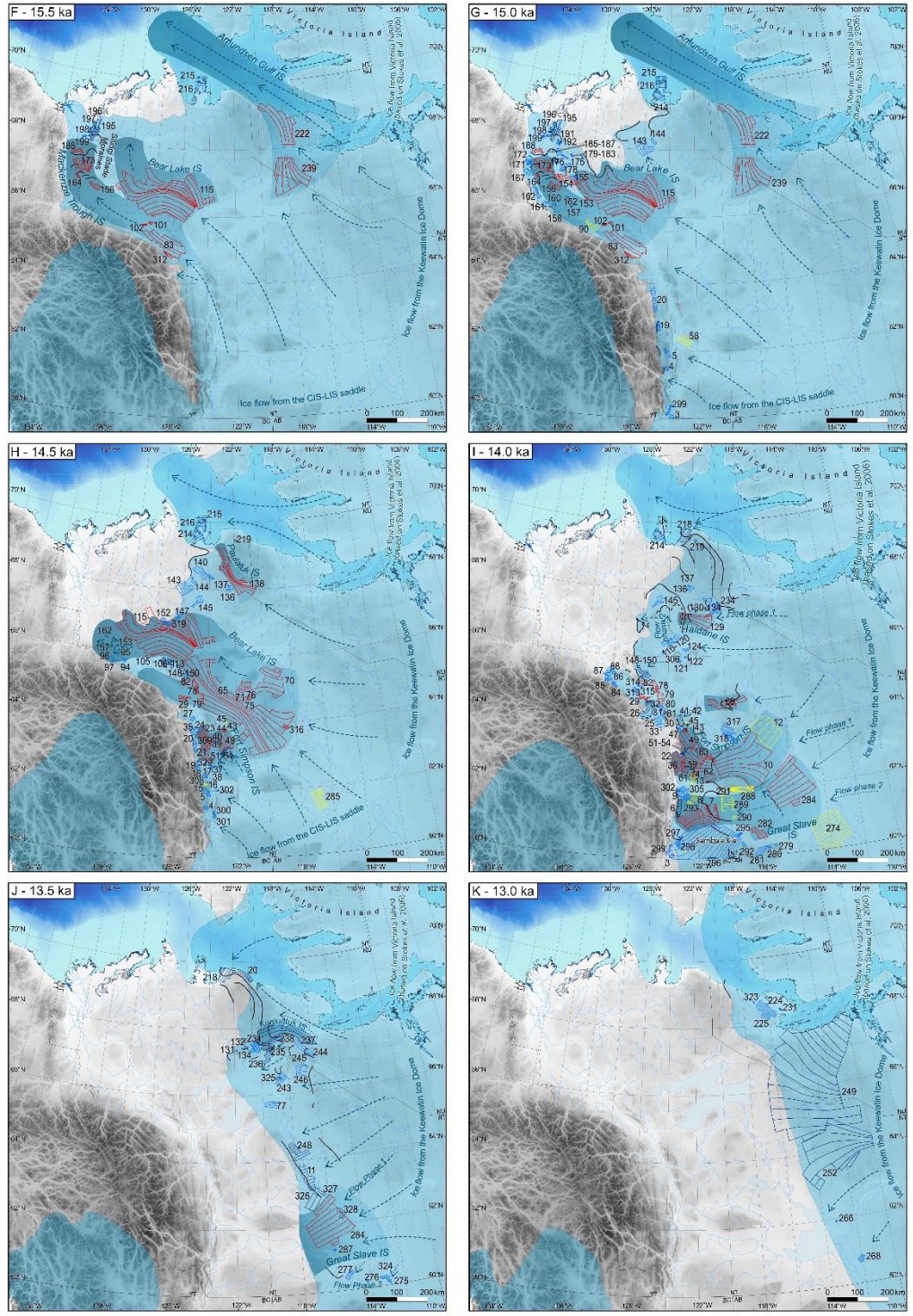

**Figure 6 cont:**

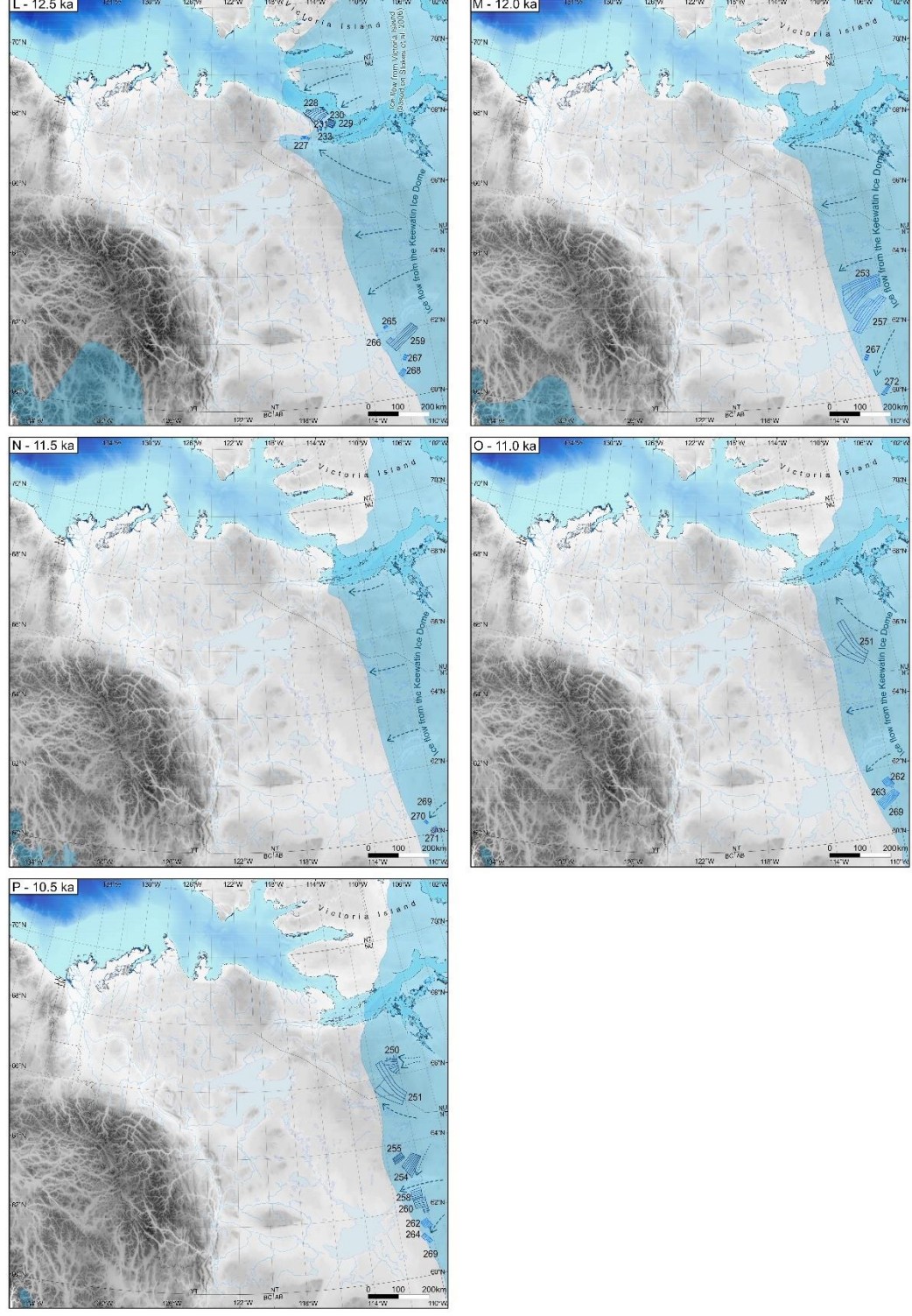

**Figure 6 cont:**

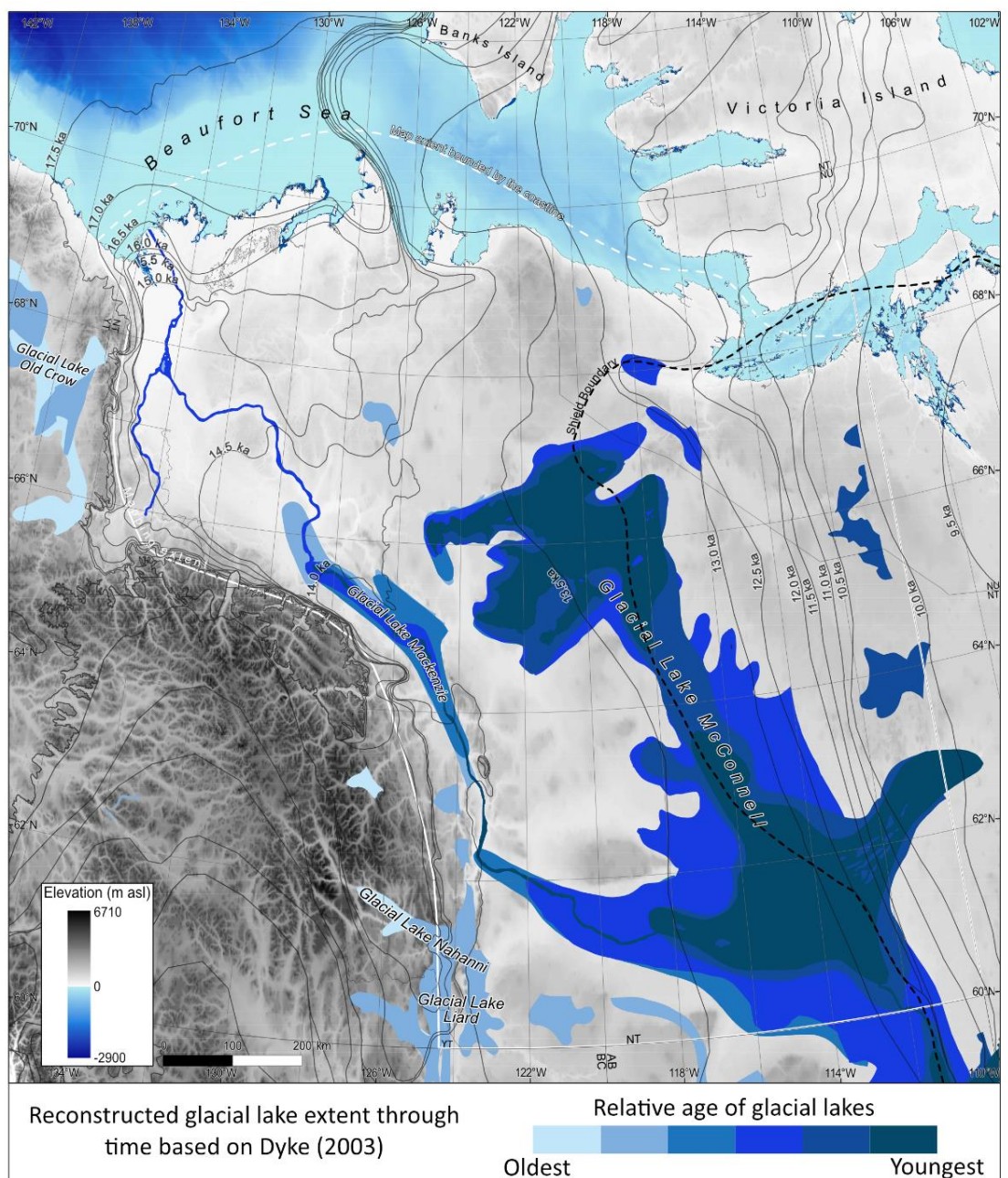

Figure 7: The relative age and extent of proglacial lakes across the study area during the last deglaciation, based on the reconstruction of Dyke et al. (2003). Black lines indicate the deglacial isochrones of Dalton et al. (2023). Note the mismatch between the isochrones and lake reconstruction as the glacial lake reconstruction has not been updated to match the new ice margin chronology.

## 6 Discussion

In our reconstruction, the northwestern LIS undergoes a rapid reconfiguration of its ice drainage network during deglaciation characterized by a change from northerly and northwesterly oriented ice flow at the local LGM to westerly-oriented flow during deglaciation. In this section, we discuss the implications of our reconstruction for the changing dominance of ice source regions and how this relates to saddle collapse, the style of ice stream activity, the nature of ice margin retreat processes, the controls on fast ice flow and the implications for ice sheet mass balance.

### 6.1 How did the separation of the ice sheets affect the ice drainage network?

The landform record displays a signal of more rapid thinning in the ice saddle region compared to the Keewatin Ice Dome, as the Keewatin Ice Dome became the dominant ice source region during deglaciation. Combined with the available chronological constraints, this provides support for the demise of the ice saddle and separation of the ice sheets occurring before the end of the Bølling–Allerød interval (Gregoire et al., 2016; Stoker et al., 2022). The dominance of northerly-oriented ice flow at the

local LGM and the start of deglaciation (Figure 6B-G) demonstrates a strong influence of the Cordilleran-Laurentide ice saddle on the ice drainage network. During the start of the Bølling–Allerød interval, the shift to more northwesterly-oriented ice flow occurs as surface mass balance / elevation feedback causes the rapid thinning of the ice saddle and weakens it relative to the Keewatin Dome (Figure 6H). The ice stream network responds to this change as the Amundsen Gulf Ice Stream (Figure 6G; Fs 222 and 239), with its strong northwards ice flow signal, terminates and is replaced by the Paulatuk Ice Stream (Figure 6H;

Fs-138) with a more northwesterly ice flow orientation. By the end of the Bølling–Allerød interval, ice flow was broadly westerly-oriented as the ice sheets had separated and the ice saddle had fully collapsed (Figure 6I).

An alternative scenario, where the rapid retreat of the Amundsen Gulf Ice Stream causes ice drawdown from the Keewatin Ice Dome and leads to the collapse of the ice saddle, has been proposed by Pico et al. (2019). This connection between Amundsen Gulf Ice Stream retreat and the ice saddle collapse is based on numerical ice sheet models forced by different

palaeotopographies. Based on radiocarbon ages, the collapse of the Amundsen Gulf Ice Stream took place after 13.0 ka and hence is used to suggest that the saddle collapse likely occurred after 13.0 ka (Lakeman et al., 2018; Pico et al., 2019). In fact, it is difficult to envisage a scenario where the Amundsen Gulf Ice Stream is not an important driver of rapid ice drawdown and margin retreat due to the large ice fluxes and high calving rates it likely delivered through the marine terminating margin. That said, it is unclear whether the collapse of the Amundsen Gulf Ice Stream preceded the loss of the CIS-LIS saddle and

whether this mechanism of drawdown is strong enough to weaken the entire ice saddle. Indeed, a recent re-evaluation of the radiocarbon constraints on the retreat of the Amundsen Gulf Ice Stream have suggested that marine mollusc incursion records the opening of the Bering Strait and not the rapid retreat of the Amundsen Gulf Ice Stream, with collapse likely occurring prior to 14.0 cal. ka BP (Vaughan et al., 2024).

Based on the empirical constraints, we favour the rapid thinning of the ice saddle region as the driver of rapid ice margin retreat

across the Northern Interior Plains. The landform record indicates that the final ice flow event in the Mackenzie Valley at ~65°N exhibits a clear southwards deflection around a topographic high of ~700m prominence, within a broad westerly ice

flow pattern from the Keewatin Ice Dome (Fs-25; Figure 6I and 9). Southerly ice flow up the Mackenzie Valley is overprinted on northerly ice flow from the ice saddle region, and therefore is only possible following the total collapse or significant weakening of the ice saddle, which would otherwise dominate the westward signal of the Keewatin Ice Dome. Both radiocarbon and cosmogenic nuclide exposure age constraints show that the Mackenzie Valley at ~65°N was deglaciated and occupied by Glacial Lake Mackenzie by the end of the Bølling–Allerød interval (Stoker et al., 2022; Dalton et al., 2023). This final ice flow event (Figure 8; Fs 24-26 and 31-32) constrains the separation of the ice sheets to before the Younger Dryas and is incompatible with the scenario proposed by Pico et al. (2019).

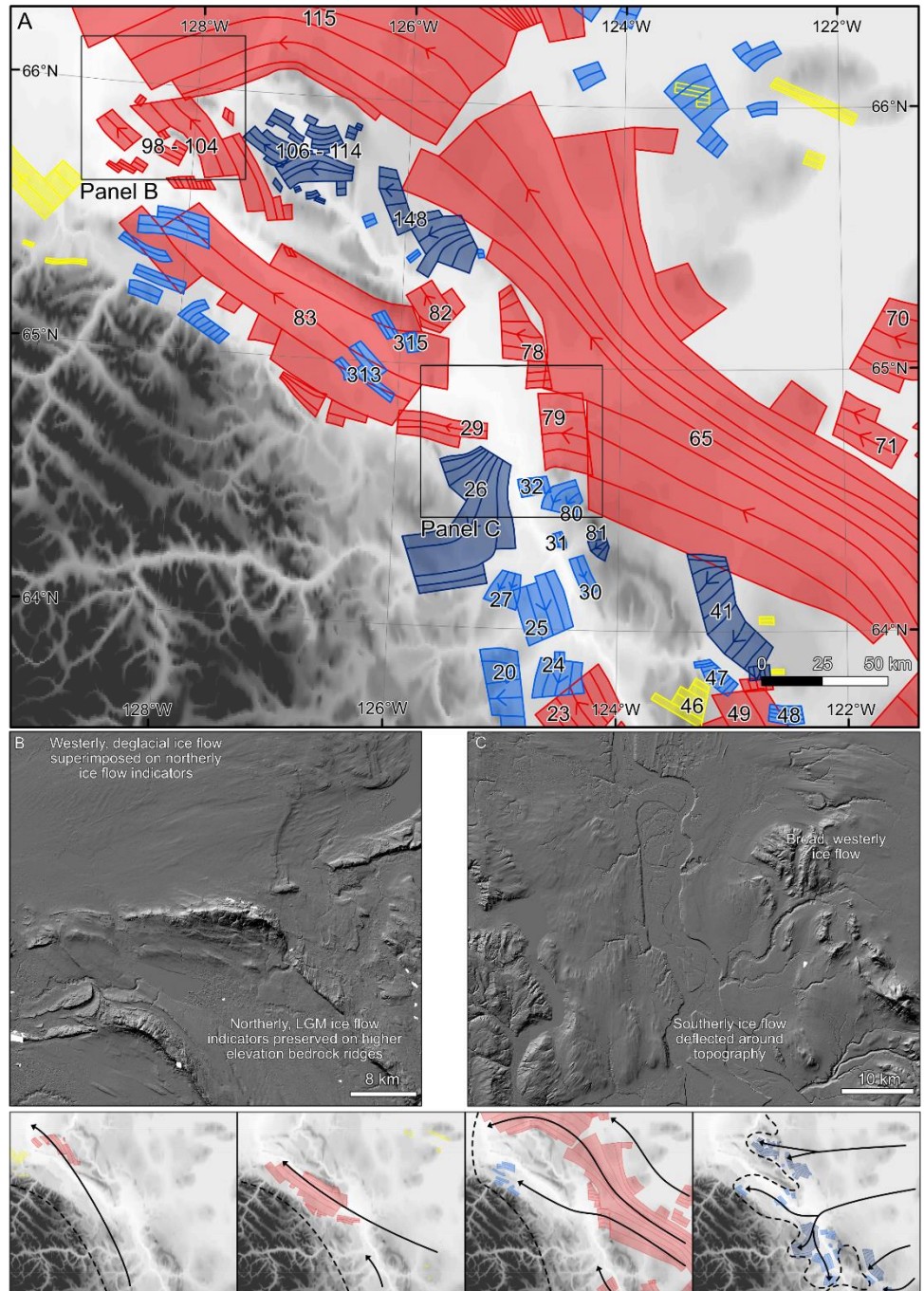

**Figure 8: (A) The ice flow patterns in the central Mackenzie Valley. Only key flowsets are numbered and unclassified flowsets are not depicted. (B) Fragmented northerly flowsets located on higher elevation bedrock ridges that formed**

during the local LGM and are overprinted by the westerly deglacial ice flow in the valley bottom that is superimposed on the northerly ice flow indicators, note the E-W trending esker which indicates final ice retreat to the eastand (C) Ice

flow reversal in the central Mackenzie Valley as broad, westerly ice flow is funnelled around a topographic high in the Mackenzie Valley. The lower panels show a schematic interpretation of the ice flow (black arrows) pattern changes during deglaciation and approximate ice margins positions (dashed lines). Note, ice margins in the lower panels are purely schematic.

## 6.2 Was deglaciation characterised by dynamic ice margin retreat or more widespread regional stagnation?

The processes of dynamic margin retreat and regional stagnation (i.e. cessation of ice flow and surface downwasting) have both been proposed to explain the style of deglaciation for different regions of the western LIS (Sharpe et al., 2021; Evans et al., 2021; Norris et al., 2023). The model of dynamic ice margin retreat has dominated, especially in broad-scale ice margin reconstructions (Dyke et al., 2003; Dalton et al., 2020), while stagnation has been proposed at a more local scale (Sharpe et

al., 2021). Historically, proponents of a dynamic style of ice margin retreat have argued for time-transgressive glacial landforms construction (Wilson, 1939; Dyke and Dredge, 1989) whereas opponents have argued for synchronous landform creation from the margin to distances far up-ice, with the ice sheet subsequently stagnating in place (Shilts et al., 1979; Shilts, 1985; Aylsworth and Shilts, 1989; Brennand, 1994, 2000). This debate has typically considered the processes of active retreat and stagnation as entirely unrelated. Instead, it is perhaps more appropriate to consider the processes of ice margin retreat as a

spectrum from active margin retreat (i.e. annual ice margin oscillations indicated by closely spaced, recessional push moraines, such as those developed at temperate Icelandic glaciers; Evans et al., 1999; Chandler et al., 2020) to regional stagnation, with the processes of pulsed recession and punctuated stagnation occurring somewhere in the middle (e.g. Dyke and Evans, 2003; Evans, 2009; Evans et al., 2020; Norris et al., 2023).

The retreat pattern of the southwestern LIS margin has recently been divided into three regional zones based on the ice marginal

processes that dominated during deglaciation (Norris et al., 2023). Initial deglaciation was characterised by rapid, active ice margin retreat based on the widespread occurrence of recessional push moraines (Norris et al., 2023). An intermediate zone of ice lobe surging and punctuated stagnation, caused by the rapid loss of the ice saddle, is proposed based on the presence of surge landsystem signatures and inter-ice stream zones of hummocky terrain. The final inner zone is characterised by inset push moraines and extensive esker networks, interpreted to indicate a return to active ice retreat. The loss of the ice saddle and

the concomitant rapid ice sheet reconfiguration is therein interpreted to have temporarily resulted in surge overprinting and ice stagnation over relatively extensive areas, with active ice retreat resuming once the ice sheet geometry readjusted (cf. Evans et al. 1999, 2008; Norris et al., 2023).

No such broad-scale characterisation of the ice marginal retreat processes and thermal regime exists for the northwestern LIS, but several studies have reconstructed these processes over a smaller geographical area. In the Smoking Hills region, the retreat

of the LIS was dominated by pulsed ice-marginal recession and associated punctuated stagnation, evidenced by ice-cored hummocky moraine belts fronted by sharp-crested push moraines. This is an assemblage representative of controlled moraine

construction in polythermal ice sheet margins that have undergone local readvances (cf. Dyke and Evans 2003; Evans, 2009; Evans et al., 2021). In contrast, Sharpe et al. (2021) suggest that ice sheet retreat across western Keewatin, to the east of Great Slave Lake, was dominated by regional stagnation. This is contested, however, with multiple studies questioning whether

stagnation was truly widespread (Campbell and Eagles, 2014; Campbell et al., 2016, 2018, 2019, 2020; Lauzon and Campbell, 2018; Livingstone et al., 2020; McMartin et al., 2021; Brouard et al., 2022; Dyke and Campbell, 2022; Vérité et al., 2024) The glacial lineation patterns here are interpreted as forming prior to, not during, deglaciation and a proposed absence of ice marginal landforms is cited as evidence of stagnation. While eskers and meltwater corridors are widespread, they are interpreted as 'long-conduit' deposits that form extensively up-ice, especially as there is an absence of associated fans or flat-

topped deposits forming esker beads and hence indicative of ice marginal positions (Sharpe et al., 2021).

Below, we separate the former bed of the northwestern LIS into three zones. For each zone we outline the thermal regime and processes of ice margin retreat based on the glacial geomorphology depicted in the map of Dulfer et al. (2023).

### 6.2.1 Retreat style during the early stages of deglaciation (Mackenzie Delta and Anderson Plains region; ~17.5 – 14.5 ka)

The deglaciation of the northwestern LIS was dominated by pulsed ice margin retreat, with localised regions of stagnation often limited to upland areas and controlled moraine belts. As outlined above, in the north, controlled moraine arcs lying inboard of sharp-crested push moraines indicate localised readvances of polythermal ice lobes (Evans et al., 2021; Figure 9A); associated with these lobes are overprinted lineation patterns indicative of the warm-based streaming ice located upflow of the frozen snouts (sensu Dyke and Evans 2003). The clearest example of this is in the lower Mackenzie Valley and Mackenzie

Delta region, where the Husky Lake thrust moraines indicate the readvance of streaming ice across the Mackenzie Delta to dislocate large masses of ground ice (Dyke and Evans, 2003; Figure 6D). The controlled moraines of the Sitidgi Stade and a series of recessional moraines associated with lobate lineation patterns record pulsed retreat of an ice lobe up the Mackenzie Valley (Figure 6F and G). The varying moraine morphology records complex changes in the thermal regime, as phases of enhanced coupling with the submarginal permafrost resulted in higher fluxes of supraglacial debris via englacial bands and the

construction of controlled moraine ridges (Dyke and Evans, 2003). In contrast, isolated regions of higher ground with hummocky terrain may represent localised areas of stagnation as ice on the uplands became separated from the relatively dynamic ice within the valleys (Figure 9B). This suggests that the pulsed style of polythermal ice margin retreat detailed by Evans et al. (2021) was commonplace during the early stages of deglaciation of the northwestern LIS.

### 6.2.2 Retreat style during the intermediate stage of deglaciation (Northern Interior Plains; ~14.5 – 13.0 ka)

The deglaciation style of the central Mackenzie Valley region during the Bølling–Allerød was dominated by rapid, dynamic ice margin retreat. Widespread deglacial flowsets with clear cross-cutting patterns indicate rapid flow reorganisations during the dynamic retreat of the margin. The landforms of the well-established Kelly Lake Phase (Hughes, 1987) provide a clear ice margin location at the edge of the Mackenzie Valley (Figure 6I). Approximately 200 km to the east, a further prominent

moraine system lies between two sites dated by cosmogenic nuclide exposure dating and indicates the ice margin position

towards the end of the Bølling–Allerød (Figure 6J). The recessional moraine sequence around the Norman Range (Figure 9C) and the absence of controlled moraines indicate a transition to warm-based margin conditions may have begun to dominate, in contrast to the early stages of deglaciation, where a complex polythermal regime was dominant.

Geomorphological evidence suggests that stagnation was localised during this period, being limited to upland areas bordering former ice streams. Along the terminus of the Bear Lake Ice Stream, extensive networks of crevasse fill ridges associated with

Fs-65 (Figure 6H) indicate a rapid shut-down of a possibly surging ice lobe and hummocky terrain with ice-walled lake plains suggest stagnation of ice on the neighbouring uplands (Figure 9D), similar to landsystem signatures reported by Evans et al. (2008, 2016, 2020) for the southwestern LIS. Although, the stagnation is much more localised for the northwestern sector than suggested for the adjacent southwestern sector (Norris et al., 2023). The overprinting of deglacial flowsets wrapping around topographic highs at lower elevations, indicates that ice margin retreat remained relatively active at lower elevations (Figure

6I, Fs-148). In the south of the region, the Liard and Snake Creek moraines record the localised readvance of the LIS margin, driven by the activation and possible surge of the Great Slave Ice Stream (Figure 6I, Fs-7 and 284). At this time, the geomorphological evidence indicates a much greater lobation of the ice sheet margin than depicted by Dalton et al. (2023). While crevasse fill ridges and hummocky terrain on the Sambaa K'e uplands indicate an abrupt shut-down of the Great Slave Ice Stream (Figure 6I), the absence of broad tracts of hummocky terrain in the inter-ice stream regions is interpreted here to

suggest that regions of ice stagnation were limited. In contrast to the southwestern LIS, where the rapid change in the ice sheet geometry resulted temporarily in relatively extensive ice stagnation (Norris et al., 2023), the northwestern LIS adjusted to the change in ice sheet configuration by dynamic ice margin retreat.

### 6.2.3 Retreat style during final deglaciation (Canadian Shield; ~13.0 – 10.5 ka)

Dynamic ice margin retreat continued as the LIS receded over the Canadian Shield, but the geomorphic expression differed to that of the Northern Interior Plains. On the Canadian Shield to the east of Great Slave Lake, Sharpe et al. (2021) interpreted meltwater corridors and associated eskers and an absence of moraines to indicate that retreat was dominated by regional stagnation of the ice sheet. In contrast, in our study region on the Canadian Shield, flat-topped deposits or fans are commonly observed in association with eskers (Figure 9E and F). Such beaded esker systems are commonly associated with sequential

formation of esker segments at or near the ice sheet margin (e.g. Mäkinen, 2003; Stoker et al., 2021). Additionally, overprinting lineation patterns record reorganisations of the ice flow units during deglaciation (Figure 9F). In some locations there is a clearly increasing influence of topography on ice flow, as small ice lobes were developed locally, conflicting with the synchronous formation of lineations proposed further to the east by Sharpe et al. (2021) (Figure 6O, Fs-251; Figure 6P, Fs-250 and 254). Therefore, we suggest that widespread stagnation did not occur for the northwestern LIS margin. Instead, we

propose that pulsed margin retreat likely dominated, with lower ice margin retreat rates (Reyes et al., 2022; Dalton et al., 2023) and with slower ice velocities (Fs 249-260) compared to the intermediate stage (14.5 – 13.0 ka). Our findings are more broadly

aligned with studies suggesting that ice margin retreat was stepwise and time-transgressive (Campbell and Eagles, 2014; Campbell et al., 2016, 2018, 2019, 2020; Lauzon and Campbell, 2018; Livingstone et al., 2020; McMartin et al., 2021; Brouard et al., 2022; Dyke and Campbell, 2022; Vérité et al., 2024), with the signature of stagnation being localised.

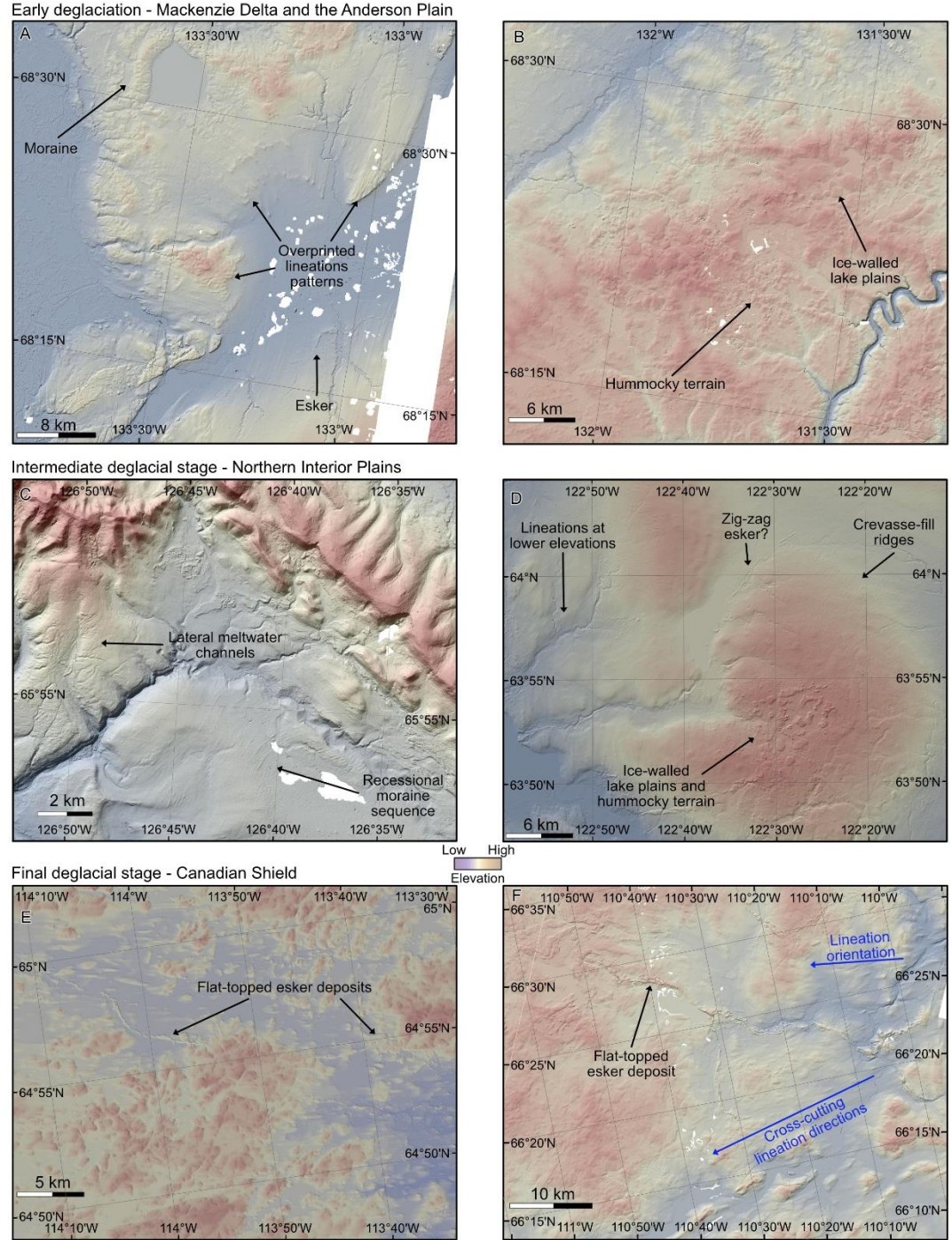

**Figure 9: Glacial landforms examples from the former northwestern LIS bed which highlight the contrasting styles of deglaciation across the region. Panel locations are indicated on Figure 1. Note some landforms may only be visible at**


higher zoom levels. (A) Overprinting lineation patterns in the Mackenzie Delta region which highlight the changing ice flow directions during dynamic margin retreat to the south. (B) Hummocky terrain on the upland areas of the Anderson Plain, including ice-walled lake plains, which suggest localised ice stagnation. (C) Recessional moraine sequence indicating ice margin retreat to the southeast. (D) High-elevation hummocky terrain and crevasse-fill ridges on the southern edge of the former Bear Lake Ice Stream indicating the rapid shutdown of the ice stream and stagnation of ice at higher elevations. (E) Flat-topped esker deposits on the Canadian Shield indicating possible former ice margin locations during active ice margin retreat. (F) Cross-cutting lineation pattern on the Canadian Shield indicating the changing ice flow geometries during retreat.

### 6.3 How did the ice stream network evolve spatially and temporally during deglaciation?

The extent of fast ice flow back into the ice sheet interior is an important consideration when reconstructing former ice drainage networks and their evolution. The formation of glacial lineations was originally assumed to occur near to the ice sheet margin (Craig, 1964; Aylsworth and Shilts, 1989), but it is now accepted that lineations can form far behind the ice sheet margin (Stokes et al., 2009) and, indeed, they have been observed forming 10s of kms up-ice under modern ice streams (King et al., 2009). Previous reconstructions of the ice streaming activity in the northwestern LIS have highlighted uncertainty over whether the Mackenzie Trough, Anderson, Bear Lake and Fort Simpson corridors of fast ice flow represent distinct ice streams operating during deglaciation or whether they record a major ice stream system whose drainage route evolved through time (Brown, 2012; Margold et al., 2015a; 2018; Stokes et al., 2016). These previous reconstructions remain somewhat ambiguous about the exact configuration of the ice stream network and its extent during deglaciation.

We reconstruct both extensive and shorter time transgressive ice streams in the northwestern LIS. The marine-terminating ice streams operating at the local LGM are reconstructed to have penetrated far into the interior of the ice sheet (up to 600 km), although the flowsets that serve as the evidence only cover a fraction of the ice stream bed (Fs 89, 98, 100, 103 and 104 in the Mackenzie Trough Ice Stream and Fs 221, 223, 226 in the Amundsen Gulf Ice Stream; Figure 6B). There is also evidence of the reorganisation of these ice streams during deglaciation (e.g. Fs 101 and 102 in the Mackenzie Trough Ice Stream and Fs 222 and 239 in the Amundsen Gulf Ice Stream; Figure 6f). These long ice streams remained active during deglaciation and stepped-back with the margin (Figure 6F, G, H). In this sense, the Mackenzie Trough, Anderson and Bear Lake ice streams represent different generations of the same large ice stream system draining into or towards the western Beaufort Sea.

The retreat of the ice margin onshore led to the shut-down of these large marine-terminating ice streams and the terrestrial ice streams that subsequently switched-on were shorter in extent (up to 200 km) and comparably short-lived, existing for only single timeslices in our reconstruction (e.g. the Haldane Ice Steam, Fs-129; Fort Simpson Ice Stream, Fs-49; Paulatuk Ice Stream, Fs-138). Some terrestrial ice streams, which were lake-terminating (Lemmen et al., 1996; Dyke, 2003; Figure 7) were larger in size (e.g. Bear Lake Ice Stream, Fs-65 and 115 and the Great Slave Ice Stream, Fs-7 and 284) (Smith, 1992, 1994). Observations of crevasse-fill ridge networks along the margin of the Bear Lake and Great Slave ice streams may indicate surge-behaviour and rapid shut-down of these ice streams (cf. Evans et al. 2008, 2016). This differs to the early, long ice

streams (Amundsen Gulf and Mackenzie Trough ice streams) which stayed active for as long as 3,000 years during deglaciation and evolved as the ice margin stepped back.

## 6.4 The influence of ice stream activity on the mass balance of the northwestern LIS

The last glacial termination was characterised by periods of rapid deglaciation interspersed with periods of ice margin stabilization/advance (Figure 10B; Laskar et al., 2004; Rasmussen et al., 2014; Praetorius and Mix., 2014). Global climate oscillations are correlated with and appear to drive these periods of enhanced or reduced rates of ice sheet deglaciation (Lambeck et al., 2014). However, the precise rates of deglaciation are often non-linearly related to climate, which suggests internal mechanisms may modulate ice sheet retreat rates (Charbit et al., 2002; Jones et al., 2021). As such, there are questions
around the relative extent to which past ice margin retreat is controlled by the millennial-scale climatic changes or changes in ice sheet dynamics (Young et al., 2020; Lowell et al., 2021).

The retreat rate of the northwestern LIS shows a clear correlation with these climatic oscillations during retreat across the Northern Interior Plains to the Canadian Shield (Reyes et al., 2022; Stoker et al., 2023). Prior to the Bølling–Allerød interval, ice margin retreat was relatively slow, at less than 100 m/year (Figure 10D). As global temperatures increased during the
Bølling–Allerød interval, the ice margin retreat rates increased up to 800 m/year across the Northern Interior Plains and over 350 m/year along Amundsen Gulf (Figure 10, Reyes et al., 2022; Stoker et al., 2022; Dalton et al., 2023). The peak ice margin retreat rates occur towards the middle of the Bølling–Allerød, over 1,000 years after temperatures began to rise (Figure 10). This is because a significant amount of deglaciation occurs through ice sheet thinning (Stoker et al., 2022), while the ice margin retreated relatively slowly. During this time, the ice sheet was characterised by a soft-bed and widespread ice-marginal glacial
lakes (Figure 7). The ice margin stabilised as it retreated onto the Canadian Shield at the start of the Younger Dryas, with retreat rates of less than 100 m/year (Figure 10c and d). The Canadian Shield constitutes a hard subglacial bed and, in our study area, displays a relative absence of glacial lakes (Figure 7). The combination of these factors result in increased basal shear stress and the stabilisation of the ice sheet margin as ice stream activity is reduced. The ice sheet geometry underwent fundamental reorganisations and changes during this time. .

### 6.4.1 Changes in ice stream activity during retreat across the Northern Interior Plains

Following the local LGM, the ice stream network was stable during generally slow margin retreat (<100 m/year; Figure 10D). Temperature increases prior to, and then more rapidly during, the Bølling–Allerød Interstadial led to a lowering of the ice sheet surface and increased ice margin retreat rates (Gregoire et al., 2016; Stoker et al., 2022; Figure 10). Ice stream activity also peaked at the start of the Bølling–Allerød Interstadial and then rapidly decreased until no ice streams were active in our study
area at the end of the Bølling–Allerød (Figure 10E).  In the saddle region, this warming resulted in a rapid expansion of the ablation area and initiated a mass-balance/ice sheet surface elevation feedback, whereby ice sheet thinning lowered larger areas of the ice saddle region to warmer elevations where enhanced melting occurred (Gregoire et al., 2012). This may have caused a steepening of the ice sheet surface profile from the Keewatin Ice Dome and increased driving stresses in ice stream onset

areas and the subsequent acceleration of ice streams (Robel and Tziperman, 2016). Through this mechanism, the climatically

triggered collapse of the ice saddle can likely explain the peak in ice stream activity observed in the empirical record (Figure 10E and F). Increases in ice streaming during ice saddle collapse events have been observed in numerical modelling simulations and have previously been attributed to increases in the area of ice margin through which ice can flow as ice sheets separate (Ely et al., 2024). As our reconstructed increase in ice stream activity begins and peaks prior to the separation of the CIS and LIS, we believe that changes in the ice sheet surface slope driven are the more likely cause of our reconstructed ice streaming

changes. Warming temperatures and enhanced surface melt likely led to the increased delivery of meltwater to the subglacial system and high subglacial water pressures may have facilitated fast ice flow (Clayton et al., 1985; Shabtaie and Bentley, 1987; Kamb, 2001; Zwally et al., 2002; Winsborrow et al., 2010; Margold et al., 2018)

In our reconstruction, ice stream activity rapidly decreases following the peak in ice stream activity and does not correspond to any climatic or geologic changes (Figure 10). Therefore, we suggest ice drawdown created a thinner ice sheet profile and

contributed to ice stream slowdown. The low ice surface slope in a thin ice sheet is associated with a reduction in the driving stresses in ice stream onset zones leading to ice stream slowdown (Robel and Tziperman, 2016). Indeed, in other regions of the LIS, ice drawdown during ice streaming has been suggested to cause a thinner ice sheet profile (Stokes and Clark, 2003). As the observed changes ice stream activity (Figure 10) occurred prior to the Younger Dryas cool period and without any observed changes in geological conditions, climatic and geological factors are not considered to be the primary factors

influencing the reduction in ice stream activity.

Ice stream activity exerts an influence on ice sheet mass balance and may have preconditioned the northwestern LIS for rapid retreat rates across the Northern Interior Plains. Ice stream acceleration is associated with enhanced calving at the ice sheet margin, while ice drawdown from the ice sheet interior to the lower elevation marginal regions causes increased surface melting (Robel and Tziperman, 2016). Broadly speaking, the pervasive fast ice flow and peak in ice stream activity across the Northern

Interior Plains between 14.5 and 14.0 ka (Figure 10E) occurs immediately prior to the period of highest ice margin retreat rates. This might indicate that ice drawdown created a thinner ice sheet profile which was vulnerable to rapid retreat (Figure 10D) (Robel and Tziperman, 2016; Gregoire et al., 2016). Previous modelling simulations investigating the influence of saddle collapse on the retreat of the northwestern LIS have not included accurate ice stream processes, despite the fact that they may have contributed to maintaining the high rates of ice margin retreat (Gregoire et al., 2012, 2016). Our observations provide

further evidence that ice streaming may be an important factor in maintaining rapid retreat rates during saddle collapse.

### 6.4.2 The ice flow regime on the Canadian Shield

During the early Bølling–Allerød, the peak in ice stream activity resulted in the rapid drawdown of ice from the interior of the ice sheet to the ice sheet margin which thinned the ice sheet profile and caused a reduction in the driving stresses (Robel and Tziperman, 2016). Ice flow on the Canadian Shield was characterised by a simple radial ice drainage network dominated by a

slower flow regime. The transition to a slow flow regime and the reduction in ice stream activity begins at 13.5 ka, before the Younger Dryas and before the ice margin had retreated onto the Canadian Shield, based on cosmogenic nuclide exposure ages

(Reyes et al., 2022; Figure 10). Therefore, we argue that the thin ice sheet profile was the principal reason for the reduction in ice stream activity (Robel and Tziperman, 2016).

The reduction in dynamic mass loss due to this slower ice flow regime might have had a positive impact on the mass balance. Margin stabilisation due to ice dynamics changes and the reduction of the ablation area during cooling in the Younger Dryas may have led to a thickening of the ice sheet (Figure 10C and D). This possible ice sheet thickening is not associated with the re-establishment of ice streaming in our study area. These factors all likely combined to drive ice margin stabilisation and hence the relative importance of climatic versus ice sheet dynamical drivers is difficult to identify.

Our understanding of the magnitude of the regional climatic changes limits our ability to evaluate the influence of climate changes on past ice margin changes. Palaeoenvironmental reconstructions in the Northwest Territories for the period of the Younger Dryas do not show any significant changes in the climate (Ritchie, 1977; Lowdon and Blake, 1981; Rampton, 1988; Spear, 1993; Dalton et al., 2024). A stable climate during the transition to the Younger Dryas would mean that an alternate driver of the ice margin stabilisation is needed and our reconstructed reduction in ice streaming may have contributed to a positive mass balance and margin stabilisation. Caution is necessary when interpreting past local climatic changes, as the number of studies of Younger Dryas climate in the Northwest Territories are limited and observations from the surrounding regions suggest a cooling period occurred across the Northern Hemisphere (e.g. Praetorius and Mix, 2014). If cooler temperatures occurred during the Younger Dryas, the production and delivery of meltwater to the subglacial system would have been reduced and may have contributed to the cessation of ice stream activity. An improved understanding of climate in the Northwest Territories during climate oscillations following the local LGM is essential to assess the effects of climate and ice dynamics on the ice sheet retreat rate.

### 6.4.3 Other factors influencing ice stream activity

Across the study area, mapped ice streams are restricted to locations of soft-bedded conditions and commonly occur where the margin is either lacustrine- or marine-terminating (Figure 7 and 11). Soft-bedded conditions, which dominate across the Northern Interior Plains, are favourable for fast ice flow, as the layer of deformable subglacial sediment enhances subglacial deformation and/or basal sliding (Alley et al., 1986; Winsborrow et al., 2010; Smith et al., 2022). In contrast, the hard-bedded granitic geology of the Canadian Shield provides a greater frictional resistance to ice flow and is resistant to erosion, inhibiting the development of ice streams in the study area (Clark, 1994; Winsborrow et al., 2010). While the climatically-driven changes in the ice sheet geometry are likely the principal control on changing ice stream activity, the subglacial bed conditions across the Northern Interior Plains were favourable for a rapidly developing ice stream system during deglaciation. In contrast, the hard-bed of the Canadian Shield likely contributed to the cessation of ice stream activity.

Increased calving at marine or lacustrine margins encourages increased ice flow velocity and ice draw-down because basal shear stresses are reduced (Stokes and Clark, 2003, 2004; Sutherland et al., 2020; Quiquet et al., 2021; Scherrenberg et al., 2024). Hence, calving margins are an important potential control on the initiation and location of ice stream activity

(Winsborrow et al., 2010). The majority of large ice streams (e.g. the Bear Lake and Great Slave Lake ice streams) across the Northern Interior Plains are lacustrine-terminating (Figure 7; Lemmen et al., 1994; Smith, 1994; Dyke et al., 2003; Couch and Eyles, 2008), while the terrestrial ice streams (e.g. the Paulatuk and Haldane ice streams) are typically smaller. This highlights how a calving margin may be important to sustain larger areas of increased ice flow.

At the Canadian Shield boundary, the change to hard-bed conditions and decrease in the proportion of the ice sheet margin that terminated in a glacial lake likely contributed to the reduction in ice stream activity, but the relative influence of these factors and changes in the ice sheet surface slope are difficult to determine (Figure 10E). The re-activiation of ice streaming only occurs on the Canadian Shield in locations where a calving margin (lacustrine or marine) allows an increase in crevasse-driven ice loss and increases in ice flow velocity (Stokes and Clark, 2003, 2004; McMartin et al., 2021).The climatic cooling during the Younger Dryas likely contributed to a thickening of the ice sheet profile, but did not lead to the re-activation of ice stream activity. Ice streaming only occurred after the ice margin retreated and allowed the development of glacial lakes in the Thelon and Back River basins which contacted the steepened ice surface profile, triggering the Dubawnt Lake Ice Stream which readvanced and dammed the Dubawnt Lake, despite hard-bed conditions (Stokes and Clark, 2003, 2004). The Dubawnt Lake Ice Stream is overprinted on a series of converging, northward pattern of lineations that constitute the marine-terminating Chantrey Inlet Ice Stream that further highlights the importance of a calving margin (McMartin et al., 2021). Similar to the hierarchy of controls on ice streaming set out by Winsborrow et al. (2010), we suggest that the presence of a calving margin (either marine or lacustrine) is a strong control on ice stream activity and can overcome the stabilising effect of increased shear stresses on a hard subglacial bed.

### 6.4.4 The influence of changes to the chronological model of deglaciation

Our interpretation that warming during the Bølling–Allerød drove a steepening of the ice surface slope and a peak in ice stream activity is dependent on a robust regional ice margin chronology that links ice flow dynamics to millennial-scale climate oscillations (Figure 10). As such, any changes to the regional chronology might influence this interpretation and chronologies solely based on cosmogenic nuclide dating are inherently vulnerable to changes in the age calculation approach or production rate (Stoker et al., 2023). The ice margin chronology of Dalton et al (2023) used a multi-chronometer approach (exposure dating, radiocarbon and luminescence method) and the current chronology satisfies all the constraints. Therefore, we suggest that any future changes to the regional chronology are likely to be minor and the chronology is relatively robust. Recent work, for example, has identified that an exposure age calculation approach for the region will likely not shift the regional deglaciation earlier as this would result in conflict between different chronometers (Stoker et al., 2023). As such, if the chronology does change it is likely to only shift the deglaciation slightly later. This would result in the peak in ice streaming still occurring within the Bolling-Allerod and still being associated with warming.

The peak in ice stream activity is inherently spatially linked to the proposed mechanism of ice surface slope changes irrespective of the precise timing of these events. Cosmogenic nuclide exposure ages collected along two vertical elevation transects record a period of rapid ice sheet thinning regardless of the exposure age calculation method chosen (Stoker et al.,

2023). The location of these vertical elevation transects (one in the central Mackenzie Valley and one in the Southern Franklin Mountains) are spatially linked to the region of enhanced ice stream activity, providing a direct link between observations of rapid ice sheet thinning and increased ice stream activity. This supports our conclusion that the mechanism driving changes in ice streaming is the steepening of the ice sheet surface slope during surface mass balance/elevation feedbacks (i.e. ice saddle collapse). While the absolute timing of this thinning event can shift through time, the occurrence of rapid thinning at this location does not change. Therefore, the association between rapid thinning and increased ice stream activity in this region is robust. Furthermore, numerical modelling studies have linked changes in the ELA and subsequent changes in the ice sheet surface slope to increased ice streaming during deglaciation (Robel and Tziperman, 2016).

### 6.4.5 Summary

Atmospheric warming caused increases in the ice sheet surface slope during deglaciation and is interpreted to be the primary control on changes in ice stream activity, with glacial lakes and the subglacial bed conditions playing a secondary role. The increase in driving stresses in ice stream onset zones due to the steepening of the ice sheet surface profile during saddle collapse provides a mechanism to explain the peak in ice stream activity during the Bølling–Allerød. The increased ice draw-down during this peak in ice stream activity ultimately led to lower surface slopes and the reduction in driving stresses which is associated with the reconstructed decline in ice stream activity towards the end of the Bølling–Allerød and before any observed climatic cooling or geological changes. The cessation of ice stream activity due to the thin ice sheet profile continued until the ice sheet had retreated beyond our study area. As climate cooled during the Younger Dryas the reduction of the ablation area likely led to a thickening of the ice sheet, but the shallower ice surface profile following the rapid ice draw-down during the Bølling–Allerød meant that ice stream activity lagged this climatic change (Robel and Tziperman, 2016). During this time, calving into large proglacial lakes and a subglacial soft-bed played a secondary role by facilitating ice streaming and controlling the distribution and size of ice streams.

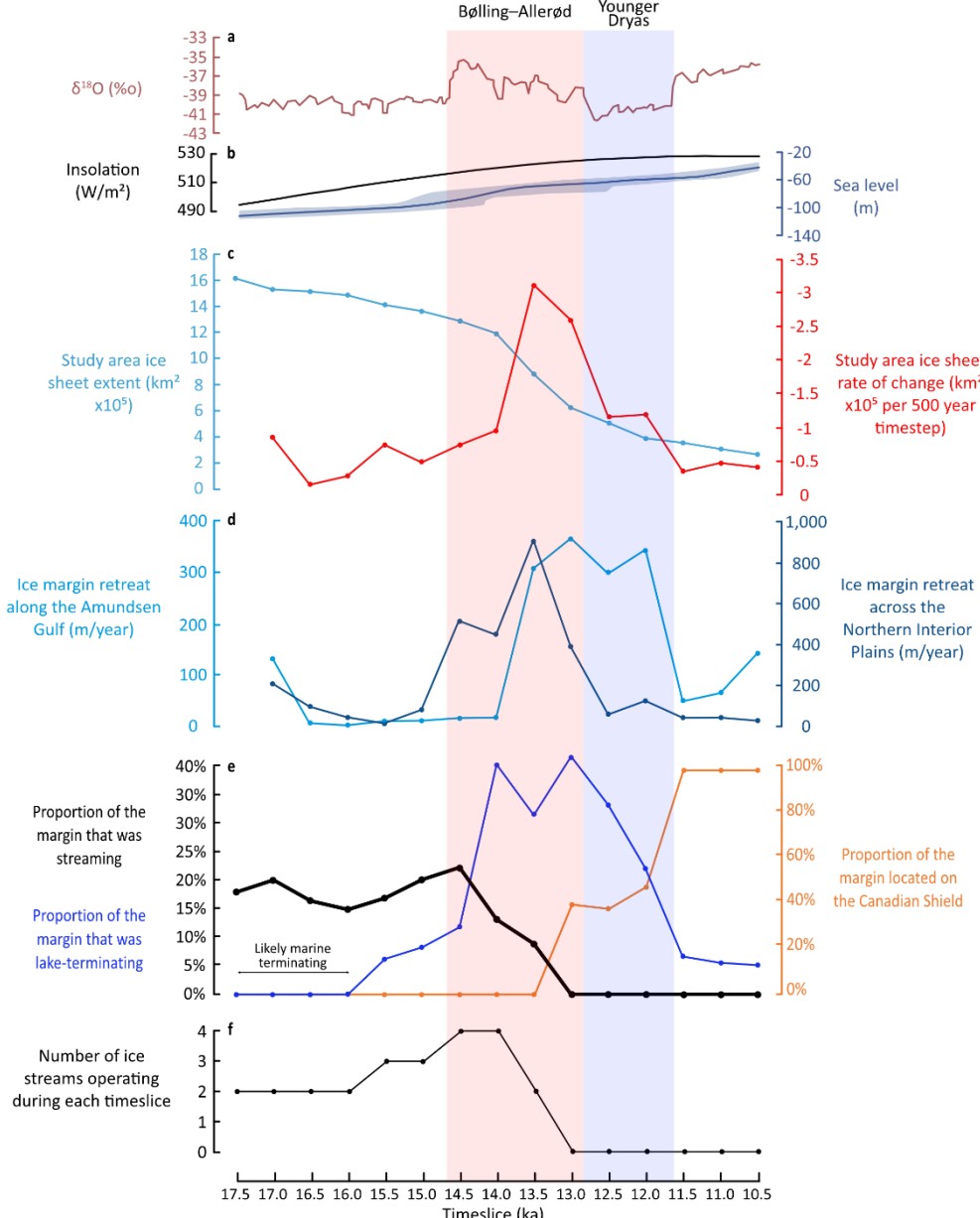

**Figure 10: (A) δ¹⁸O record from the Greenland GRIP ice core displayed in red (Rasmussen et al., 2014; Seierstad et al., 2014), (B) Relative sea level curve is blue is adapted from Carlson and Clark (2012) and based on a compilation of far-field sites and the mean summer insolation at 65°N is in black (Laskar et al., 2004). (C) The total extent of the LIS (blue line) in our study area (displayed in Figure 1) at each 500-year timestep covered in our reconstruction, based upon the reconstruction of Dalton et al. (2023). When the ice sheets were merged, the extent was calculated east of 127°W. The**

rate of ice sheet area loss per 500-year timestep is displayed in red. (D) The rate of ice margin retreat along two separate transects (displayed in Figure 1) for each 500-year timestep. (E) The proportion of the ice sheet margin in the study area that was streaming (thick black line) at each 500-year timestep based on our reconstruction (Figure 6). The proportion of the ice sheet margin that was lake-terminating (blue; calculated by measuring the extent of the ice margin from the reconstruction of Dalton et al (2023) that is located in an area of glacial lake extent from Dyke et al. (2003)) and the proportion of the ice sheet margin that is located on the hard-bedded Canadian Shield for each 500-year timestep (orange). (F) The total number of ice streams operating at each timestep of our reconstruction (Figure 6).

## 7 Conclusions

Using the glacial geomorphological map of Dulfer et al. (2023) we apply the established practice of flowset mapping and the glacial inversion method to reconstruct the ice flow evolution of the northwestern LIS at 500-year intervals. We categorise over 300 flowsets based on lineation morphology, orientation, patterns of overprinting and their relationship with other landforms.

The ice margin retreat of the northwestern LIS was dominated by pulsed recession of the margin, with stagnation restricted to controlled moraine deposits and isolated uplands. In the early stages of deglaciation of the Northern Interior Plains, widespread controlled moraines record the pulsed recession of a polythermal margin, with sharp ridges and push moraines marking the extent of readvances of this margin. To the south of ~66°N, there is a transition to a polythermal marginal regime and rapid retreat, with some evidence of active, warm-based retreat around the Norman Range. Stagnation is limited to the upland inter-ice stream areas, which became disconnected from the dynamic ice at lower elevations. Even though regional stagnation has previously been suggested for the deglaciation of the LIS margin across western Keewatin, we do not observe any indicators of widespread stagnation across the Canadian Shield (Sharpe et al., 2021).

We identify rapid changes in the ice drainage geometry and network, whereby predominantly northerly local LGM ice flow from the Cordilleran-Laurentide ice saddle transitioned to a westwards ice flow from the Keewatin Ice Dome over a period of 2000 years. The reorganisation of the ice drainage network is consistent with rapid thinning in the ice saddle region, driven by surface mass balance feedbacks, compared to a slower deglaciation of the Keewatin Ice Dome. The increased driving stresses due to the oversteepening of the ice sheet surface profile during the rapid thinning of the ice saddle provides a mechanism to understand the peak in ice stream activity during the Bølling–Allerød interval. The ice draw-down from the Keewatin Ice Dome following this peak in ice stream activity may have lowered the ice sheet surface profile and led to a reduction of ice stream activity towards the end of the Bølling–Allerød interval. During deglaciation, the presence of lacustrine calving margins and soft-bedded subglacial conditions appear to play a modulating role on ice stream activity. As the ice sheet retreated on to the Canadian Shield, the relative absence of a calving margin and hard-bedded conditions likely combined with climatically-driven changes in the ice sheet surface profile to cause the complete cessation of ice stream activity. Numerical ice sheet modelling studies have not yet fully captured the role of ice streaming processes when simulating the collapse of the

Cordilleran-Laurentide ice saddle. Our mapping results and reconstruction might provide a useful template for guiding and testing modelling experiments and we suggest future studies could aim to better represent ice streaming processes to investigate the drivers of rapid ice sheet retreat in the region.

**Data availability**

The data referred to in this paper have all been provided within the tables and figures in the main text and in the Supplement. The supplementary materials related to this article are available in the Supplement and at https://doi.org/10.6084/m9.figshare.25003559.v1.

**Funding sources**

BJS and HED acknowledge support by the project Grant Schemes at Charles University (reg. no. CZ.02.2.69/0.0/0.0/19_073/0016935; START/SCI/055). BJS and HED were fully supported by the START research project during all stages of this work. CDC was supported by the European Research Council, H2020 (PalGlac grant no. 787263). DF was supported by grants from the Natural Science and Engineering Research Council of Canada and Natural Resources Canada Polar Continental Shelf Program.

**Competing interests**

At least one of the (co-)authors is a member of the editorial board of The Cryosphere.

**Author contributions**

BJS: conceptualisation, methodology, formal analysis, writing – original draft, writing – review and editing, project administration, visualisation.

HED: conceptualisation, methodology, formal analysis, writing – original draft, writing – review and editing

CRS: conceptualisation, methodology, writing – original draft, writing – review and editing, supervision

VHB: writing – original draft, writing – review and editing

CDC: conceptualisation, writing – original draft, writing – review and editing

CÓC: writing – original draft, writing – review and editing

DJAE: writing – original draft, writing – review and editing

DF: conceptualisation, writing – original draft, writing – review and editing, supervision

SLN: conceptualisation, writing – original draft, writing – review and editing, supervision

MM: conceptualisation, methodology, writing – original draft, writing – review and editing, supervision

**Acknowledgements**

This study area covers the territories of many First Nations communities across the Northwest Territories and Nunavut and we acknowledge the First Nations people as the traditional owners of the land. We thank Rod Smith who has always been open in sharing his knowledge through multiple discussions and conversations during the course of this research. We are grateful to

1155 the two reviewers (Marion McKenzie and one anonymous reviewer) and to colleagues from the Geological Survey of Canada (Isabelle McMartin, Etienne Brouard, Janet Campbell and Pierre-Marc Godbout) who kindly took the time to read, review and provide comments which significantly improved the manuscript. We also thank Neil Glasser for his work in handling and editing the manuscript.

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
