# Peer review of "Ice flow dynamics of the northwestern Laurentide Ice Sheet during the last deglaciation"

_EGUsphere, 2024_

## Referee Comment (RC2)

**Ice flow dynamics of the northwestern Laurentide Ice Sheet during the last deglaciation**

Benjamin J. Stoker[1*], Helen E. Dulfer[1,2], Chris R. Stokes[3], Victoria H. Brown[3], Christopher D. Clark[2], Colm Ó Cofaigh[3], David J.A. Evans[3], Duane Froese[4], Sophie L. Norris[5] and Martin Margold[1*]

5  1 Department of Physical Geography and Geoecology, Faculty of Science, Charles University in Prague, Albertov 6, 12800, Praha, Czech Republic.

2 Department of Geography, The University of Sheffield, Sheffield S102TN, UK.

3 Department of Geography, Durham University, Durham, United Kingdom.

4 Department of Earth and Atmospheric Sciences, Faculty of Science, University of Alberta, Edmonton, AB, Canada.

5 Department of Geography, David Turpin Building, University of Victoria, Victoria, V8P 5C2, British Columbia, Canada.

10  *Correspondence to*: Benjamin J. Stoker (stokerb@natur.cuni.cz) or Martin Margold (margold@natur.cuni.cz)

**Abstract.** Reconstructions of palaeo-ice stream activity provide an insight into the processes governing ice stream evolution over millennial timescales. The northwestern sector of the Laurentide Ice Sheet experienced a period of rapid retreat driven by warming during the Bølling–Allerød (14.7 – 12.9 ka) which may have contributed significantly to global mean sea level rise during this time. [1] therefore provides an opportunity to investigate ice sheet dynamics during a phase of rapid ice sheet retreat.

15  Here, we classify coherent groups of ice flow parallel lineations into 326 flowsets and then categorise them as ice stream, deglacial, inferred deglacial or event type flowsets. Combined with ice marginal landforms and a new ice margin chronology (Dalton et al., 2023), we present the first reconstruction of ice flow dynamics of the northwestern Laurentide Ice Sheet at 500-year timesteps through the 2st deglaciation. At the local Last Glacial Maximum (17.5 ka), the ice stream network was dominated by large, marine-terminating ice streams (>1000 km long) that were fed by the Laurentide-Cordilleran ice saddle

20  to the south and the Keewatin Ice Dome to the east. As the ice margin retreated onshore, the drainage network was characterised by shorter, land-terminating ice streams (<200 km long), with the exception of the Bear Lake and Great Slave Lake ice streams (~600 km long) that terminated in large glacial lakes. Rapid reorganisation of the ice drainage network, from predominantly northerly ice flow to westerly ice flow, occurred over ~2000 years, coinciding with a period of rapid ice sheet surface lowering in the ice saddle region. We note a peak in ice stream activity during the Bølling–Allerød that we suggest is a result of increased

25  ablation and a steepening of the ice surface slope in ice stream onset zones and the increase in driving stresses which contributed to rapid ice drawdown. The subsequent cessation of ice stream activity by the end of the Bølling–Allerød was a result of ice drawdown lowering the ice surface profile, reducing driving stresses and leading to widespread ice stream shut-down.

**Summary of Comments on egusphere-2024-137.pdf**

**Page: 1**

Number: 1 Author: reviewer    Subject: Comment on Text    Date: 7/1/24, 3:52:34 PM
Clarify what "it" means in this sentence.  This will help with clarity for the reader.

Number: 2 Author: reviewer    Subject: Comment on Text    Date: 7/1/24, 3:58:00 PM
Define the period over which you mean for the reader.

[Figure]

**1 Introduction**

Ice streams are narrow regions of fast flowing ice that exert an important influence on ice sheet mass balance by drawing down ice to lower elevations, where it is lost through melting (surface or basal) or calving (Bamber et al., 2000; Bennett, 2003; Robel and Tziperman, 2016). As such, ice streams are a key process in numerical models to project the future evolution of contemporary ice sheets (Gandy et al., 2019). The geomorphological record created by Late Quaternary ice sheets provides a valuable opportunity to investigate ice stream dynamics, including the controls on ice stream activation, acceleration and shut-down (Winsborrow et al., 2010; Margold et al., 2018; Stokes et al., 2016). Crucially, the palaeoglaciological record allows us to investigate these systems over much longer timescales (1000s of years) than the contemporary observational record (10s of years) (Stokes et al., 2016; Gandy et al., 2019). The Laurentide Ice Sheet (LIS) was the largest component of the North American Ice Sheet Complex, which also included the Cordilleran Ice Sheet (CIS) and Innuitian Ice Sheet, during the last glaciation (Dyke et al., 2002, 2003; England et al. 2006; Seguinot et al. 2016; Stokes, 2017; Dalton et al., 2023). At the Last Glacial Maximum (LGM), 1 attained a similar extent to the present-day Antarctic Ice Sheet, providing an ideal location to study the behaviour of palaeo-ice streams.

The LIS reached 2s all-time maximum extent at the global LGM when the western margin coalesced with the CIS for the first time in the Quaternary, creating an ice saddle between the two ice sheets (Levson and Rutter, 1996; Jackson et al., 1997; Bednarski and Smith, 2007). Most ice sheets had reached their maximum between 26.5 ka and 19 ka, although there are significant regional variations in the timing ice sheet sectors reached their maximum and here 3e use the term 'LGM' to refer to the local maximum extent of the northwestern sector at ~17.5 ka (Clark et al., 2009; Dalton et al., 2023). 4umerical modelling indicates that abrupt climate warming during the 5ølling-Allerød interval led to the expansion of the 6blation area. This resulted in a mass balance-elevation feedback and rapid surface lowering in the ice saddle area, which is hypothesized to have caused rapid global mean sea level rise (Gregoire et al., 2012; Gomez et al., 2015; Gregoire et al., 2016). However, this scenario is subject to debate, with Pico et al. (2019) linking the demise of the ice saddle with the shut-down and retreat of the Amundsen Gulf Ice Stream between 8B ka and 11.5 ka and thereby suggesting that the 7ss of the ice saddle occurred during the cooler period of the Younger Dryas. Nevertheless, recently published chronological constraints from the Northwest Territories identify a period of rapid ice sheet thinning during the Bølling-Allerød interval (Stoker et al., 2022).

The rapid collapse of the Cordilleran-Laurentide ice saddle is likely to have had a dramatic impact on the ice drainage network and ice stream activity during the last deglaciation, but this has not yet been investigated in detail (Margold et al., 2015a, b, 2018). A review of ice stream activity of the LIS through the last deglaciation (Margold et al., 2015a, b; 2018) shows that there is clear evidence for major ice stream systems that drained the ice saddle and Keewatin Dome of the LIS during the last deglaciation (Figure 1) (Winsborrow et al., 2004; Kleman and Glasser, 2007; Brown et al., 2011, Brown, 2012). However, it remains unclear whether these large ice stream systems operated synchronously over large distances, or time-transgressively, switching on and off, with their trajectories migrating through time (Brown, 2012; Margold et al., 2015b, 2018). To the south of the saddle, on the southern Interior Plains in Alberta and Saskatchewan, for example, the
* * *
Number: 1 Author: reviewer    Subject: Comment on Text    Date: 7/1/24, 4:06:13 PM

I am not sure what the authors mean here.. Are the authors saying it (the LIS) attained a similar area as the modern AIS?

In any case, I don't think the extent really matters for why the ice sheet provides a test case for studying ice streams. I think the opportunity is simply that there are geomorphic features that may indicate paleo-ice streams and that alone makes this a nice place to study ice streams.
* * *
Number: 2 Author: reviewer    Subject: Comment on Text    Date: 7/1/24, 4:08:10 PM

In the abstract it says the local LGM was at 17.5 ka. Here the authors say the maximum was during the global LGM which is defined in Clark et al. 2009 and does not extend to 17.5ka. I suggest the authors clarify this so the readers are not confused.
* * *
Number: 3 Author: reviewer    Subject: Comment on Text    Date: 7/1/24, 4:10:51 PM

I think the authors should not attempt to re-classify LGM. Clark et al. 2009 has done a nice job of defining this and I think to not confuse the readers the authors should use local LGM which they do in the abstract. Redefining will confuse people and "local" allows flexibility in the regional definition.
* * *
Number: 4 Author: reviewer    Subject: Comment on Text    Date: 7/1/24, 4:11:28 PM

Provide a citation
* * *
Number: 5 Author: reviewer    Subject: Comment on Text    Date: 7/1/24, 4:13:53 PM

Define the time period here.
* * *
Number: 6 Author: reviewer    Subject: Comment on Text    Date: 7/1/24, 4:12:39 PM

Of the LIS or all NH ice sheets? Be explicit for the reader.
* * *
Number: 7 Author: reviewer    Subject: Comment on Text    Date: 7/1/24, 4:17:43 PM

This link the authors are calling on is a little hard to decipher. Perhaps in this sentence less is more and the authors can simply end the sentence at "....13.5 and 11.5 ka."
* * *
Number: 8 Author: reviewer    Subject: Comment on Text    Date: 7/1/24, 4:15:08 PM

This is confusing. The BA interval pre-dates this interval, so how is this connected?

[Figure]

[Figure]

**Figure 1: The ice stream reconstruction of the northwestern Laurentide Ice Sheet from Margold et al. (2015b) and selected glacial geomorphology (eskers, moraines and lineations) from the Glacial Map of Canada (Prest, 1968; Fulton, 1995). In the inset figure, the North American Ice Sheet Complex extent at the local LGM (17.5 ka) is depicted from Dalton et al. (2023). The approximate location of ice domes and ice divides is based on Margold et al. (2018) (LIS = Laurentide Ice Sheet, CIS = Cordilleran Ice Sheet, K = Keewatin, Q-L = Québec-Labrador dome, and F-B = Foxe-Baffin dome).**

[Figure]

70   geomorphological record presented by Norris et al. (2023) indicates that major shifts in ice flow were associated with rapid downwasting of the Cordilleran and Laurentide ice saddle (Norris et al., 2022), but similar evidence has yet to be identified in the Northwest Territories.

Recent advances in the resolution of remotely sensed data allow us to investigate large areas of the subglacial bed of former ice sheets in detail (Chandler et al., 2018). Dulfer et al. (2023) recently used the high-resolution (2 m) ArcticDEM to produce

75   a detailed glacial landform map of the northwestern LIS. Here we use this map to resolve the ice flow dynamics across the northwestern sector of the ice sheet. We employ a flowset mapping approach (e.g. Clark 1993, 1997; Kleman et al., 1997, 2006; Hughes et al., 2014) to reconstruct complex ice flow geometries and the relative ice flow dynamics, together with the distribution of ice-marginal landforms to identify former ice margin positions during ice retreat. The optimal ice margin chronology from Dalton et al. (2023) provides a temporal framework for our reconstruction, which allows us to investigate the

80   following questions:

  - How did the ice stream drainage network evolve spatially and temporally?
  - How did the lobation and thermal regime of the ice margin evolve through deglaciation?
  - How did the coalescence and separation of the ice sheets affect the configuration, timing and dynamics of the ice drainage network?
85   - What controlled the activation and shut-down of ice streams of the northwestern LIS?

**2 Regional glacial history**

Despite the remote nature of the region, the northwestern LIS, which spans western Nunavut and the majority of the Northwest Territories in Canada, has a long glacial research history. Since the early 1970s, extensive field surveys and aerial image mapping campaigns undertaken by the Geological Survey of Canada have detailed the surficial geology across the majority of

90   the former bed of this ice sheet sector (Hughes and Hodgson, 1972; Hughes, 1987; Rutter et al., 1993). This mapping has been at a range of scales, from detailed 1:50,000 maps (e.g. Bednarski, 2002, 2003a-o) to coarser scale 1:1,000,000 compilation maps (e.g. Duk-Rodkin, 1999; Duk-Rodkin, 2022) and the majority of the work has been conducted at the relatively coarse 1:125,000 scale (e.g. Klassen, 1971; Rutter et al., 1980; Paulen and Smith, 2022). Recent projects have sought to deliver more comprehensive coverage or to improve upon the existing mapping (e.g. Smith et al., 2021; Hagedorn et al., 2022; Paulen and

95   Smith, 2022). These studies have contributed to a detailed understanding of the former ice sheet limits and retreat pattern across the region, although it remains understudied compared to many other sectors of the former ice sheet (Rampton, 1988; Margold et al., 2018; Duk-Rodkin, 2022).

Our understanding of the northwestern LIS prior to the LGM is limited by the patchy record of older glaciations. Based on remote sensing evidence, Kleman et al. (2010) identified only a single westerly-directed ice flow event, which they correlated

100   to Marine Isotope Stage (MIS) 4. In the central Keewatin region, a series of six till units from drillcore logs record a counter-

Number: 1 Author: reviewer     Subject: Comment on Text     Date: 7/1/24, 4:25:06 PM
These questions are great to provide for the reader but I think equally important is provide the reader with the hypotheses that are being tested as well in this paper.

[revised manuscript text omitted]

Number: 1 Author: reviewer     Subject: Comment on Text     Date: 7/1/24, 4:35:53 PM
I am confused with this part of the sentence.  Do you mean it was short lived? Can you just say how long it remained at it's maximum position?

[Figure]

**3 Methods**

Our reconstruction is based on the glacial landform map of the northwestern sector of the LIS by Dulfer et al. (2023) which followed a uniform mapping approach across the entire region and was verified against surficial geological maps. This recent

170  mapping facilitates a consistent ice flow reconstruction across the entire study area at relatively high-resolution, albeit with some inherent limitations (discussed in Section 3.4). The map contains twelve landform categories, including ice flow parallel lineations, subglacial ribs, crevasse fill ridges (geometric ridge networks), major and minor moraine crests, hummocky terrain complexes and ridges, shear margin moraines, major and minor meltwater channels, lateral and submarginal meltwater channels, esker ridges and complexes, glaciofluvial complexes, perched deltas, raised shorelines and aeolian dunes. Here we

175  use the spatial distribution and characteristics of these glacial landforms to determine how ice flow and the related ice margins evolved over time, as outlined below.

**3.1 Flowset mapping**

We use the ~76,000 ice flow parallel lineations mapped by Dulfer et al. (2023) to decipher ice flow variation over time following the established approach of glacial flowset mapping (e.g. Clark 1993, 1997; Kleman et al., 1997, 2006; Greenwood

180  and Clark, 2009; Hughes et al., 2014). We first reduced the amount of lineation data by drawing generalized ice flow lines parallel to the lineations and then used these flowlines to group coherent patterns of lineations into discrete flowsets (Kleman and Borgström, 1996; Kleman et al., 2006; Figure 2). We acknowledge that this process is subjective and some flowsets may have formed time-transgressively, but the different flow phases could not be distinguished in the landform record.  Each flowset was assigned a number and the characteristics of the flowset were recorded, including the morphology of the flowset, the ice

185  flow direction, the location of cross-cutting lineations and associated glacial landforms (see Supplementary Table 1). During the creation of the final flowset reconstruction, some flowsets were deleted, merged or split, leading to some flowset numbers being absent from the final map (e.g. Fs 210, 211, 264).

Flowsets were classified into one of the following categories based on cross-cutting relationships and an assessment of the diagnostic criteria listed in Table 1: ice stream, deglacial, inferred deglacial and event flowsets (see also Figure 2). The ice

190  stream flowset category represents areas of past fast ice flow and are principally defined based on the elongation and parallel conformity of lineations, and the abrupt lateral edge of the flowset (Clark, 1993, 1999; Stokes and Clark, 1999). Ice stream flowsets can form in any position within an ice sheet, either near to the margin or towards the interior (Table 1, Figure 2e). In contrast, event and deglacial flowsets formed under a slower ice flow regime (Table 1) and are defined based on their location of formation within the ice sheet (Figure 2e). Deglacial flowsets formed near to the ice sheet margin and are identified based

195  on their association with other deglacial landforms (e.g. eskers or moraines, Table 1). Inferred deglacial flowsets are suggested to have formed near the ice sheet margin but are not associated with deglacial landforms. Rather, ice-marginal formation is based on fan-shaped lineation patterns or a clear topographic influence on lineations (Table 1). Event flowsets are inferred to have formed within the interior of the ice sheet and are identified based on the absence of, or discordance with, deglacial

Number: 1 Author: reviewer    Subject: Comment on Text    Date: 7/1/24, 4:49:51 PM

It's not clear to me how they were associated.  Is this defined elsewhere?  Is it done through cross cutting relations or simply if eskers and/or moraines are near flow sets?

[revised manuscript text omitted]

Number: 1 Author: reviewer     Subject: Comment on Text     Date: 7/1/24, 9:52:36 PM

It isn't clear what "matches" means.  Do the authors mean agreement?  Or, is it something else.?  If 64% match, does this mean that 36% are mismatched and therefore agreement is not very good?  Some clarity here would help the reader.

[Figure]

[Figure]

(Evans et al., 2021), but former ice margin positions in this region are also delineated by end moraines and lateral/submarginal meltwater channels. On the Canadian Shield to the east, these landforms become much less common and the deglacial retreat pattern is recorded by eskers. Here the ice margin is extrapolated between eskers based on the assumption that eskers form

380 time-transgressively perpendicular and proximal to the ice margin (Shreve, 1985; Storrar et al. 2014a; Livingstone et al., 2015). Our record of deglacial ice margin positions shows a large variation in ice flow direction throughout deglaciation (Figure 5). In the north of the study area, two opposing ice retreat directions are mapped: (1) ice retreat to the west, marked by the Husky Lake Moraines; and (2) ice retreat towards Victoria Island in the east (Figure 5). On the Horton Plain and in the northern Mackenzie Valley, the ice-contact landforms were deposited as the ice margin retreated to the south and southeast respectively

385 (Figure 5). Over the eastern portion of the study area, including the Great Slave Plain and the Canadian Shield, the former ice margins record a general ice retreat pattern towards the east, with localised southerly ice flow around topographic features (Figure 5).

In the north, where opposing ice retreat patterns are mapped, the ice margin retreat pattern is complex due to the separation of different major ice lobes, with the landform record documenting several interlobate ice margins. The most notable example is

390 located in the Coronation Hills and is delineated by a prominent moraine and esker system. Deglacial interlobate ice configurations have also been previously identified during deglaciation of the LIS in the Smoking Hills region of our study area (Evans et al., 2021). We provide a simplified reconstruction of the ice margin retreat pattern in this region.

[Figure]

[Figure]

**Figure 5: The ice margin retreat pattern of the northwestern Laurentide Ice Sheet based on the deglacial landforms from Dulfer et al. (2023). We label the landforms that have previously been identified and used to establish the deglacial retreat pattern in the literature. This includes the Liard Moraine (Smith, 1994), Snake Creek Moraine (Lemmen et al., 1994), landforms of the Kelly Lake Phase and Tutsieta Lake Phase (Hughes, 1987), Sitidgi Stade moraines (Rampton, 1988), and Husky Lakes thrust moraines (Dyke and Evans, 2003; Evans et al., 2021). The ice margin retreat pattern in**

[Figure]

**the Smoking Hills area is simplified due to the scale of the map, for a more detailed ice margin reconstruction in this**
400 **area see Evans et al. (2021).**

**5 Interpretation**

**5.1 Overview of ice flow evolution through time**

Prior to the local LGM, the northwestern LIS advanced westwards towards the Canadian Cordillera (Kleman and Glasser, 2007; Margold et al., 2018) (Figure 6a). The Amundsen Gulf Ice Stream was likely active during ice advance to the local LGM
405 extent and flowed into the eastern Beaufort Sea (Figure 6b; Stokes et al., 2006; Batchelor et al., 2014). However, there is limited geomorphological evidence of other ice streams being active at this time, which is likely because the evidence of ice advance has been overprinted by the deglacial signature or postglacial processes. The local LGM of the northwestern LIS occurred relatively late compared to the rest of the ice sheet (
17.5 ka, Dalton et al., 2023), following the coalescence of the LIS with the CIS and the formation of an ice saddle between the ice sheets. At this time, ice flow was directed predominantly
410 to the north or northwest across the study region. The ice flow network was drained by two major, marine-terminating ice streams, the Amundsen Gulf Ice Stream and the Mackenzie Trough Ice Stream, the former fed with ice from the Keewatin Ice Dome and the latter from the CIS-LIS ice saddle (Figure 6b; Batchelor et al., 2013, 2014; Margold et al., 2018).
Deglaciation of the northwestern LIS was characterized by periods of rapid retreat and thinning interspersed with periods of slow retreat (Reyes et al., 2022; Stoker et al., 2022; Dalton et al., 2023). The ice drainage and the ice stream network evolved
415 in response to these changes. The initial deglaciation of the northwestern LIS was characterized by slow ice margin retreat and a relatively stable ice drainage network (Dalton et al., 2023). The Amundsen Gulf Ice Stream was the main drainage outlet of the LIS in the eastern Beaufort Sea area and remained active throughout the early period of deglaciation, with minor reorganisations of the ice stream source area (Figure 6B-F; Stokes et al., 2006, 2009). The main drainage outlet in the western Beaufort Sea alternated between the Mackenzie Trough Ice Stream and the Anderson Ice Stream, which were likely not active
420 at the same time and were fed by the ice saddle between the CIS and LIS (Figure 6B-F).
The Bølling–Allerød interval (14.6 – 12.9 ka) was a millennial-scale warming event that occurred during the last deglaciation that is associated with rapid retreat of Northern Hemispheric ice sheets and glaciers (Lambeck et al., 2014; Menounos et al., 2017). Increasing temperatures during the Bølling–Allerød interval have been reconstructed to have triggered the collapse of the ice saddle between the CIS and LIS, which was followed by a period of rapid ice sheet thinning and retreat (Gomez et al.,
425 2015; Gregoire et al., 2016; Stoker et al., 2022). In response, the ice drainage network evolved rapidly, and the Keewatin Ice Dome became increasingly dominant as an ice source. This change in the relative dominance of ice source area is recorded in a shift from broadly northerly-oriented ice flow from the ice saddle to northwesterly flowing ice and finally westerly-oriented ice flow from the Keewatin Dome (Figure 6G-J). The ice stream network also adjusted rapidly to these changes. The loss of the ice saddle resulted in a limited contribution to the Amundsen Gulf Ice Stream directly from the mainland of Canada, with
430 ice for this ice stream coming from the Keewatin dome through Victoria Island (Figure 6G). The Paulatuk Ice Stream became

Number: 1 Author: reviewer     Subject: Comment on Text     Date: 7/3/24, 11:05:43 AM

I think it's also important to cite the original chronologic work that provided the timing of this age.  See references within the Dalton et al. paper.  This is important for attributing the original people who did thie work and not just the compilation paper of Dalton et la.

[Figure]

active in the area immediately west of the former Amundsen Gulf Ice Stream and its northwesterly-oriented flow direction reflects the increasing importance of the Keewatin Ice Dome (Figure 6H). In the northern Mackenzie Valley region, ice stream activity became dominated by westerly flow, as demonstrated by the activation of the Bear Lake Ice Stream (Figure 6G). Rapid retreat continued across the Northern Interior Plains following the separation of the CIS and LIS, with this period characterized

435 by westerly-oriented ice flow and relatively short-lived ice streams (Figure 6I and J).

The Bølling–Allerød interstadial was followed by the Younger Dryas Stadial (12.9 – 11.8 ka), which was a cool period that has been associated with glacier and ice sheet stabilization or advance (Lambeck et al., 2014), primarily in regions surrounding the North Atlantic (Rea et al., 2020; Mangerud, 2021). The beginning of the Younger Dryas coincides with a slowdown in the retreat rate of the northwestern LIS margin as it reached the Canadian Shield. This slow-down in ice retreat was accompanied

440 by a change in the ice flow dynamics (Figure 6K-P) from fast ice flow with numerous ice streams, which evolved rapidly through the deglaciation on the Northern Interior Plains, to a slower ice flow regime and a radial ice flow pattern fed by the Keewatin Ice Dome on the Canadian Shield. The exact control on this change in ice flow dynamics is unclear but is likely related to the switch from the soft-bedded, lake-terminating margin on the Northern Interior Plains to the hard-bedded, terrestrial margin on the Canadian Shield (see section 6.3 for a more detailed discussion).

445 In the following section, we describe in detail the changes in ice flow direction and dynamics at 500-year timesteps, constrained by the ice margin chronology of Dalton et al. (2023) (Figure 6). At various timesteps, we use our reconstructed ice margins (Figure 5) to provide further detail on the ice margin retreat pattern. The relatively coarse timestep of the ice margin chronology compared to our ice flow reconstruction means that conflicting flow directions may occur within a single timestep. In these situations, we define the distinct flow events as flow phases (whereby flow phase 1 represents an earlier flow event, which has

450 been overprinted by flow phase 2) and in Figure 6 we use different ice flow line styles to depict these separate flow phases.

**1.2 Detailed reconstruction of ice flow**

**5.2.1 Pre-LGM flowsets**

The signature of pre-LGM ice flow is sparse across the study area because the surficial geomorphological record typically only

455 records the most recent flow event (for more detail see section 3.4). Despite this, we assign multiple flowsets to the pre-LGM period (Figure 6A). Fs-240 (Figure 6A, mid-figure) is one of the oldest flowsets we identify and it represents early ice flow towards the west, as the LIS advanced over this region and before the initiation of the Amundsen Gulf Ice Stream. This pre-LGM westwards advance is well-established (Kleman and Glasser, 2007; Margold et al., 2018). The Amundsen Gulf Ice Stream operated as a major drainage outlet of the northwestern LIS throughout the Quaternary and likely operated prior to the

460 local LGM in this region (Batchelor et al., 2014). We interpret Fs-221, 223, and 226 (Figure 6A, top-right) as forming in the onset zone of an early Amundsen Gulf Ice Stream. In contrast, the Mackenzie Trough has a relatively short history of glaciation, with the offshore stratigraphic record suggesting only two previous Quaternary ice advances (Batchelor et al., 2013). The most

Number: 1 Author: reviewer     Subject: Comment on Text     Date: 7/3/24, 11:30:09 AM

I am finding it very difficult to see how the authors are able to parse out the ice sheet changes in 500 year intervals when the chronology in this region doesn't allow for it. I understand the authors' assumptions and can appreciate what they are trying to do, but laying out the structure in this detailed manner (to me) is giving the impression to the reader to much confidence in the chronology and relating to the geomorphic structures. Perhaps other experts will be okay with this but I think there might be too much inferring to make these detailed reconstructions at 500 year time steps. This of course is the central crux of this paper but I would prefer the authors be more conservative with their reconstructions and at least bin them into intervals of 1000s of years and not in these 500 year snapshots which I don't find convincing without more detailed chronology.

See my comments on Figure 6 regarding this section. Providing the chronologic constraints on the figure would go a long ways in giving the reader a better sense of how the authors are deriving these reconstructions. It will also provide the readers with all the information. I understand they are providing the published 500 years intervals from Dalton et la. but I still think providing the location and ages that those reconstructions are derived is important for this more detailed study.

Number: 2 Author: reviewer     Subject: Comment on Text     Date: 7/3/24, 11:15:28 AM

Cite Dalton et al. here who also noted this lack of information from 25-17ka based on lack of chronology as well.

[revised manuscript text omitted]

Number: 1 Author: reviewer     Subject: Comment on Text     Date: 7/3/24, 11:26:55 AM
To make this figure more useful to the reader, I think the authors should provide the location and chronologic constraints for their interpretation on the map (ie dates from whence the data came from and location)..

The reader shouldn't have to go back and forth between this paper and the Dalton et al. work to do this., and I think the authors would be making a clearer point about their assumptions by providing the ages on this plot.

[Figure]

figure, we only number key flowsets. An A4 version of each individual panel is provided in the Supplementary Folder where every flowset is numbered.

[Figure]

[Figure]

**Figure 6 cont:**

680

[Figure]

**Figure 6 cont:**

[Figure]

[Figure]

685

**Figure 7:** **The contrasting lineation morphology around Great Bear Lake. Note how the wavy, sub-parallel lineations at lower elevations differ from the more parallel features at higher elevations which they are overprinted on, probably due to ice flowing in a lightly grounded lacustrine environment.**

**6 Discussion**

690 In our reconstruction, the northwestern LIS undergoes a rapid reconfiguration of its ice drainage network during deglaciation characterized by a change from northerly and northwesterly oriented ice flow at the LGM to westerly-oriented flow during deglaciation. In this section, we discuss the implications of our reconstruction for the changing dominance of ice source regions and how this relates to saddle collapse, the style of ice stream activity, the nature of ice margin retreat processes, the controls on fast ice flow and the implications for ice sheet mass balance.

695

Number: 1 Author: reviewer     Subject: Comment on Text     Date: 7/3/24, 11:34:29 AM
This is beautiful DEM but I am not sure it is necessary for the main body of the manuscript.  I would consider moving it to supplemental.

[Figure]

[Figure]

**Figure 8:** **The relative age and extent of proglacial lakes across the study area during the last deglaciation, based on the reconstruction of Dyke et al. (2003). Black lines indicate the deglacial isochrones of Dalton et al. (2023). Note the mismatch between the isochrones and lake reconstruction as the glacial lake reconstruction has not been updated to match the new ice margin chronology.**

Number: 1 Author: reviewer    Subject: Comment on Text    Date: 7/3/24, 1:54:23 PM
This figure is interesting but I'm not sure it is necessary for the main body of the paper since the paper is largely about the ice history and not the lakes.

[Figure]

**6.1 How did the separation of the ice sheets affect the ice drainage network?**

The landform record displays a signal of relatively more rapid rates of thinning in the ice saddle region as the Keewatin Ice Dome became the dominant ice source region. Combined with the available chronological constraints, this provides support

705 for the demise of the ice saddle and separation of the ice sheets occurring before the end of the Bølling–Allerød interval. The dominance of northerly-oriented ice flow at the LGM and the start of deglaciation (Figure 6B-G) demonstrates a strong influence of the Cordilleran-Laurentide ice saddle on the ice drainage network. During the start of the Bølling–Allerød interval, the shift to more northwesterly-oriented ice flow occurs as surface mass balance / elevation feedback causes the rapid thinning of the ice saddle and weakens it relative to the Keewatin Dome (Figure 6H). The ice stream network responds to this change

710 as the Amundsen Gulf Ice Stream (Figure 6G; Fs 222 and 239), with its strong northwards signal, terminates and is replaced by the Paulatuk Ice Stream (Figure 6H; Fs-138) with a more northwesterly ice flow orientation. By the end of the Bølling–Allerød interval, ice flow was broadly westerly-oriented as the ice sheets had separated and the ice saddle had fully collapsed (Figure 6I).

An alternative scenario, where the rapid retreat of the Amundsen Gulf Ice Stream causes ice drawdown from the Keewatin Ice

715 Dome and leads to the collapse of the ice saddle, has been proposed by Pico et al. (2019). This connection between Amundsen Gulf Ice Stream retreat and the ice saddle collapse is based on numerical ice sheet models forced by different palaeotopographies. Based on radiocarbon ages, the collapse of the Amundsen Gulf Ice Stream took place after 13.0 ka and hence is used to suggest that the saddle collapse likely occurred after 13.0 ka (Lakeman et al., 2018; Pico et al., 2019). In fact, it is difficult to envisage a scenario where the Amundsen Gulf Ice Stream is not an important driver of rapid ice drawdown

720 and margin retreat due to the large ice fluxes it likely delivered through the marine terminating margin.

Based on the empirical constraints, we favour the rapid thinning of the ice saddle region as the driver of rapid ice margin retreat across the Northern Interior Plains. The landform record indicates that the final ice flow event in the Mackenzie Valley at ~65°N exhibits a clear southwards deflection around topography, within a broad westerly ice flow pattern from the Keewatin Ice Dome (Fs-25; Figure 6I and 9). Southerly ice flow up the Mackenzie Valley is overprinted on northerly ice flow from the

725 ice saddle region, and therefore is only possible following the total collapse or significant weakening of the ice saddle, which would otherwise dominate the westward signal of the Keewatin Ice Dome. Both radiocarbon and cosmogenic nuclide exposure age constraints show that the Mackenzie Valley at ~65°N was deglaciated and occupied by Glacial Lake Mackenzie by the end of the Bølling–Allerød interval (Stoker et al., 2022; Dalton et al., 2023). This final ice flow event (Figure 9; Fs 24-26 and 31-32) constrains the separation of the ice sheets to before the Younger Dryas and is incompatible with the scenario proposed

730 by Pico et al. (2019).

Number: 1 Author: reviewer        Subject: Comment on Text        Date: 7/3/24, 11:31:50 AM
Relative to what?  This should be clarified.

Number: 2 Author: reviewer        Subject: Comment on Text        Date: 7/3/24, 1:55:56 PM
What is meant by strong?  Clarify for the reader.

Number: 3 Author: reviewer        Subject: Comment on Text        Date: 7/3/24, 1:57:45 PM
This may very well be the case but I think it is important to provide the reader with the information about why "envisaging" this is difficult.

Number: 4 Author: reviewer        Subject: Cross-Out    Date: 7/3/24, 1:59:14 PM

Number: 5 Author: reviewer        Subject: Comment on Text        Date: 7/3/24, 3:54:06 PM
Again, I think it is very important to show some of these dates with the new reconstructions.  It provide the reader with enough information to make their own assessment of what the authors are arguing.

[revised manuscript text omitted]

Number: 1 Author: reviewer    Subject: Comment on Text    Date: 7/4/24, 8:45:43 AM

This section also seems tangential to the main points of the paper. I appreciate these sections but they seems to distract from the main thrust of the paper.

Number: 2 Author: reviewer    Subject: Inserted Text    Date: 7/4/24, 8:31:57 AM

and

Number: 3 Author: reviewer    Subject: Comment on Text    Date: 7/4/24, 8:35:52 AM

Again, and sorry to go on about this, but I find statements like this hard to justify without more information on the chronologic basis of the reconstruction of this paper. Without the ages clearly outlined for the reader, I find most of the descriptions in the text about the geomorphology general fine but without any basis. It is hard for me to assess this paper without that information and difficult to provide constructive feedback to the authors.

[Figure]

[Figure]
igure 11: (A) δ¹⁸O record from the Greenland GRIP ice core displayed in red (Rasmussen et al., 2014; Seierstad et al., 2014), (B) Relative sea level curve is blue is adapted from Carlson and Clark (2012) and based on a compilation of far-field sites and the mean summer insolation at 65°N is in black (Laskar et al., 2004). (C) The total extent of the LIS (blue line) in our study area (displayed in Figure 1) at each 500-year timestep covered in our reconstruction, based upon the

Number: 1 Author: reviewer     Subject: Comment on Text     Date: 7/4/24, 8:47:47 AM
I don't find this figure adds too much to the paper's main focus. I would consider removing to help streamline the paper and focus it on the main points as outlined in the conclusion.

[revised manuscript text omitted]

1050 acknowledge the First Nations people as the traditional owners of the land. We thank Rod Smith who helped shape the manuscript through multiple discussions and conversations.

Number: 1 Author: reviewer     Subject: Comment on Text      Date: 7/4/24, 8:58:16 AM
Good to acknowledge the original people but better if they were included meaningfully in the paper.

In any case, can the authors acknowledge specific people they worked with during the project (ie tribal members, leaders, or councilors)?  Otherwise, I personally find these statements performative as a person with indigenous heritage - e.g. the nations are recognized but were never asked for permission to map and describe their lands.  I could say more but I leave this for the authors' to consider and apologize if one of the coauthors is a tribal member and I overlooked their affiliations and/or heritage.

[revised manuscript text omitted]

---

## Community Comment (CC1)

**Field record of ice-flow indicators**

The "striation" symbol appearing on regional surficial geology maps includes glacial striae and any other small-scale erosional forms on bedrock (crescentic fractures and gouges, rat tails, nailhead striae, stoss-and-lee topography, etc., e.g. McMartin and Paulen, 2009). These features provide information on direction, sense, and relative age of ice flows, and on the record of older glaciations, commonly poorly preserved in the geomorphic record. The general orientation of roches moutonnées is also represented by a different symbol on the maps and provides an additional record of the ice flow direction.

It is mentioned in Section 3.4 that "Glacial striations also provide an opportunity for reconstructing former ice flow patterns and may preserve older flow traces on bedrock outcrops where abrasion was limited during deglaciation (Kleman er al., 1990)". However, we are concerned about the lack of consideration and integration between some of the field-based record and the remotely mapped geomorphology to inform and resolve the flowset directions and age relationships, particularly in areas with complex flow patterns during deglaciation. As mentioned in Section 2, the "understanding of ice flow dynamics … suffers from the disconnected nature of studies at varying scales". Although the ice-flow indicator record can be very detailed locally, striations indicated on surficial geology maps are often distributed regionally and can inform on former and deglacial ice flow patterns where no landforms are preserved. Field-based ice-flow indicators can easily be extracted from surficial geology maps or Open File reports, all available in digital format (using Advanced search in https://ostrnrcan-dostrncan.canada.ca/home), and compiled at the ice sheet sector scale. Such large-scale compilations have been completed recently directly east of the studied area in the west-central Keewatin sector of the LIS (Brouard et al., 2022; Fig. 5 below) and further east along Hudson Bay (Behnia et al., 2020; McMartin et al., 2021; DataS2c below) to help constrain the regional glacial history.

[Figure]

*Figure 5. Location of ice-flow measurements in west-central Keewatin and their associated ice-flow orientation (From Brouard et al., 2022).*

[Figure]

*DataS2c. Generalized ice-flow indicator map, Central Mainland Nunavut (From McMartin et al., 2021).*

Interpreted ice-flow indicators and glacial histories from some regional field-based studies were considered (e.g. Smoking Hills), but other available regional maps and Open File reports should also be verified in light of the proposed reconstruction. In this regard, an ice-flow data compilation in the Great Bear Magmatic Zone (Normandeau and McMartin, 2013; Fig. 4 and Appendix II) should be taken into consideration, as well as a regional deglacial history reconstruction in the northeastern Horton Plain region, particularly for the interpretation of a NE(?) flowset (Fs-230) and other relative ages of flowsets 224, 225, 231, 323 (St-Onge and McMartin, 1995 and references herein).

We also suggest a comparative analysis with Dyke and Prest (1987), the only other reconstruction offering a continental overview that shows ice-flow drainage and incorporates the extensive fieldwork data accomplished in Canada since the end of the 19th century. This comparison could provide a richer context for this study, especially in contrast to works primarily based on remote mapping. While the cost-effective approach outlined in this study is certainly practical, a more thorough integration of fieldwork data might offer a comprehensive perspective, reducing the reliance on subjective interpretations, particularly regarding the provision of relative chronology through striations.

Regional stagnation to the east of the studied area

In section 6.2.3, it says "Therefore, we suggest that widespread stagnation had not begun for the northwestern LIS margin to the west of 210°W but may have occurred further to the east, as documented by Sharpe et al. (2021)." First, it should probably be 110°W instead of 210°W, and in the Conclusions on

page 47 again: "If widespread stagnation of the margin occurred, we suggest that it must have occurred after the LIS retreated to the east of 110°W."

More importantly, no basis for the assumption that widespread stagnation may have occurred further east of the studied area is provided, other than what is presented in Sharpe et al. (2021). Recent and current high-resolution geomorphic mapping using ArcticDEM and extensive field-based studies from east of 108°W to the Hudson Bay coast does not support or indicate widespread stagnation (Campbell and Eagles, 2014; Campbell et al., 2016, 2018, 2019, 2020; Lauzon and Campbell, 2018; Livingstone et al., 2020; McMartin et al., 2021; Brouard et al., 2022, 2023; Dyke and Campbell, 2022; Vérité et al., 2024). In contrast, numerous evidence for a sequential, time-transgressive retreat is presented in these publications. If the authors of this study have not mapped east of 110°W, they should not assume or propose that a widespread stagnation characterized ice retreat east of their studied area unless a proper discussion with alternative views is presented.

Ice streams on the Canadian Shield

At Line 961, it says: "While hard-bed conditions are known to exert a stabilising influence on ice flow, large ice streams are observed across the Canadian Shield." Yes, further east in Keewatin over the Shield, large ice stream footprints have been reconstructed from geomorphic mapping (including high-resolution mapping with ArcticDEM), including not only the Dubawnt Lake ice stream (e.g. Stokes and Clark, 2003, 2004) but several other large ice streams (Margold et al., 2015, 2018; McMartin et al., 2021) and not necessarily involving glacial lakes (i.e. marine-terminating ice streams).

At Line 965, it says that "The development of the ice-marginal Dubawnt Lake to the east of our study is hypothesised to have triggered the Dubawnt Lake Ice Stream… (Stokes and Clark, 2003, 2004)". This is incorrect. Stokes and Clark conclude that "As the ice margin retreats back, early glacial lakes form in the Thelon and Back River drainage basins. These early glacial lakes in the Thelon River basin increased in size and deepened, triggering the Dubawnt Lake ice stream." They further suggest that "This results in widespread thinning of the ice sheet and produces a large glacial lake to the south of the ice stream, dammed by the Dubawnt Lake ice stream lobe ". This glacial lake is west of Dubawnt Lake (Stokes and Clark, 2004; Fig 8).

Another point concerns the influence of crystalline bedrock on ice-margin stability. The absence of ice-flow landforms on the Canadian Shield could be attributed to several factors. 1) The scarcity of sediments associated with harder-to-erode crystalline bedrock may hinder landform production. Thus, the absence of ice-flow landforms on crystalline bedrock does not imply the absence of ice streaming (especially on the eastern part of McConnell Lake); it merely indicates that no landforms were preserved. 2) Because crystalline bedrock is more challenging to erode, ice-flow landforms that directly record bedrock are generally smaller and likely too small to be noticed at the scale mapped. Utilizing ArcticDEM at a more detailed scale, such as 1:20,000 or 1:10,000, might have revealed these features.

ArcticDEM and high-resolution mapping

While discussing the benefits of high-resolution data for understanding glacial history, the authors did not actually map at the highest spatial resolution provided by ArcticDEM (2 m). Instead, they mapped at scales ranging from 1:50,000 to 1:100,000 (Dulfer et al., 2023) to cover as much area as possible. This mapping scale is similar to, or even lower in resolution than that in studies that used aerial photographs (1:15,000

to 1:60,000). The advantage of ArcticDEM lies in its complementary data, which enable the production of hillshades at a higher resolution than previously available (about 20-30 m resolution). While we understand the necessity of cost-effective methods, it appears there was an aim to produce a regional-scale product with supposedly high-resolution data, yet the execution was only partially completed. The authors should be cautious with their claims of using high-resolution data, as their practice does not match their assertion.

Uncertainties

Furthermore, the authors acknowledge a 1000-year uncertainty commonly associated with cosmogenic ($^{10}$Be) ages, which is foundational to the study's framework for ice-margin retreat. While this initial recognition is valuable, a more thorough discussion on how this uncertainty influences subsequent interpretations, particularly regarding suggested peak ice-stream activity at the onset of the Bølling-Allerød interstadial, would greatly enhance the narrative's robustness. Given the acknowledged uncertainties and the inherently subjective nature of both the approach and Dalton's ice-margin reconstruction, a more detailed exploration of these aspects could help strengthen the study's conclusions. While we agree with the authors' assertion that the Bølling-Allerød warming significantly influenced ice sheet melt and dynamic changes, further elucidation on how these uncertainties were navigated in reaching such conclusions would provide clearer insight into the analytical rigor applied throughout the study.

References

Behnia, P., McMartin, I., Campbell, J.E., Godbout, P.-M., and Tremblay, T. (2020). Northern Canada glacial geomorphology database: part 1 — central mainland Nunavut; Geological Survey of Canada, Open File 8717 (ver. 2020), 1 zip file, https://doi.org/10.4095/327796.

Brouard, E., Campbell, J.E., McMartin, I., and Godbout, P.M., 2022. Compilation of surficial geology field data for the west-central Keewatin Sector of the Laurentide Ice Sheet (Northwest Territories and Nunavut); Geological Survey of Canada, Open File 8915, 9 p, https://doi.org/10.4095/330559.

Brouard, E., Campbell, J.E., McMartin, I., Godbout, P.-M., and Roy, M., 2023. Ice-flow reconstruction and deglacial patterns in the west-central Keewatin Sector of the Laurentide Ice Sheet (Northwest Territories and Nunavut, Canada). INQUA, Rome, Italy, 1 poster.

Campbell, J.E., and Eagles, S., 2014. Report of 2014 activities for the geologic and metallogenic framework of the south Rae Craton, southeast Northwest Territories: reconnaissance surficial and bedrock fieldwork in the GEM 2 South Rae project area; Geological Survey of Canada, Open File 7701, 7 p., https://doi.org/10.4095/295463.

Campbell, J.E., Lauzon, G., Dyke, A.S., Haiblen, A.M., Roy, M. 2016. Report of 2016 activities for the regional surficial geological mapping of the south Rae Craton, southeast NWT: GEM 2 South Rae Quaternary and Bedrock Project. Geological Survey of Canada, Open File 8143, Natural Resources Canada, 1-16, https://doi.org/10.4095/299391.

Campbell, J.E., Normandeau, P.X., Godbout, P.M., and McMartin, I. 2018. GEM-2 Glacial Synthesis Project: highlights of 2018 field activities in the Healey Lake area, Northwest Territories and Nunavut. In:

Irwin, D., Gervais, S.D., and Terlaky, V. (compilers), 2018. 46th Annual Yellowknife Geoscience Forum Abstracts; Northwest Territories Geological Survey, Yellowknife, NT. YKGSF Abstracts Volume 2018, p. 8. https://www.nwtgeoscience.ca/forum/session/gem-2-glacial-synthesis-project-highlights-2018-field-activities-healey-lake-area

Campbell, J.E., McMartin, I, Normandeau, P.X., and Godbout, P-M., 2019. Report of 2018 activities for the GEM-2 Rae Project glacial history activity in the eastern Northwest Territories and the Kitikmeot and Kivalliq Regions, Nunavut; Geological Survey of Canada, Open File 8586, 16 p., https://doi.org/10.4095/314741.

Campbell, J.E., McCurdy, M.W., Lauzon, G., Regis, D. and Wygergangs, M., 2020. Field data, till composition and ice-flow history, south Rae craton, NWT: results from the GEM2 South Rae project - Surficial Mapping activity; Geological Survey of Canada, Open File 8714, 40 p., https://doi.org/10.4095/327218.

Dyke, A.S., and Campbell, J.E., 2022. Surficial geology, Abitau Lake, Northwest Territories, NTS 75-B; Geological Survey of Canada, Canadian Geoscience Map 350, scale 1:100 000, https://doi.org/10.4095/330072.

Lauzon, G., and Campbell, J.E., 2018. Surficial geology, Wholdaia Lake south, Northwest Territories, NTS 75-A south; Geological Survey of Canada, Canadian Geoscience Map 342, scale 1:100 000, https://doi.org/10.4095/306373.

Livingstone, S.J., Lewington, E.L., Clark, C.D., Storrar, R.D., Sole, A.J., McMartin, I., Dewald, N., and Ng, F., 2020. A quasi-annual record of time-transgressive esker formation: implications for ice-sheet reconstruction and subglacial hydrology; The Cryosphere 14(6), 1989–2004, https://doi.org/10.5194/tc-14-1989-2020.

Margold, M., Stokes, C.R., Clark, C.D., and Kleman, J., 2015. Ice streams in the Laurentide Ice Sheet: a new mapping inventory; Journal of Maps 11 (3), 380-395, https://doi.org/10.1080/17445647.2014.912036.

Margold, M., Stokes, C.R., and Clark C.D., 2018. Reconciling records of ice streaming and ice margin retreat to produce a palaeogeographic reconstruction of the deglaciation of the Laurentide Ice Sheet; Earth-Science Reviews 143, 117-146, https://doi.org/10.1016/j.quascirev.2018.03.013.

McMartin, I., and Paulen, R.C., 2009. Ice-flow indicators and the importance of ice-flow mapping for drift prospecting. In: Paulen, R.C., McMartin, I. (Eds.), Application of Till and Stream Sediment Heavy Mineral and Geochemical Methods to Mineral Exploration in Western and Northern Canada; Geological Association of Canada, Short Course Notes 18, Geological Association of Canada, 15-34.

McMartin, I., Godbout, P.-M., Campbell, J.E., Tremblay, T., and Behnia, P., 2021. A new map of glacigenic features and glacial landsystems in central mainland Nunavut, Canada; Boreas 50(1), 51-75, https://doi.org/10.1111/bor.12479.

Normandeau, P.-X. and McMartin, I., 2013. Composition of till and bedrock across the Great Bear magmatic zone: Quaternary field database and analytical results from the GEM IOCG-Great Bear Project; Geological Survey of Canada, Open File 7307, 22 p., https://doi.org/10.4095/292560.

St-Onge, D.A. and McMartin, I. 1995. Quaternary geology of the Inman River area, Northwest Territories. Geological Survey of Canada, Bulletin 446, 59 p., https://doi.org/10.4095/203578.

Stokes, C.R., and Clark, C.D., 2003. The Dubawnt Lake palaeo-ice stream: evidence for dynamic ice sheet behaviour on the Canadian Shield and insights regarding the controls on ice-stream location and vigour; Boreas 32(1), 263-279, https://doi.org/10.1080/03009480310001155.

Stokes, C.R., and Clark, C.D., 2004. Evolution of late glacial ice-marginal lakes on the northwestern Canadian Shield and their influence on the location of the Dubawnt Lake palaeo-ice stream; Palaeogeography, Palaeoclimatology, Palaeoecology 215(1-2), 155-171, https://doi.org/10.1016/j.palaeo.2004.09.006.

Vérité, J., Livingstone, S.J., Ravier, E., McMartin, I., Campbell, J., Lewington, E.L.M., Dewald, N., Clark, C.D., Sole, A.J., and Storrar, R.D., 2024. Conceptual model for the formation of bedforms along subglacial meltwater corridors (SMCs) by variable ice-water-bed interactions; Earth Surface Processes and Landforms 49(1), 170-196, https://doi.org/10.1016/j.epsl.2023.118510.

---

## Author Comment (AC1)

**Response to reviewers**

We are grateful for receiving three detailed and constructive comments, including two reviews (Marion McKenzie and an anonymous reviewer) and one open comment by a group of colleagues (Isabelle McMartin et al.). The main issue raised has been an inclusion and discussion of chronological constraints on our reconstruction. To address this, we have decided to produce an A4 version of each panel of Figure 6 and add the available chronological constraints to these detailed tiles in a new supplementary document. We provide further discussion of this issue and other comments in the detailed response below.

**Please note that the reviewer comments are posted in black and our responses are posted in blue**

**Response to Marion McKenzie**

In this manuscript, B. Stoker and others present a flowset model and cross-comparison with geomorphic data across the deglaciated northernwestern LIS to provide a vast spatial and temporal analysis of ice streaming and retreat over the Bølling–Allerød and Younger Dryas periods. This work presents compelling evidence supporting collapse of the CIS-LIS ice saddle contributing to increased ice streaming and ice output reorganization during the Bølling–Allerød and varied styles of ice retreat following this collapse. I believe this work fills a knowledge gap in incorporating glacial geomorphic data with geochronology and ice modeling outputs for this region. However, I do think this work could be improved through further development of the interpretation and discussion sections with added supplemental materials to maintain focus in the final sections of the paper. There are some pieces in this paper that I think have merit but may not need to be included in this work specifically. Please see my structural and figure notes and thorough line comments to address areas in which this manuscript could be improved.

Thank you for your overall positive review of our manuscript and for the helpful comments you have provided. Below we provide detailed responses to the comments and describe the changes we intend to make to the manuscript.

**Specific structural comments:**

Please provide more information about the ages used to constrain the flowsets and associated glacial features to specific timeframes. Including a table with this information or providing a section in the introduction introducing dates and context for geochronological situated retreat (Stoker et al., 2022) would greatly clarify a lot of questions I had about how some of your interpretations were tied to temporal constraints.

Stoker et al. Ice flow dynamics of the northwestern Laurentide Ice Sheet

We will create a supplementary figure which will include an A4 version of each time slice panel of Figure 6 with the relevant age constraints for the depicted ice margin location and the type and age of these constraints. This will provide the reader with the necessary information without adding too much extra text to the manuscript that would distract from our main message. We are reluctant to add much more detail to the main text of the manuscript as recent publications have discussed the regional chronology in detail and the principal focus of this manuscript is the geomorphology. However, we will add a further, brief section of text clarifying how our interpretations are tied to certain chronological constraints and the potential implications of any future changes in the chronological framework.

In general, watch the use of ambiguous identifiers. Often the use of "that" "this" and "they" can get lost in a line of logic, so make sure you're being very explicit when making statements.

Thank you. This is an important point that reviewer 2 also raised and we will double-check the text and clarify these statements.

Section 3.2 should be split into two separate sections. One that describes ice streaming and the inferences made to topography and expand on the deglacial dynamics section.

The title for section 3.2 is somewhat confusing, we have renamed this to 'Deglacial ice margin retreat pattern' to better describe that we are simply reconstructing ice margin retreat patterns and not dynamics.

Section 3.3 could be renamed to "Ice margin during retreat" to clarify that you are describing the ice margin boundary you developed and then the one you used for temporal constraints.

We will amend this section to avoid any confusion. The intention of section 3.3 is purely to provide a brief overview of the NADI-1 ice margin chronology and highlight why we opt to use it as the framework for our ice flow reconstruction. While section 3.2 was purely to describe our process of identifying the former ice margin retreat pattern as depicted in Figure 3 and 5.

Section 5.1: for this first section to be an overview of ice flow interpretation, I don't think the evidence from your results is connected enough. You make statements about ice behavior referencing your figures without tying in the significance of cut-through dynamics or the different types of flow sets you used to make these inferences. I see you do this throughout section 5.2 and am not completely convinced the overview in 5.1 adds to the readability and competes with the merit showcased in 5.2. I would suggest either switching the order of sections 5.1 and 5.2 or simplifying/removing 5.1 and combining it with section 6, which essentially restates a lot of the overview but with better supported evidence.

Stoker et al. Ice flow dynamics of the northwestern Laurentide Ice Sheet

Thank you for the suggestion. We agree that it is probably more appropriate to move this to the end of section 5 and simplifying it to provide a summary of the ice flow interpretation.

I would consider renaming your section 5.2 headers. I understand the interest in continuity across sections, but I think readability and interest may increase if you give headings related to why the time slices are split the way they are. For example: "5.2.2 Local LGM (17.5 to 17.0 ka)" or "5.2.3 Ice stream network reorganization (16.5 to 16.0 ka)".

We will adopt the suggested subheading style.

Section 6.4: I feel like this section does not incorporate many novel results that you have not already discussed in this paper. I am fine with the 6.3 sections on ice retreat variation but feel like a one to two paragraph summary of this work could be incorporate prior to section 6.5 Additionally, in the 6.4.2 section on the Canadian Shield, this ends up being a discussion of local responses and climate responses to the Younger Dryas and no evidence is presented from the Canadian Shield. If you keep this section, I encourage including information about basal conditions on hard bed surfaces that would impact a slow in ice movement. You also state ice stream slowing is not a result of geology changes, but that directly contradicts what you say in the introduction to section 6.4 and is counterintuitive to the 6.4.2 section header. You address this well in the first paragraph of 6.4.3 so I suggest you remove 6.4.1 and 6.4.2 and focus on results from 6.4 intro and 6.4.3.

Thank you for the comment, it is clear that the key message of 6.4 is not as clear as we had hoped. The key aim of section 6.4 is to guide the reader through Figure 11, describing the changes in ice streaming, explaining the hypothesised mechanisms for these changes and speculating on the possible implications for ice sheet mass balance and ice margin retreat rates during these variations in ice stream activity. This section will be rewritten to better fulfil this aim, following your suggestions.

The conclusion could be much more concise—highlight the main findings without naming the specific evidence.

We will amend the conclusion as suggested.

**Figure and table comments:**

Figure 1E: If you're going to create a hypothetical flowline model with example flowsets, I would either make a schematic of factors used to determine flowset styles in parts A-D or develop a hypothetical model that includes the same description for all flowset types (maybe with some extra text or drawn geomorphic features on the figure to explain differences). This may also help clarify the differences between event and ice stream flow sets which I believe needs a bit more explanation.

Stoker et al. Ice flow dynamics of the northwestern Laurentide Ice Sheet

We will add further annotations to Figure 1E to help define event and ice stream flow sets and further clarifying edits will be made to the main text to highlight the differences between these flowsets.

Table 2: "Ice marginal position" should be renamed to "Deposition process" as you are describing how these features are developed.

Done.

In the terminal moraine section of "Ice marginal position" I recommend you refine the statement "deposition" to "deposition by bulldozing or transit to the margin" because the process of "deposition" is very broad.

Done.

Can you give some examples of what you mean by "a combination of processes" even if it's just to mention plucking, meltwater redistribution, etc.

Hummocky terrain is a catch-all term that covers a landform composed of multiple distinct features and forming from a range of processes. The description of hummocky terrain will be significantly expanded to highlight the range of features it includes and the associated processes better. For example, raised, flat surfaces interpreted as ice-walled lake plains indicating localised ice stagnation processes, sharp-crested ridges within the controlled moraine which represent readvances of the ice margin.

Figure 3. Can you increase the contrast of the DEM? It is hard to identify the individual features you have mapped.

The visibility of some of the features is limited due to their small size and the necessity of having a large area covered by this figure panel to highlight our approach of defining ice margins. Unfortunately changing the contrast of the DEM does not help in showing any further detail.

Please also clarify the definitions between esker complex and ridges (i.e., individual eskers could not be identified and rather there is a complex network of meltwater features) and your identification system for major vs. minor moraine crests.

We follow definitions outlined in Dulfer et al. (2022) and we will describe these in the text. Esker ridges are mapped where there is a clearly defined single esker ridge feature that can be depicted by a polyline, while esker complex is used to describe networks of anastomosing eskers or single esker ridges with a complex morphology that are better depicted with a polygon. Major moraines are polygons features mapped when the moraine is >200m wide and minor moraines are polyline features mapped when the moraine is <200m wide.

Perhaps you could zoom in on some of these features or provide specific DEM examples in Table 2.

We will provide a further column in Table 2 with DEM examples of these features.

Figure 4: it is difficult to see the difference between colors of deglacial and inferred deglacial flowsets. I understand the draw of using two near-similar colors because of the similarity in flowset formation, but especially for colorblind readers, I can imagine this difference would be too minuscule to be able to visualize.

We will continue using blue to depict the deglacial flowsets but will use a greater contrast between the blue colour used for these flowsets.

Figure 6: I appreciate what I'm sure was a considerable amount of time and effort in developing Figure 6 – that is an incredible amount of data to visually represent across such a vast spatial area. My only suggestion would be to possibly increase contrast between flowset and ice extent colors – sometimes the flowsets are difficult to see over the intense blue of the ice and the dark DEM underneath. I also have trouble reading the flowset numbers at times, I would consider creating a close-up map of some highly congested areas (section I) in a supplemental figure.

As previously mentioned, we will create a supplemental version of each panel presented in Figure 6. Each supplemental figure will be created at A4 size to increase the readability and will include the age constraints which relate to the specific timestep. We will also increase the transparency of the basemap and ice to increase readability.

Figure 7: This is great – I appreciate the visual representation here. I would consider making this figure supplemental, though. It is not entirely central to the argument you make in the section it is discussed.

Agreed. We will move this figure to the supplement.

Figure 8 seems to be non-essential to the arguments made in this work. I would recommend combining Figure 8 and Figure 5 to relate concepts of ice-margin interactions with glacial lakes or move this figure to a supplemental file.

Figure 8 is important in providing the reader with context on the location and timing of glacial lakes across the study region and aids in answering one of our key questions (what were the controls on ice streaming?). Without this figure, our claims relating glacial lake presence/absence to changes in ice streaming is less supported. We attempted to combine the glacial lake distribution with Figure 5 but unfortunately the figure becomes too cluttered with these sets of information combined. Additionally, the disconnect between the newly created ice margins and the glacial lake reconstruction (adopted from Dyke et al. 2003) was confusing. So we will retain Figure 8 as it is.

Figure 9 caption: Mackenzie is spelled incorrectly in (B). Clarify what you mean by "topography".

Stoker et al. Ice flow dynamics of the northwestern Laurentide Ice Sheet

Thank you, we have corrected the spelling. By topography we simply mean that ice flow follows the orientation of the valley and ice flow indicators are located around a topographic obstacle. We will include this in the text.

Figure 10D: what is a "zig-zag esker"?

'Zig-zag eskers' is used to describe eskers with a zig-zag shaped planform and that may also be referred to as concertina eskers (e.g. Storrar et al., 2015).

**Line comments:**

Line 14: Suggest change from "it" to "this retreat" or something similar.

This will be changed to 'This ice sheet sector...'

Line 32: clarify calving for marine-terminating ice systems, the first part of the statement could refer to terrestrial and marine ice streaming.

This will be changed.

Line 41: suggest change from "it" to "Laurentide Ice Sheet" could read as ambiguously referring to the North American Ice Sheet complex.

This will be changed.

Line 48: clarify the ablation area of the northwestern sector.

This will be clarified to 'ablation area in the region of the Cordilleran-Laurentide ice saddle'.

Line 49: Direct evidence for the statement in the first half of this sentence?

The evidence for this statement is based solely on the numerical modelling simulations referenced at the end of this sentence.

Line 57: Suggest add "geomorphic-based evidence of ice stream activity"

This will be changed.

Line 83: Clarify "the LIS and CIS ice sheets through the saddle"

This will be changed.

Line 115, remove commas around undated

This will be changed.

Line 126: "Range and were dammed […]"

This will be changed.

Line 132: Suggest "has led […]"

Stoker et al. Ice flow dynamics of the northwestern Laurentide Ice Sheet

This will be changed.

Line 134: add parentheses timing of Younger Dryas Stade for context

This will be changed.

Line 136: here is this argument still incorporating an early 30ka maximum, or are they arguing a total maximum later, at 20ka? Please clarify this point as these two comparative sentences are not exactly congruent.

The more recent reconstructions do not include a maximum extent at 30ka. We will clarify this in the text.

Lines 150 and 152: The ice sheet wide and across the entire ice sheet in the same sentence is redundant.

This will be changed.

Line 155: Reference figure 1 after "Smoking Hills-Horton River area"

We will add this reference.

Line 160: Cordilleran to CIS

Done.

Line 169: Clarify whether the "uniform mapping approach" was a manually conducted mapping effort or if there were automated tools involved or machine learning approaches to landscape analysis.

The mapping approach was done entirely manually by two separate mappers. We used the term 'uniform mapping approach' to highlight that we mapped the same landforms, scale, and using the same datasets across the entire region, meaning there are no spatial biases in our data. We will clarify this in the text.

Line 184: clarify what you mean by "morphology of the flowset". Do you mean here you are identifying the types of streamlined bedforms? Elongation ratios? Orientation and parallel conformity?

We will clarify this in the text. The morphology of the flowset refers to the overall shape of the flowset of grouped lineations, for example, whether they display any diverging or converging patterns, an hourglass shape, etc. We also describe the morphology of and classify the lineations within the flowsets, but do not define elongation ratios specifically.

Lines 194-197: The comparison between these three sentences is a little difficult to follow. I would add a contrasting argument before introducing the inferred deglacial flowset (i.e., "Conversely, *inferred* deglacial flowsets […]" and in the final sentence

Stoker et al. Ice flow dynamics of the northwestern Laurentide Ice Sheet

describing the fan-shaped lineations, I would suggest saying "The proposed ice-marginal formation of inferred deglacial flowsets is based on […]".

We will adopt these suggested changes.

Line 198: There needs to be more clarification on the difference between ice stream flowsets and event flowsets and possible overlap between the two. Both have abrupt lateral margins, both could occur on the interior, both may be overprinted, and the elongated nature of the event landforms is unclear. I would choose several classifying characteristics for each of the flowset types and make sure you identify the characteristics for all the flowsets so that they may all be directly contrasted and compared.

You raise an important point that we will clarify in the text. Our flowset classification follows that of Kleman et al. (2006) whereby flowsets are first divided based on whether the ice flow event that formed them was fast (ice stream flowset) or slow (event and deglacial flowsets). Following this, there was a secondary division of slow flow regime flowsets depending on whether they formed near to the ice sheet margin (deglacial) or towards the interior of the ice sheet (event). As such, an ice stream flowset can form in both an 'event' or 'deglacial' position. While the term 'event' can be somewhat confusing we do not want to redefine terminology where we can avoid it and so we follow the naming conventions of Kleman et al. (2006).

Line 198: I would also like more clarification on the name "event" for these flowsets. Is this suggesting that these flowsets were developed very quickly in a singular streaming event and were then discontinued? This should be clarified in the definitions.

Thank you for the comment. I hope this has been clarified in the response to the above point and we will explain this in the text.

Line 277: Consider looking at ICE-D and AskICE-D for standardized and recalculated cosmogenic nuclide exposure dates – this is a global dataset that has many CIS and LIS ages from published work that can be compared.

Thank you for the suggestion. The ICE-D database highlights some of the issues relating to changing exposure age calculation methods, including how different production rate, scaling methods, etc might impact the calculated age.

Line 285: I commend the authors on making all shapefiles available from the paper.

Thank you.

Line 297: You name deglacial flowsets as the second-most prevalent flowset type yet have not named the first most prevalent type yet. I would make this clarification or reorder the presentation of flowsets otherwise this statement seems out of place.

Stoker et al. Ice flow dynamics of the northwestern Laurentide Ice Sheet

We will reorder the presentation of flowsets so that 'inferred deglacial' flowsets, the most prevalent flowset type, is described first.

Lines 302-307: This interpretation of topographically influenced streaming seems like it may fit better in discussion where you will have more room to justify this argument. Based on what is currently in these sentences, I am not quite clear on how you are making this interpretation and if the bedforms in the foothills of the Mackenzie Hills are more affected by topography than those on the Northern Interior Plains.

This section is meant as a brief overview of the setting of each flowset type and how it varies across the region. The topographic influence described is purely a product of the greater topographic relief present in the foothills. We will rewrite this to better describe the observed patterns of topographic funnelling down valleys and the deflection of ice flow around bedrock obstacles and to include less interpretation.

Additionally, you distinguish between drumlins and mega-scale glacial lineations but did not explicitly state your classification guidelines in the methods. I would assume you used Clark et al's 2010 identification of a 10:1 elongation ratio difference between the two bedform types, but if that is the case, then I would mention this somewhere in the methods.

You are correct, we will include this definition in the text.

The sentence between lines 305 and 307 is difficult to follow. I would recommend splitting this up into more than one statement.

We will rephrase this to:

In the central Mackenzie Valley, north-oriented flowsets were formed when the ice surface slope was the dominant control on ice flow direction underneath a thick ice sheet during the local LGM. While the adjacent south-oriented flowsets record the increasing importance of the topographic relief to funnel ice flow up the Mackenzie Valley during deglaciation, as the ice sheet thinned.

Lines 318-319: Again, an interpretation of the influence of local topography on diverging flow patterns.

We will describe the pattern of topographic influence (funnelling of ice flow by the topography) rather than the vague interpretation we previously stated.

Line 329: You use the term "topographic influence" broadly several times in this results section without necessarily defining it or naming the different influences you identify (e.g., funneling from valleys, divergence of flow around bumps in the bed, or separation between flowsets by ridges) please provide examples and be specific when discussing your identified flowsets. Please see McKenzie et al., 2022 for resources regarding topographic and lithologic influence and streamlined subglacial bedforms and

Stoker et al. Ice flow dynamics of the northwestern Laurentide Ice Sheet

McKenzie et al., 2023 for evidence of subglacial bump influence on streamlined subglacial bedform morphologies.

Thank you for the suggestion. We will revisit this section and where we use vague interpretations we will clarify the individual influences we identify based on the suggested references.

Line 340: This relates back to another comment about event flowsets, but what do you mean by "different events" in this context. Please provide examples (i.e. surging events, subglacial lake outbursts providing lubrication to the bed, etc.).

We hope this was clarified in the previous comments.

Line 341: please provide the "n=" value for all datasets, not just the unclassified flowsets.

This will be included.

Line 370: cite the "previous studies" you are referring in this line.

This will be done.

Line 375: I would clarify "Terrestrial ice-contact landforms" here. There are areas where marine-terminating ice contact landforms are visible at the surface from landscape evolution, so I would just be abundantly clear.

This will be done.

Lines 406-407: Claiming that the geomorphological evidence does not support ice stream activity during the same time as the Amundsen Ice Stream feels rather unsupported. This would take geochronological data to support the timing statement. Your argument of deglacial facies overprinting ice streaming is fine, but here is a good opportunity to pull in your findings of the most common flowset being the deglacial flowsets.

We do not intend to suggest that there was no other ice stream activity during the advance phase and we will clarify this. Instead, we wanted to highlight how any advance phase ice streams have been overprinted by the deglacial record. The presence of the Cameron Hills fragment supports the presence of other advance ice streams and it's fragmented nature, with the only remaining evidence being preserved at high elevation, supports that the hypothesis that deglacial flow has removed the majority of evidence for advance phase ice streams. We have included the Cameron Hills fragment description to justify this claim and will better explain this in the text.

Line 414: I think further supporting this sentence would make it stronger (i.e., "[...] with periods of slow retreat as seen from the presence of periodic recessional moraines and some larger moraine crests [...]").

Stoker et al. Ice flow dynamics of the northwestern Laurentide Ice Sheet

This will be changed as suggested.

Line 414: Do you mean meltwater drainage by ice drainage, or do you mean ice drainage through ice stream networks? Make this clearer.

Ice drainage through the ice flow network, we will clarify this.

Line 415: "these changes" be clearer, are you talking about the periods of rapid ice loss or long-term stabilization that could have caused further erosion/transport to the margin?

'These changes' is in reference to both the periods of rapid retreat and of stabilisation. We will amend this to clarify and state 'to the variations in the rate of deglaciation'.

Line 420: Were likely not active at the same time as each other or at the same time as the Amundsen Gulf Stream during early deglaciation? Make the connection between this sentence and the prior more explicit.

Likely not active at the same time as each other. This sentence is distinct from the previous sentence.

Line 425: I understand the connections here, but I think you need to make them a bit more explicit to readers. I understand how you determine which flow came first through crosscutting relationships, but to make it clear this occurred as a result of the Bølling–Allerød, I suggest framing the argument like "Without the source from the ice saddle, the ice streams of this region began to primarily receive input from the Keewatin Ice Dome to the west as seen in the northerly flowsets transitioning to northwesterly and eventually westerly-facing signatures of ice flow." You do this in later lines but I would move this up to the start of this argument.

We will amend this as suggested.

Line 444: Connect to the flowset data – potential to include "which is seen to be associated with more truncated streamlined subglacial bedforms and inferred decrease in ice flow speed and subglacial sedimentation organization (McKenzie et al., 2022)."

We will amend this as suggested.

Line 445: the last paragraph in this section could be moved to an introduction between header 5.2 and 5.2.1 to make it more fluid for readers.

We will move this section as suggested.

Line 470: multiple younger flowsets?

We will amend this as suggested.

Line 509: I would break the sentence after "deglaciation" – it took me a few times of reading this to make sense of the additions to the first statement.

This will be amended as suggested.

Line 513: Add "between 16.5 to 16.0 ka" to the end of this sentence for clarity of that is the time slice in which you're referring.

This will be included.

Line 521: Change "this" to "the later flowset"

We will change this as suggested.

Line 523: Change "collapse of the ice saddle during the Bølling–Allerød" to "collapse of the ice saddle, which occurred during the Bølling–Allerød" for clarity.

We will make the suggested change.

Line 529-532: This statement is either not well supported or could be written better. If you are stating that the ice margin shows a slower ice flow regime that transitions to a faster ice flow regime as you *spatially* move upstream into the Mackenzie Valley, then please clarify you're referring to the spatial variability across the region that was occurring simultaneously. If you instead are stating that the slow ice flow regime becomes a faster flowing ice regime at this single location over time, I think that statement needs more support, specifically in stating how your flowsets capture that variability (e.g., because more and less elongate elongate features co-exist in single ice stream systems (McKenzie et al., 2022)).

The statement will be clarified. We are referring to the first situation you describe where the slower to faster flow regime transition occur spatially upstream.

Line 565: What do you mean by "margin retreat was active"? Please clarify if you're referring to sedimentary processes deforming the bed near the margin or timesteps of retreat or something else.

Active ice margin retreat is characterised by the stepwise, time-transgressive retreat of the ice sheet margin (often resulting in the formation of terminal moraines) as opposed to the widespread stagnation of the ice sheet margin by the process of surface melting with little active ice motion (as commonly associated with 'hummocky terrain'). We will clarify this further within the introduction to this section.

Line 622: I suggest this be changed to "overprinted on flowsets derived directly from the Great Slave Ice Stream."

We will make the suggested change.

Line 629: "topographic complexity led to complex cross-cutting flow patterns" – maybe clarify what you mean by "topographic complexity" to reduce the use of the word complex in this sentence.

Stoker et al. Ice flow dynamics of the northwestern Laurentide Ice Sheet

We will clarify this to say 'large variations in topographic relief'.

Line 633: What does "this" refer to? The topographic complexity, the complex cross-cutting relationships, or the esker? Please clarify.

This refers to the separation of the ice lobes and will be clarified in the text.

Line 638: In the beginning of this section, you name the geochronological tool used to determine the timeslice. I recommend you do this somewhere in all other 5.2 sections. I assumed it was all using cosmogenic nuclides, but after it being explicitly stated only here, now I am not sure.

We will include a description of the chronological constraints in each timeslice section.

Line 705: References at the end of this sentence?

References will be added.

Line 720: At the end of this sentence maybe add something to the effect of "but it is unclear whether this mechanism of drawdown is strong enough to weaken the entire ice-saddle" to better tie this observation to the following paragraph and the opposing arguments.

We will include this suggestion.

Line 723: Be clearer with the word "topography" – a topographic high? Of what size?

'a topographic high of ~700m prominence' will be added.

Line 752: Provide examples of what regional stagnation would look like in the deglacial record like your examples for active margin retreat.

We do not see any examples of regional stagnation across the study region, instead only seeing localised signatures of ice stagnation which are displayed in Figure 10B and D. So, we cannot include any examples of regional ice stagnation.

Line 771: What about the hummocky terrain? If there are any other possible explanations for lack of ice marginal landforms, these should be presented here as well.

This section was unnecessarily oversimplified and following comments from McMartin et al (see open comment), we will expand upon this section to highlight the wealth of evidence and arguments for active ice margin retreat across the Canadian Shield to the east of our study area.

Line 798: Evidence that these moraines are from the end of the Bølling–Allerød?

These moraines fall between cosmogenic nuclide exposure ages and radiocarbon dates within the Mackenzie Valley and cosmogenic nuclide exposure ages on the Canadian

Stoker et al. Ice flow dynamics of the northwestern Laurentide Ice Sheet

Shield. The newly proposed supplemental figure will highlight these ages to support this statement.

Line 829: The "Instead" at the beginning of this line makes the argument confusing because you used a "but" previously. Please make these two sentences clearer.

We will remove the reference to ice margin retreat processes beyond our study area to avoid any confusion.

Also, please provide evidence or citations for the flowsets you use to assume lower retreat rates and slower ice velocities (are the flowsets less elongate?).

The lower retreat rates are principally based on the ice margin chronology of Dalton et al. (2022) and the chronological constraints depicted on the new supplemental figure will support that. The slower ice velocities are demonstrated by the transition to deglacial flowsets and lineations across the Canadian Shield which indicate a broad-scale 'sheet' flow across the whole ice sheet sector, with the absence of any ice streams.

Line 853: Just say "We reconstruct both extensive and shorter time transgressive […]"

We will amend this as suggested.

Line 867-869: This mention of crevasse-fill ridge networks could use some clarification. How does the presence of these features indicate surging behavior?

We use crevasse-fill ridge corridors as indicative of the shutdown of a surging ice lobe as suggested by Evans et al. (2016).

Evans, D.J.A., Storrar, R.D. and Rea, B.R., 2016. Crevasse-squeeze ridge corridors: diagnostic features of late-stage palaeo ice stream activity. Geomorphology, 258, pp.40-50

Line 875: I would add a "However' at the beginning of this sentence because these statements contradict each other. Also add a reference at the end of this sentence.

We will amend this as suggested.

Line 889: Expand on the explanation for this. Maybe include something like "allowing for basal shear stress to increase and stabilize the ice during retreat and slow streaming."

We will expand on the mechanisms for this change as you suggest.

Line 908: clarify that "they" refers to geological conditions?

We will clarify this as you suggest.

Line 921-922: This is not a complete statement. Please clarify this sentence.

This will be amended from:

Stoker et al. Ice flow dynamics of the northwestern Laurentide Ice Sheet

Rapid ice drawdown during the peak in ice stream activity during the early Bølling–Allerød meant that the thin ice sheet profile on the Canadian Shield and low driving stresses

To:

During the early Bølling–Allerød, the peak in ice stream activity resulted in the rapid drawdown of ice in the interior of the ice sheet and caused a thin ice sheet profile over the Canadian Shield with low driving stresses.

Line 961-965: These lines contain a lot of statements that contradict each other. I would simplify this to say ice streaming occurs across the Canadian Shield only after xyz circumstances are met. I would also make sure you don't mention earlier that there is not ice streaming on the Canadian Shield because there is some back and forth in this and previous sections.

We will make the changes you suggest.

Line 968: I take issue with naming "glacial lakes" as a more important control on ice stream formation than subglacial geology. What is the *mechanism* that is the stronger control? Because if this were marine-terminating, I argue it would be the same pattern, so it's not the lakes but perhaps the onset of crevasse-driven ice loss, or increased ice breakage from loss of buttressing.

This is correct and we will amend the text to better refer to the development of a calving margin as the mechanism for driving ice streaming relating to glacial lakes. Due to our focus on the onshore record, we do not reconstruct any marine-terminating margins, but it is misleading not to highlight the possible similarities to lake-terminating margins.

Also, how does this tie into the topography? Can you say anything about the role of topography in relation to the geologic control. I think it could also be argued that the lakes are a function of topography because the topo has allowed for lakes to develop, so does your argument inherently agree with Winsborrow et al., 2010's argument that topography has a higher control on ice behavior than geology?

We will highlight the possible influence of topography on ice streaming in the text. The extent to which glacial lake development is a function of topography is difficult to determine for this region. The glacial lake location is a controlled by the GIA response of the topography to deglaciation, the retreat of the ice margin and the incision of spillways. While the topography in this situation may play a role in controlling ice behaviour, we opt to avoid these discussions due to the uncertainties.

---

## Author Comment (AC2)

Stoker et al. Ice flow dynamics of the northwestern Laurentide Ice Sheet

**Response to reviewers**

We are grateful for receiving three detailed and constructive comments, including two reviews (Marion McKenzie and an anonymous reviewer) and one open comment by a group of colleagues (Isabelle McMartin et al.). The main issue raised has been an inclusion and discussion of chronological constraints on our reconstruction. To address this, we have decided to produce an A4 version of each panel of Figure 6 and add the available chronological constraints to these detailed tiles in a new supplementary document. We provide further discussion of this issue and other comments in the detailed response below.

**Please note that the reviewer comments are posted in black and our responses are posted in blue**

**Response to reviewer 2**

The paper by Stoker et al. provides a detailed assessment of the geomophic features eroded and deposited by the NW sector of the LIS. The authors provide detailed flowlines of the LIS from the LGM through the last deglaciation and from these reconstruction attempt to interpret the style of deglaciation of this sector of the ice sheet. Overall, I find what the authors have done to be worthwhile and useful for interpreting how this sector of the LIS transgressed from the LGM to end of YD. However, I find it difficult to ascertain how the authors have arrived at their conclusions about the time-transgressive nature of the LIS demise without the chronologic information provided alongside their reconstructions. To me, this is the major weakness of the paper and without this information it does not allow the reader the ability to properly assess their interpretations, or at least easily assess it without digging through other publications and comparing and contrasting. I think with this information provided in the figures and within some of the text, it will improve the paper substantially and give readers a very nice assessment of the history of this part of the LIS alongside their geomorphic mapping and interpretations. My detailed comments are provided in the accompanying PDF with other suggestions for improving the manuscript.

Thank you for your overall positive review of our manuscript and for the constructive comments. Below we provide detailed responses to the comments and describe the changes we intend to make to the manuscript.

Line by line comments:

**Page 1:**

**Line 14** - Clarify what "it" means in this sentence. This will help with clarity for the reader.

This will be clarified to 'the northwestern Laurentide Ice Sheet'.

Stoker et al. Ice flow dynamics of the northwestern Laurentide Ice Sheet

**Line 18** - Define the period over which you mean for the reader.

This will be added.

**Page 2:**

**Line 41** - I am not sure what the authors mean here.. Are the authors saying it (the LIS) attained a similar area as the modern AIS?

In any case, I don't think the extent really matters for why the ice sheet provides a test case for studying ice streams. I think the opportunity is simply that there are geomorphic features that may indicate paleo-ice streams and that alone makes this a nice place to study ice streams.

In this sentence 'it' referred to the Laurentide Ice Sheet. We will amend the text to clarify.

**Line 43** - In the abstract it says the local LGM was at 17.5 ka. Here the authors say the maximum was during the global LGM which is defined in Clark et al. 2009 and does not extend to 17.5ka. I suggest the authors clarify this so the readers are not confused.

In the abstract we refer to the local LGM of the northwestern sector. The 'maximum' in line 43 refers to the maximum extent of the Laurentide Ice Sheet, not an individual sector. We will amend this to clarify.

**Line 46** - I think the authors should not attempt to re-classify LGM. Clark et al. 2009 has done a nice job of defining this and I think to not confuse the readers the authors should use local LGM which they do in the abstract. Redefining will confuse people and "local" allows flexibility in the regional definition.

We will adopt this terminology as recommended.

**Line 47 and 48, relating to 'numerical modelling'** - Provide a citation

We will include a reference here to Gregoire et al. (2016).

**Line 48, relating to 'Bolling-Allerod'** - Define the time period here.

We will include the time period.

**Line 48, relating to 'ablation area'** - Of the LIS or all NH ice sheets? Be explicit for the reader.

We will clarify that this refers to the western Laurentide Ice Sheet and the ice saddle with the Cordilleran Ice Sheet.

**Line 50 to line 53** - This link the authors are calling on is a little hard to decipher. Perhaps in this sentence less is more and the authors can simply end the sentence at "....13.5 and 11.5 ka."

Stoker et al. Ice flow dynamics of the northwestern Laurentide Ice Sheet

This is confusing. The BA interval pre-dates this interval, so how is this connected?

Pico et al. (2019) do not place the collapse of the ice saddle within the BA interval and instead suggest it occurs much later, during the Younger Dryas. We will clarify this in the text.

**Page 4:**

**Line 80** - These questions are great to provide for the reader but I think equally important is provide the reader with the hypotheses that are being tested as well in this paper.

In general, we agree that hypothesis-based research provides the best framework to help guide readers through the manuscript. But, in our situation we believe that some of our research questions have clear hypotheses (e.g. question 1) while some do not (e.g. question 2). Developing hypotheses for each question would require us to split some of the research questions into multiple elements and would become overly complicated and confusing for the reader. So we opt to retain only the research questions.

**Page 6:**

**Line 136, relating to 'a short-lived'** - I am confused with this part of the sentence. Do you mean it was short lived? Can you just say how long it remained at it's maximum position?

Within this sentence we provide to dates of the maximum extent (~20ka) and the approximate timing of deglaciation (~18ka). We believe this is sufficient for the readers to determine the length of time this ice sheet sector remained at its maximum extent. Especially compared to the extended duration proposed in the previous sentence (~30ka to ~22ka).

**Page 7:**

**Line 195** - It's not clear to me how they were associated. Is this defined elsewhere? Is it done through cross cutting relations or simply if eskers and/or moraines are near flow sets?

The criteria used to define flowsets is outlined in Table 1.

**Page 13:**

**Line 261, relating to 'all-time maximum'** - These are redundant.

This statement is not redundant. It is used to clarify that no glaciations prior to the LGM were larger.

**Page 17:**

Stoker et al. Ice flow dynamics of the northwestern Laurentide Ice Sheet

**Line 353** - It isn't clear what "matches" means. Do the authors mean agreement? Or, is it something else.? If 64% match, does this mean that 36% are mismatched and therefore agreement is not very good? Some clarity here would help the reader.

The term 'match' is a purely subjective measure of whether flowsets from our reconstruction were also identified in other reconstruction. While a 36% omission of flowsets might sound significant, as we state in the text, the majority of the flowsets that we newly identify are small in size and only add detail to our reconstruction.

**Page 20:**

**Line 408** - I think it's also important to cite the original chronologic work that provided the timing of this age. See references within the Dalton et al. paper. This is important for attributing the original people who did thie work and not just the compilation paper of Dalton et la.

We will add these references.

**Page 21:**

**Line 451** - I am finding it very difficult to see how the authors are able to parse out the ice sheet changes in 500 year intervals when the chronology in this region doesn't allow for it. I understand the authors' assumptions and can appreciate what they are trying to do, but laying out the structure in this detailed manner (to me) is giving the impression to the reader to much confidence in the chronology and relating to the geomorphic structures. Perhaps other experts will be okay with this but I think there might be too much inferring to make these detailed reconstructions at 500 year time steps. This of course is the central crux of this paper but I would prefer the authors be more conservative with their reconstructions and at least bin them into intervals of 1000s of years and not in these 500 year snapshots which I don't find convincing without more detailed chronology. See my comments on Figure 6 regarding this section.

The reconstruction of ice flow changes itself is not based on the chronological data. Instead, a relative chronology of ice flow changes is based on the observed cross-cutting locations, as detailed in the supplementary table and in the supplementary flowset map figure. This relative chronology is then fit to the chronological framework of Dalton et al. (2022). As is visible in the figures, there are dramatic variations in ice flow direction and dynamics that occur even at shorter timescales than our 500-year windows depict. This meant in some figures it was necessary to highlight two separate flow directions within one figure panel (e.g. Figure 6I). For example, over ~1,000 years at the start of the Bølling–Allerød there is a complete 180∘ reversal in ice flow direction in the Mackenzie Valley at ~65N. If we were to use 1,000 year timesteps then the complexity of these variations would not be depicted. These changes, recorded by cross-cutting relationships (as shown in detail in Figure 9), are key to identify the

changing dominance of different ice source regions. While new chronological constraints or methodological improvements change the exact timing of our reconstructed ice dynamics, we believe that the sequence of events and how the ice flow dynamics relate to various ice margin positions is robust.

Providing the chronologic constraints on the figure would go a long ways in giving the reader a better sense of how the authors are deriving these reconstructions. It will also provide the readers with all the information. I understand they are providing the published 500 years intervals from Dalton et la. but I still think providing the location and ages that those reconstructions are derived is important for this more detailed study.

There has been extensive discussion of the chronology of the western Laurentide Ice Sheet in recent years, we do not seek to recite these discussions here as the manuscript is already quite long. To avoid adding further detail on to the small panels of Figure 6, we will produce a supplementary figure that recreates figure 6 with each panel at A4 size and we will include the chronological constraints on these panels. There will also be a brief sentence added for each timeslice description in section 5.2 to highlight the geochronological constraints that are available.

**Line 454 and 455** - Cite Dalton et al. here who also noted this lack of information from 25-17ka based on lack of chronology as well.

This will be included.

**Page 29:**

**Line 675** - To make this figure more useful to the reader, I think the authors should provide the location and chronologic constraints for their interpretation on the map (ie dates from whence the data came from and location).. The reader shouldn't have to go back and forth between this paper and the Dalton et al. work to do this., and I think the authors would be making a clearer point about their assumptions by providing the ages on this plot.

As detailed in the previous comment, we will include this information on a supplementary figure.

**Page 33:**

**Line 686** - This is beautiful DEM but I am not sure it is necessary for the main body of the manuscript. I would consider moving it to supplemental.

We will move this to the supplementary information.

**Page 34:**

**Line 697** - This figure is interesting but I'm not sure it is necessary for the main body of the paper since the paper is largely about the ice history and not the lakes.

We believe that this figure should remain in the main text. One of our key aims is to understand the controls and drivers of ice stream activity and one of these factors that we investigate is glacial lakes. We believe it is important to show the location of the main glacial lakes in the region for the reader. It was not possible to include this information on one of the other figures.

**Page 35:**

**Line 703** - Relative to what? This should be clarified.

We will re-write this sentence to clarify that there is more rapid thinning in the ice saddle zone compared to the Keewatin Ice Dome.

**Line 710** - What is meant by strong? Clarify for the reader.

This will be changed to 'northwards ice flow signal'.

**Line 719** - This may very well be the case but I think it is important to provide the reader with the information about why "envisaging" this is difficult.

We will amend this to clarify that the Amundsen Gulf Ice Stream was likely important due to it's large size and likely high calving rates at it's marine-terminating margin.

**Line 726 to 728** - Again, I think it is very important to show some of these dates with the new reconstructions. It provide the reader with enough information to make their own assessment of what the authors are arguing.

These constraints will be detailed on the previously described supplementary figure.

**Page 37:**

**Line 741** - Without the ages to constrain the geomorphic features, I cannot provide much feedback in this part of the discussion. I understand that the authors have used the reconstructions from Dalton et al. but it would be beneficial to see those ages on the figures in this paper and to evaluate what the authors have interpreted. Otherwise, I am not sure it's possible to provide meaningful feedback in this section.

As we previously mentioned, the chronological constraints will be marked on a supplementary figure. However, we do not believe the chronological data is essential to determine whether the nature of the ice margin retreat processes. Instead, the reconstruction of the process of ice margin retreat builds upon a series of previous publications that rely principally on analysis of the ice marginal geomorphological signature, without any mention of chronological data (e.g. Dyke and Evans, 2003; Evans et al. 2021; Livingstone et al., 2020; McMartin et al., 2021; Sharpe et al., 2021).

**Page 40:**

**Figure 10D** - What is a "zig-zag' esker?

'Zig-zag eskers' is used to describe eskers with a zig-zag shaped planform and that may also be referred to as concertina eskers (e.g. Storrar et al., 2015).

**Page 43:**

**Line 923 to 925** - Hard to ascertain without the age information in the figure.

The retreat of the ice margin on to the Canadian Shield is directly constrained by two sites dated with cosmogenic nuclide exposure dating (Reyes et al., 2020). This will be detailed on the supplementary figure.

**Page 44:**

**Line 942** - I'm not sure this section is necessary and seems to just be a review of what drives ice streams.

This section is intended to highlight some of the other factors that may have influenced ice stream activity and were not described in the previous section due to the difficulty to test them within our study. It aims to provide an element of transparency to the reader and highlight factors that future studies could attempt to better test.

**Page 45:**

**Line 969** - This section also seems tangential to the main points of the paper. I appreciate these sections but they seems to distract from the main thrust of the paper.

Agreed. This will be removed and replaced with a sentence in the main portion of the discussion to highlight this as an uncertainty.

**Line 977** - 'empirical record is difficult and we are thus'

We will make this correction.

**Line 985 and 986** - Again, and sorry to go on about this, but I find statements like this hard to justify without more information on the chronologic basis of the reconstruction of this paper. Without the ages clearly outlined for the reader, I find most of the descriptions in the text about the geomorphology general fine but without any basis. It is hard for me to assess this paper without that information and difficult to provide constructive feedback to the authors.

We will include the age constraints on a supplementary figure that will illustrate how these changes in ice stream activity are tied to both radiocarbon and cosmogenic nuclide exposure ages. We will also include a brief section that will highlight the uncertainties in the chronology and what the implications of changing the chronology would have for our interpretations.

**Page 46:**

**Figure 11** - I don't find this figure adds too much to the paper's main focus. I would consider removing to help streamline the paper and focus it on the main points as outlined in the conclusion.

We disagree. This figure is key to understanding the controls on changing ice stream activity and underpins section 6.4. The key finding that relates the peak in ice stream activity to climate-driven changes in the ice sheet surface slope is dependent on this figure. We will amend the text to better highlight the importance of this figure and interpretation.

**Page 48:**

**Line 1049 to 1050** - Good to acknowledge the original people but better if they were included meaningfully in the paper.

In any case, can the authors acknowledge specific people they worked with during the project (ie tribal members, leaders, or councilors)?  Otherwise, I personally find these statements performative as a person with indigenous heritage- e.g. the nations are recognized but were never asked for permission to map and describe their lands.  I could say more but I leave this for the authors' to consider and apologize if one of the coauthors is a tribal member and I overlooked their affiliations and/or heritage.

Thank you for this comment and for raising an important issue. We agree that these statements can be performative if they are empty and without any thought or action. The research permitting system for the Northwest Territories is principally focused on field-based studies, with no permitting system to facilitate communication with First Nations communities for remote studies. However, during the course of our research in the region, we have been involved in multiple applications to the Aurora Research Institute for fieldwork permits. These applications are predominantly focused on the field component of our research but do also describe our intentions to undertake remote-sensing, geomorphological mapping work. Following the publication of this research it is our intention to submit a one-page summary report to the Aurora Research Institute and interested local communities (e.g. the Sahtu Renewable Resources Board, who have previously expressed an interest in this work).

---

## Author Comment (AC3)

Stoker et al. Ice flow dynamics of the northwestern Laurentide Ice Sheet

**Response to reviewers**

We are grateful for receiving three detailed and constructive comments, including two reviews (Marion McKenzie and an anonymous reviewer) and one open comment by a group of colleagues (Isabelle McMartin et al.). The main issue raised has been an inclusion and discussion of chronological constraints on our reconstruction. To address this, we have decided to produce an A4 version of each panel of Figure 6 and add the available chronological constraints to these detailed tiles in a new supplementary document. We provide further discussion of this issue and other comments in the detailed response below.

**Please note that the reviewer comments are posted in black and our responses are posted in blue**

**Response to Isabelle McMartin and others**

Dear authors

Reconstruction of ice flowsets in the northwestern sector of the Laurentide Ice Sheet is important to increase our understanding of ice sheet dynamics in this understudied region of Arctic Canada. You have done a lot of work integrating results from high-resolution remote geomorphic mapping (Dulfer et al., 2023) with the new ice marginal chronology of Dalton et al. (2023). Previous regional reconstructions of ice flow dynamics and ice streams were considered but we are concerned about the lack of consideration for some field-based observations available from regional surficial geology maps, mainly published by the Geological Survey of Canada. Field-based evidence such as striations, till fabrics, and erratic distributions offer direct indicators of past ice movements. Their integration could serve to validate the ice flowsets identified or introduce nuances to the interpretations drawn from geomorphic mapping alone. Please find details in the attached pdf where we briefly present these concerns (and a few others) that could impact some of your interpretations.

Regards,

Isabelle McMartin, Etienne Brouard, Janet Campbell and Pierre-Marc Godbout

We would like to thank these researchers for taking the time to read and provide feedback on our manuscript and for their overall positive comments. We have copied all comments into this document, and we address individual comments below. We would like to acknowledge that some field-based evidence was missed in error and we are grateful that you have brought these publications to our attention so that they may be properly consulted and referenced. At the same time, we would like to highlight that we have made efforts to consult multiple lines of field-based evidence while building and verifying our reconstruction. This includes the recent surficial mapping work from the GSC around Great Slave Lake and the till fabrics described therein (e.g. Hagedorn et al.,

2022; Paulen and Smith, 2022) or recently published reconstructions (e.g. Evans et al., 2021). We do not directly include these data in our reconstruction but instead seek to use them to validate our reconstruction and outline our reasons for this approach in detail below. To allow the reader to more effectively compare and validate our reconstruction against the pre-existing data we will create a compilation of selected geomorphology (ice flow indicators and moraines) from previously published works. This compilation figure will form a secondary panel of Figure 1. This will also highlight the progress in surficial mapping that has occurred since Fulton (1995) that is currently displayed in Figure 1.

Field record of ice-flow indicators

The "striation" symbol appearing on regional surficial geology maps includes glacial striae and any other small-scale erosional forms on bedrock (crescentic fractures and gouges, rat tails, nail head striae, stoss and-lee topography, etc., e.g. McMartin and Paulen, 2009). These features provide information on direction, sense, and relative age of ice flows, and on the record of older glaciations, commonly poorly preserved in the geomorphic record. The general orientation of roches moutonnées is also represented by a different symbol on the maps and provides an additional record of the ice flow direction.

It is mentioned in Section 3.4 that "Glacial striations also provide an opportunity for reconstructing former ice flow patterns and may preserve older flow traces on bedrock outcrops where abrasion was limited during deglaciation (Kleman er al., 1990)". However, we are concerned about the lack of consideration and integration between some of the field-based record and the remotely mapped geomorphology to inform and resolve the flowset directions and age relationships, particularly in areas with complex flow patterns during deglaciation. As mentioned in Section 2, the "understanding of ice flow dynamics … suffers from the disconnected nature of studies at varying scales". Although the ice-flow indicator record can be very detailed locally, striations indicated on surficial geology maps are often distributed regionally and can inform on former and deglacial ice flow patterns where no landforms are preserved. Field-based ice-flow indicators can easily be extracted from surficial geology maps or Open File reports, all available in digital format (using Advanced search in https://ostrnrcandostrncan.canada.ca/home), and compiled at the ice sheet sector scale. Such large-scale compilations have been completed recently directly east of the studied area in the west-central Keewatin sector of the LIS (Brouard et al., 2022; Fig. 5 below) and further east along Hudson Bay (Behnia et al., 2020; McMartin et al., 2021; DataS2c below) to help constrain the regional glacial history.

[Figure]

Figure 5. Location of ice-flow measurements in west-central Keewatin and their associated ice-flow orientation (From Brouard et al., 2022).

[Figure]

DataS2c. Generalized ice-flow indicator map, Central Mainland Nunavut (From McMartin et al., 2021).

We have chosen not to incorporate ice flow indicators from mapped glacial striation directly in our reconstruction because maps of striations are not available consistently across our study area. The surficial geological maps where the data is recorded are

produced at a wide range of scales (1:50,000 to 1:250,000), some are produced solely as a 'reconnaissance' map from remote sensing data (Kerr, 2014 and further references in the section below discussing map resolution), and some regions are currently unmapped. As such, the integration of this data would not be compatible with our aim to produce a reconstruction that is consistent across the whole study area. A high-quality, broad-scale compilation (such as the recently completed Brouard et al., 2022) would be very welcome for our study region and would allow for the integration of striation data into our reconstruction. But, unfortunately the map area of Brouard et al. (2022) falls outside of our study area and the creation of a compilation like this is beyond the scope of our study. However, during the production of the glacial geomorphological map in Dulfer et al. (2023) we did consult and verify our geomorphological map against the existing surficial geological maps for the region and in the publication we described the accuracy and completeness of our map compared to these data sources.

Interpreted ice-flow indicators and glacial histories from some regional field-based studies were considered (e.g. Smoking Hills), but other available regional maps and Open File reports should also be verified in light of the proposed reconstruction. In this regard, an ice-flow data compilation in the Great Bear Magmatic Zone (Normandeau and McMartin, 2013; Fig. 4 and Appendix II) should be taken into consideration, as well as a regional deglacial history reconstruction in the northeastern Horton Plain region, particularly for the interpretation of a NE(?) flowset (Fs-230) and other relative ages of flowsets 224, 225, 231, 323 (St-Onge and McMartin, 1995 and references herein).

Thank you for drawing our attention to these reconstructions (Normandeau and McMartin, 2013; St-Onge and McMartin, 1995), they were omitted in error and a comparison to these reconstructions will be made in the revised manuscript.

- A note on our work process and the justification of it:

Throughout this research, from the creation of the geomorphological map presented in Dulfer et al. (2023) to the ice flow reconstruction, we made efforts to verify our work against the pre-existing work. The final step in the creation of the geomorphological map involved the comparison of our mapped landforms to the available surficial geological maps produced by the GSC to ensure a reliable end product, but we did not seek to directly include any features from the surficial maps that we could not independently verify. This approach allowed us to verify and validate our map to ensure that any inaccuracies or limitations in our map were identified while maintaining a consistent map product across the entire area.

During the ice flow reconstruction, we followed a similar approach of verifying our reconstruction against the available regional ice flow reconstructions. The intention of this was to highlight any potential shortcomings of our work or to identify where our

reconstruction has allowed us to add further detail to the existing knowledge. Unfortunately, due to the scale of the region some key references were missed and we regret the omission of these from the manuscript. Thank you for highlighting these studies and we will make sure to properly cite them in the updated manuscript.

However, the guiding principle throughout our process was to produce a (relatively) high-resolution reconstruction of the changing ice flow dynamics during the deglaciation at a consistent scale and level of detail across the whole study region. As such, we did not seek to directly include any pre-existing reconstructions within ours, as this would lead to spatial inconsistencies in the reconstruction. This meant that even data from co-authors on this paper (see Evans et al., 2021) was not directly integrated into this reconstruction, but instead referenced and highlighted as a more detailed regional study. This allowed us to produce a consistent reconstruction across the ice sheet sector and then to make comparisons between our reconstruction and the pre-existing data to validate our efforts or highlight issues. This includes published papers (e.g. Evans et al., 2021) and also GSC maps and reports where possible (e.g. Hagedorn et al., 2022; Hagedorn 2022 and other examples).

We also suggest a comparative analysis with Dyke and Prest (1987), the only other reconstruction offering a continental overview that shows ice-flow drainage and incorporates the extensive fieldwork data accomplished in Canada since the end of the 19th century. This comparison could provide a richer context for this study, especially in contrast to works primarily based on remote mapping. While the cost-effective approach outlined in this study is certainly practical, a more thorough integration of fieldwork data might offer a comprehensive perspective, reducing the reliance on subjective interpretations, particularly regarding the provision of relative chronology through striations.

Thank you for your comment. During the preparation of the geomorphological map produced in Dulfer et al. (2023) and in the reconstruction produced here comparisons were made to Dyke and Prest (1987). The relatively broad-scale nature of Dyke and Prest (1987) ultimately meant that it did not contribute much to our reconstruction or provide any new details and we opted not to discuss this further within the text.

We are unsure what is meant by 'subjective interpretations'. Beginning from the mapping procedure, two mappers independently mapped and verified each others mapping to ensure the reproducibility of the map product. The relative chronology developed here is based on the same geologic principles that underpin reconstructions based on striations and was carried out multiple times, independently by two separate researchers (Helen Dulfer and Ben Stoker). Additionally, all cross-cutting relationships, the details of the relative chronology and the justification of choices is provided in both the manuscript and in the supplementary table.

Throughout the whole process we prioritised transparency and reproducibility, hence the provision of all shapefiles, locations of identified cross-cuts, etc. in the supplementary material. We believe our methodology has produced a robust reconstruction and our transparency will allow future researchers to independently verify our mapping and interpretation using desk-based, low-cost methods or easily integrate future detailed field studies to improve upon our reconstruction. The integration of striation data is not within the aim of our reconstruction due to spatial inconsistencies in the scale of mapping and the absence of striation mapping from entire portions of our study area.

In conclusion, we follow an established flowset mapping approach and we have sought to be as thorough and transparent in our approach as possible. To achieve this we explain in detail throughout the text our choices, include supplementary files with extensive information on our flowsets, including the location of cross-cuts that inform our relative chronology.

Regional stagnation to the east of the studied area

In section 6.2.3, it says "Therefore, we suggest that widespread stagnation had not begun for the northwestern LIS margin to the west of 210°W but may have occurred further to the east, as documented by Sharpe et al. (2021)." First, it should probably be 110°W instead of 210°W, and in the Conclusions on page 47 again: "If widespread stagnation of the margin occurred, we suggest that it must have occurred after the LIS retreated to the east of 110°W."

Thank you for the note, it is correct that it should read 110°W.

More importantly, no basis for the assumption that widespread stagnation may have occurred further east of the studied area is provided, other than what is presented in Sharpe et al. (2021). Recent and current high-resolution geomorphic mapping using ArcticDEM and extensive field-based studies from east of 108°W to the Hudson Bay coast does not support or indicate widespread stagnation (Campbell and Eagles, 2014; Campbell et al., 2016, 2018, 2019, 2020; Lauzon and Campbell, 2018; Livingstone et al., 2020; McMartin et al., 2021; Brouard et al., 2022, 2023; Dyke and Campbell, 2022; Vérité et al., 2024). In contrast, numerous evidence for a sequential, time-transgressive retreat is presented in these publications. If the authors of this study have not mapped east of 110°W, they should not assume or propose that a widespread stagnation characterized ice retreat east of their studied area unless a proper discussion with alternative views is presented.

We agree with this comment and will amend the text to better reflect the differing views on ice margin retreat processes across the Canadian Shield. The formulation of the sentence was not intended to imply that widespread stagnation does occur to the east of 110°W. Rather, it was our intention to describe the ice marginal retreat processes we

reconstruct to the west of 110°W and then we contrast these processes to the ice marginal processes documented by Sharpe et al. (2021) to highlight that we have not observed similar evidence. As we had not mapped to the east of 110°W, we wanted to remain ambiguous about the processes which occurred in this region. Unfortunately, the wording is a little clumsy and does not accurately represent this. It will be re-written to include the references you suggest and to highlight the evidence for time-transgressive, dynamic margin retreat, which matches with the processes we observe in our study area.

Ice streams on the Canadian Shield

At Line 961, it says: "While hard-bed conditions are known to exert a stabilising influence on ice flow, large ice streams are observed across the Canadian Shield." Yes, further east in Keewatin over the Shield, large ice stream footprints have been reconstructed from geomorphic mapping (including high-resolution mapping with ArcticDEM), including not only the Dubawnt Lake ice stream (e.g. Stokes and Clark, 2003, 2004) but several other large ice streams (Margold et al., 2015, 2018; McMartin et al., 2021) and not necessarily involving glacial lakes (i.e. marine-terminating ice streams).

We thank the authors for their suggestion. We agree that it is probably an important point to highlight that glacial lakes themselves that act as the direct trigger for ice streaming, but the presence of a calving margin which can occur at lake or marine terminating margins. We reference the Dubawnt Lake ice stream as it represents the first observed reactivation of ice streaming on the Canadian Shield for this specific ice stream sector. In Stokes and Clark (2003, 2004), it is suggested that the development of glacial lakes acted as a trigger for ice stream. As such, we use the Dubawnt Lake Ice Stream purely as an example to highlight that although ice streaming is reduced on the hard-bedded Shield, it is still possible, and to gain an insight into the broader controls on ice streaming. The examples you raise (e.g. McClintock Channel Ice Stream, McMartin et al., 2021) also provide good examples of hard-bedded ice streams and will be integrated into the text to provide further depth to our discussion.

At Line 965, it says that "The development of the ice-marginal Dubawnt Lake to the east of our study is hypothesised to have triggered the Dubawnt Lake Ice Stream… (Stokes and Clark, 2003, 2004)". This is incorrect. Stokes and Clark conclude that "As the ice margin retreats back, early glacial lakes form in the Thelon and Back River drainage basins. These early glacial lakes in the Thelon River basin increased in size and deepened, triggering the Dubawnt Lake ice stream." They further suggest that "This results in widespread thinning of the ice sheet and produces a large glacial lake to the south of the ice stream, dammed by the Dubawnt Lake ice stream lobe ". This glacial lake is west of Dubawnt Lake (Stokes and Clark, 2004; Fig 8).

Stoker et al. Ice flow dynamics of the northwestern Laurentide Ice Sheet

You are right. We will correct this within the text.

Another point concerns the influence of crystalline bedrock on ice-margin stability. The absence of ice-flow landforms on the Canadian Shield could be attributed to several factors. 1) The scarcity of sediments associated with harder-to-erode crystalline bedrock may hinder landform production. Thus, the absence of ice-flow landforms on crystalline bedrock does not imply the absence of ice streaming (especially on the eastern part of McConnell Lake); it merely indicates that no landforms were preserved. 2) Because crystalline bedrock is more challenging to erode, ice-flow landforms that directly record bedrock are generally smaller and likely too small to be noticed at the scale mapped. Utilizing ArcticDEM at a more detailed scale, such as 1:20,000 or 1:10,000, might have revealed these features.

> We are confident that we have captured all large-scale ice flow (e.g. ice streams) across our map area, including on the Canadian Shield, that is recorded in the glacial landform record for the following reasons:
>
> 1) Our flowset coverage highlights landforms are observed over much of the region, including the eastern part of glacial lake McConnell. While it is true that the geomorphological imprint of ice streams on hard-beds would be different to that of ice streams on soft-beds, we do not imply an absence of ice streams based purely on the absence of landforms (as we have landforms mapped), nor solely by the elongation of these landforms. Instead, the broad-scale, diverging sheet flow that we observe on the Canadian Shield is used to infer the absence of ice streaming. There is evidence of ice flow across much of the Canadian Shield in our study area, but none of this ice flow evidence indicates any large-scale ice streaming activity (e.g. hourglass-shaped, converging and diverging flow patterns).
>
> 2) While we state in Dulfer et al. (2023) that mapping was performed at 1:50,000 to 1:100,000, this was a conservative estimate used to guarantee the minimum level of detail provided by our map. In areas where landforms were absent, a more detailed mapping scale was adopted to verify that the absence of landforms wasn't an artefact of the mapping scale. In addition to this, all mapping was compared to available surficial geological maps to ensure that there were no large areas of ice flow indicators absent from the map. Indeed, this comparison did not identify any large omissions in our map.

ArcticDEM and high-resolution mapping

While discussing the benefits of high-resolution data for understanding glacial history, the authors did not actually map at the highest spatial resolution provided by ArcticDEM (2 m). Instead, they mapped at scales ranging from 1:50,000 to 1:100,000 (Dulfer et al.,

2023) to cover as much area as possible. This mapping scale is similar to, or even lower in resolution than that in studies that used aerial photographs (1:15,000 to 1:60,000). The advantage of ArcticDEM lies in its complementary data, which enable the production of hillshades at a higher resolution than previously available (about 20-30 m resolution). While we understand the necessity of cost-effective methods, it appears there was an aim to produce a regional scale product with supposedly high-resolution data, yet the execution was only partially completed. The authors should be cautious with their claims of using high-resolution data, as their practice does not match their assertion.

As previously mentioned, the map scale provided (1:50,000 to 1:100,000) is a conservative estimate to best highlight the minimum map product provided, mapping of course occurred at much higher scales in many locations.

You highlight our mapping scale is lower than the resolution of aerial photographs but the resolution of surficial map products produced by the GSC is not always produced at the true resolution of aerial photographs. In total there are 82 NTS map tiles covered by our reconstruction. As of the publication of Dulfer et al. (2023), 8 of these NTS tiles are unmapped by the GSC, 1 tile is mapped at 1:50,000, 45 tiles are mapped at 1:125,000, 20 are mapped at 1:250,000, 2 full tiles and ¾ of an NTS tile are mapped at 1:100,000. Finally, 7 tiles are only covered by district-scale surficial geological maps at 1:500,000 scale (Craig, 1960; Rampton, 1988). In comparison, our mapping is presented at a much higher resolution for the vast majority of the study region. Although, some individual surficial geological map tiles do exceed the resolution of our map, especially the recently completed surficial geological mapping around the Great Slave Lake region (Hagedorn et al., 2022; Paulen and Smith, 2022). As such, we believe it is a fair comment to say that our reconstruction is at a high-resolution.

In addition to the range of mapping scales, the mapping procedure employed is not consistent across the entire region. Some NTS tiles mapped by the GSC are only covered by reconnaissance mapping efforts, relying on remote sensing data and limited field evidence (i.e. similar procedures to this study, see Table 1 in this document, below this comment). These reconnaissance maps are vital to providing total coverage of the area. Some of the striation data is taken from unpublished datasets, preventing these data from being scrutinised (e.g. Kerr, 2018).

Throughout the production of the Dulfer et al. (2023) map, the surficial geological maps from the region were compiled and extensively compared to our map product to assess the effectiveness and reliability of our mapping procedure.

Table 1: A brief summary of some of the variations in mapping procedure across existing surficial maps

| Reference | Map scale | Fieldwork? | Data source |
|---|---|---|---|
| Kerr (2014, 2018) | 1:60,000 | 'Limited fieldwork', some striation data is based on previously unpublished work | Airphoto interpretation |
| Kerr (2022) | 1:60,000 | None | Airphoto interpretation |
| Kerr and O'Neill (2017) | 1:125,000 | Striation information taken from older publications | Airphoto interpretation |
| Kerr and O'Neill (2018a and b, 2019a and b) | 1:125,000 | Striation data from recent publications | Airphoto interpretation |
| Kerr and O'Neill (2019c) | 1:70,000 | Striations from older publications (Craig, 1960) | Airphoto interpretation |
| Kerr and O'Neill (2020 and 2021) | 1:60,000 | No fieldwork | Airphoto interpretation |
| Kerr et al (2014) | 1:60,000 | Limited fieldwork | Airphoto interpretation |
| Kerr et al (2016) | 1:70,000 | No fieldwork | Airphoto interpretation |
| Olthof et al (2014) | 1:125,000 | No fieldwork but takes striation data from Kerr (1990) which only partially covers NTS tile 85-P | Predictive mapping based on LandSat data |

Uncertainties

Furthermore, the authors acknowledge a 1000-year uncertainty commonly associated with cosmogenic (10Be) ages, which is foundational to the study's framework for ice-margin retreat. While this initial recognition is valuable, a more thorough discussion on how this uncertainty influences subsequent interpretations, particularly regarding suggested peak ice-stream activity at the onset of the Bølling Allerød interstadial, would greatly enhance the narrative's robustness. Given the acknowledged uncertainties and the inherently subjective nature of both the approach and Dalton's ice-margin reconstruction, a more detailed exploration of these aspects could help strengthen the study's conclusions. While we agree with the authors' assertion that the Bølling-Allerød

warming significantly influenced ice sheet melt and dynamic changes, further elucidation on how these uncertainties were navigated in reaching such conclusions would provide clearer insight into the analytical rigor applied throughout the study.

Due to the length of the paper, which is already quite long, we are reluctant to add much greater discussion on the chronology beyond what was mentioned in previous papers (e.g. Stoker et al., 2022; Dalton et al., 2023). However, you raise an important point regarding how the chronology might influence our conclusions and I will provide further information on why we do not think it will impact out conclusions. Within the text we will provide a brief paragraph summarising the influence of age calculation on our reconstruction and why we do not believe the chronology will significantly change in a manner that would impact our conclusions.

- Age calculation approach:

In brief, there are three methods of calculating exposure ages that have been used for the northwestern Laurentide Ice Sheet. I will explain simply the implications of these methods on the calculated age, rather than any reasoning behind the calculation method (covered in Stoker et al., 2022).

The exposure age calculation approach by Dalton et al. (2022) falls in the middle of the age calculations, so provides a middle ground in reference to the other two approaches. The calculation approach employed by Reyes et al. (2022) results in exposure ages that are approximately 500 years older than the reconstruction of Dalton et al. (2023). In contrast, the calculation approach used in Stoker et al. (2022) results in exposure ages that are approximately 500 years younger than the approach of Dalton et al. (2023). In essence, this means that the peak in ice streaming would occur approximately one timestep earlier if the ages were calculated following the Reyes et al. (2022) approach or one timestep later if following the Stoker et al. (2022) approach.

This means that (based on Figure 11 in the manuscript) using the age calculation method of Reyes et al (2022) the peak in ice stream activity would already occur immediately prior to the Bølling–Allerød. In the reconstruction of Dalton et al. (2023), ice stream activity slowly increases immediately prior to the Bølling–Allerød but then peaks at the start of the Bølling–Allerød before slowing down towards the end of this period. Based on the age calculation approach presented in Stoker et al. (2022), the peak in ice stream activity would be situated later in the Bølling–Allerød and the slowdown in ice stream activity would still occur during the Bølling–Allerød.

We opt to use the Dalton et al. (2023) reconstruction as it manages to effectively combine all pre-existing constraints (exposure ages, luminescence ages and radiocarbon ages) and it represents the current best guess reconstruction of the ice sheet chronology. This multi-chronometer method means that we can be fairly certain that the chronology will not change significantly, as all existing age constraints are

satisfied and it is not solely dependent on cosmogenic nuclide exposure ages. As outlined in Stoker et al. (2022), we believe that the chronology created by Reyes et al. (2022) is slightly too old to be compatible with the pre-existing age constraints, so we reject this chronological framework. While the chronology presented in Stoker et al. (2022) is also compatible with other chronometers, the change by using this chronology would be to better align the peak in ice streaming with climatic oscillations during the Bølling–Allerød and Younger Dryas and still supports our conclusions.

Therefore, based on the Dalton et al. (2023) and Stoker et al. (2022) chronologies, we can be reasonably confident that this peak in ice streaming occurred during the Bølling–Allerød and the slowdown in ice streaming also occurred (or at least began) prior to the Younger Dryas.

- The spatial relationship between ice sheet thinning and ice stream activity:

Changes in the age calculation approach shift the reconstructed ice stream activity in time and could change the association with different climate events. However, a period of rapid ice sheet thinning is spatially linked to our reconstructed peak in ice streaming. This supports our conclusion that the mechanism driving changes in ice streaming is the steepening of the ice sheet surface slope during surface mass balance/elevation feedbacks (i.e. ice saddle collapse). Regardless of the age calculation approach, two cosmogenic nuclide dipsticks in the Mackenzie Valley region (one at 63N and one at 65N) indicate a period of rapid thinning in this region (Stoker et al., 2022). While the absolute timing of this thinning event can shift through time, the occurrence of rapid thinning at this location does not change. Therefore, we can associate the process of rapid thinning with increased ice stream activity in this region, even if we do not link these events to a climate event. Furthermore, numerical modelling studies have linked changes in the ELA and subsequent changes in the ice sheet surface slope to increased ice streaming during deglaciation (Robel and Tziperman, 2016).

We provide an extensive discussion of the uncertainties of dating in Stoker et al (2022), including an explanation of why we favour a 'younger' exposure age calculation that better fits with the pre-existing constraints in the region. In Dalton et al. (2023) there is also a clear explanation and appreciation of the uncertainties, with the exposure age calculation approach of Dalton et al. (2023) trending slightly older than the method used in Stoker et al. (2022). We do not seek to reproduce those discussions here.